# Ship-based lidar measurements for validating ASCAT-derived and ERA5 offshore wind profiles

Hugo Rubio[a,b], Daniel Hatfield[c], Charlotte Bay Hasager[d], Martin Kühn[b], and Julia Gottschall[a]

[a]Fraunhofer Institute for Wind Energy Systems IWES, 27572 Bremerhaven, Germany
[b]ForWind, Institute of Physics, Carl von Ossietzky Universität Oldenburg, Küpkersweg 70, 26129 Oldenburg, Germany
[c]C2Wind ApS, Vesterballevej 5, 7000 Fredericia, Denmark
[d]Department of Wind Energy and Systems, Technical University of Denmark, Frederiksborgvej 399, 4000 Roskilde, Denmark

**Correspondence:** Hugo Rubio (hugo.rubio1@uni-oldenburg.de) and Julia Gottschall (julia.gottschall@iwes.fraunhofer.de)

**Abstract.**

   The accurate characterization of offshore wind resources is crucial for the efficient planning and design of wind energy projects. However, the scarcity of in situ observations in marine environments requires the exploration of alternative and reliable data sources. In response to this challenge, this study presents a comprehensive comparison between wind profiles derived from the Advanced Scatterometer (ASCAT) satellite observations and the ERA5 reanalysis dataset against ship-based lidar measurements in the Northern Baltic Sea. The aim is to investigate the applicability of ship-based lidar measurements for validating these datasets and to better understand the reliability, accuracy and limitations of ASCAT- and ERA5-derived wind statistics for offshore wind characterization at wind turbines operating heights. To extrapolate ASCAT observations, a probabilistic adaptation of the Monin-Obukhov similarity theory was implemented. The comparison between the two gridded datasets, extrapolated ASCAT and ERA5, reveals an overall good agreement in average wind speeds at 100 m height, with AS-CAT exhibiting overall mean wind speeds approximately 0.6 m s$^{-1}$ higher than ERA5 across the entire study region. However, excluding regions within 40 km of the coastline reduces this bias to around 0.4 m s$^{-1}$, highlighting the negative impact of coastal contamination in ASCAT measurements and the difficulties ERA5 faces in accurately capturing wind conditions in complex coastal areas due to its coarse resolution. Validating these datasets against ship-based lidar measurements reveals a consistent underestimation of ERA5 and overestimation of ASCAT profiles. Both datasets show deteriorating performance with height, which is particularly notable in ASCAT profiles, with rapidly increasing biases above 170 m, peaking at around 0.5 m s$^{-1}$ at 270 m. This is mainly due to the limitations of the extrapolating methodology applied to ASCAT wind field measurements.

## 1 Introduction

Offshore wind energy has experienced significant growth in recent years, and this trend is expected to continue over the coming decade. Forecasts indicate that the world's installed capacity for this technology will increase from 73 GW in 2023 (International Renewable Energy Agency, 2024) to around 486 GW by the end of 2033 (Global Wind Energy Council, 2024). This rapid development of offshore wind farms, coupled with the maturation of floating technology as an alternative to fixed-bottom turbines (Wind Europe, 2021), is accelerating the demand for accurate wind observations in coastal and far offshore

areas. However, in situ wind observations at turbine-relevant heights in the marine environment are sparse in both time and space due to the constructional limitations and high installation and operational costs of traditionally employed meteorological masts (met masts).

Floating lidar systems offer a cost-efficient alternative to offshore met masts (Clifton et al., 2015), thanks to their robustness and reliability (Gottschall et al., 2017; Carbon Trust, 2018), and the potential to enhance flexibility and reduce costs of offshore measurement campaigns. Although buoy-based floating lidar systems can be relocated to different locations, they are generally used to measure at a single location during a specific period. In contrast, profiling lidar systems installed in cruising ships, in particular, are capable of providing reliable wind profile measurements over large areas. However, before ship-mounted profiling lidar systems can become a generally accepted alternative for offshore met masts and buoy-based lidars, specific challenges need to be overcome, such as the validation of these data against reference measurements and the quantification of the associated uncertainty (Rubio and Gottschall, 2022). Still, ship based lidar has already been used in different wind energy related studies. For instance, in Wolken-Möhlmann and Gottschall (2014), ship-based lidar measurements were used to measure offshore wind farm wakes. In Witha et al. (2019a); Gottschall et al. (2018); Savazzi et al. (2022), ship-borne measurements were used for validating numerical models datasets and in Pichugina et al. (2017); Rubio et al. (2022) for characterizing low-level jets in different offshore regions.

Numerical weather prediction (NWP) models in re-analyses mode are commonly used by the industry to obtain wind information in offshore regions where in situ measurements are unavailable. These models provide long-term wind time series at multiple vertical levels within the boundary layer, along with an extensive spatial coverage. However, while numerical models have demonstrated good performance in shallow-water offshore regions when compared to in situ measurements (Witha et al., 2019b; Wijnant et al., 2019), they face difficulties in areas with significant variations in surface roughness, such as coastal regions. Each NWP model comes with inherent limitations due to factors like grid resolution, physical modelling, and parameterization choices. These limitations introduce uncertainties in wind statistics derived from these datasets, which can be quantified by comparing model NWP outputs against available validation measurements. However, conducting such validation is particularly challenging in deep-water offshore regions, where in situ measurements are sparse.

Satellite remote sensing devices are a valuable additional source of wind information in these data sparse regions, providing global wind field measurements capable of capturing the horizontal wind variability over a temporal coverage exceeding 15 years. For this reason, several studies have focused on characterizing offshore wind resources using satellite measurements (Remmers et al., 2019; Ahsbahs et al., 2020; Hasager et al., 2020). One of the most well-known satellite-based instruments used for wind energy purposes is the Advanced Scatterometer (ASCAT), mounted onboard the European Space Agency´s MetOp series of polar orbiting satellites. ASCAT provides global ocean wind measurements on a 12.5 km grid spacing. However, the application of satellite measurements for wind energy purposes has been limited by two main factors. First, the limited temporal resolution of polar-orbiting satellites restricts wind measurements to a few fixed times per day, rendering these products unable to fully capture the diurnal wind speed variability. Secondly, satellite measurements are provided at 10 m above the sea surface, requiring the implementation of extrapolation methods to derive wind information at turbine operating heights.

The Baltic Sea is an area of great interests for offshore wind development due to its strong and consistent wind resource, relatively shallow water depths, and proximity to large population centres. However, it is a complex and dynamic environment,
characterized by strong land-sea interactions and atmospheric processes that generate significant wind speed and direction gradients, as well as specific mesoscale phenomena such as sea breezes and low-level jets (Smedman et al., 1997). Consequently, the Baltic Sea has been extensively studied in previous literature aiming to accurately characterize the available wind resource in the region. In Svensson (2018), numerical models and different types of measurements were used to characterize mesoscale processes. In Hasager et al. (2011); Karagali et al. (2014); Badger et al. (2016); Karagali et al. (2018), wind resource statistics
were derived from satellite measurements. In Hatfield et al. (2022), ship-based lidar measurements were extrapolated down to 10 m and compared against observations from FINO2 met mast and ASCAT, as well as against the New European Wind Atlas (NEWA) mesoscale simulations.

The objective of this paper is to assess the accuracy of ASCAT-derived wind speed profiles in nearshore and offshore locations of the Northern Baltic Sea by conducting a comprehensive comparison against ship-based lidar measurements. To
70 derive wind profiles from the ASCAT 10 m measurements, we employ the mean stability correction approach presented in Kelly and Gryning (2010) and implemented in Badger et al. (2016). For this, we utilize atmospheric stability information from the ECMWF Reanalysis 5th generation (ERA5) and compare two different collocation methods to evaluate the potential influence of the limited temporal resolution of satellite overpasses in the ASCAT extrapolated profiles. Not only the ASCAT derived wind profiles, but also the wind profiles from ERA5 are compared against the lidar profiles to evaluate and highlight
the differences in wind profiles obtained through the application of these different datasets. Furthermore, we introduce a novel collocation strategy for comparing ASCAT-derived and ERA5 profiles against the ship-mounted lidar observations, which has not been previously reported. To the authors' knowledge, this study represents the first comprehensive comparison of vertically extrapolated ASCAT wind profiles (hereafter referred to as ASCAT wind profiles) to wind turbine operational heights against non-stationary in situ measurements, covering a wide horizontal extent from nearshore to offshore locations.
Therefore, this work aims to contribute significantly to a better understanding of the reliability, limitations, and accuracy of satellite measurements derived wind statistics and ERA5 wind data for offshore wind characterization at wind energy-relevant heights.

The paper is structured as follows. Section 2 presents the ship-based lidar measurement campaign, as well as the ERA5 and ASCAT datasets used in this study, along with the implemented data processing methods. This section also provides a
85 detailed description of the mean stability correction method used for ASCAT wind extrapolation and the collocation procedure employed for the comparison of the three datasets. Section 3 contains the main results obtained in this investigation. The discussion of these findings and the main extracted conclusions are included in Sections 4 and 5, respectively.

## 2 Data and Methods

This section describes the three datasets used in this work. In addition, the methodology used for processing the different datasets is detailed, as well as the methodology to extrapolate ASCAT winds and the collocation approach used for their comparison against the ship-based lidar measurements.

### 2.1 Ship-based lidar measurements

The ship-based lidar observations used in this study were acquired through the execution of a novel ship-based lidar measurement campaign designed and conducted by the Fraunhofer Institute for Wind Energy Systems IWES (Germany). In this campaign, a wind lidar profiler was installed on-board the ferry ship *Stena Gothica*, operated by the company Stena Line, along the regular route between the harbours of Nynäshamn (Sweden) and Hanko (Finland) in the Northern Baltic Sea. Figure 1a shows the average route of the *Stena Gothica* ferry; only small deviations from this route occurred during the execution of the campaign. The ship covers this route on a daily basis, travelling from one harbour to the other within one day, and travelling back the following day. The frequency distribution of the ship location versus the hour of the day is presented in Fig. 1b. As can be observed, the ship typically remains at the harbours during the central hours of the day (from 7:00 to 17:00 UTC), while travelling from one harbour to the other between the evening and the early morning. The consistent relationship between the time of the day and the ship´s location is a particular aspect of these sort of campaigns, already observed in similar experiments such as the NEWA Ferry Lidar Experiment (Gottschall et al., 2018; Rubio et al., 2022).

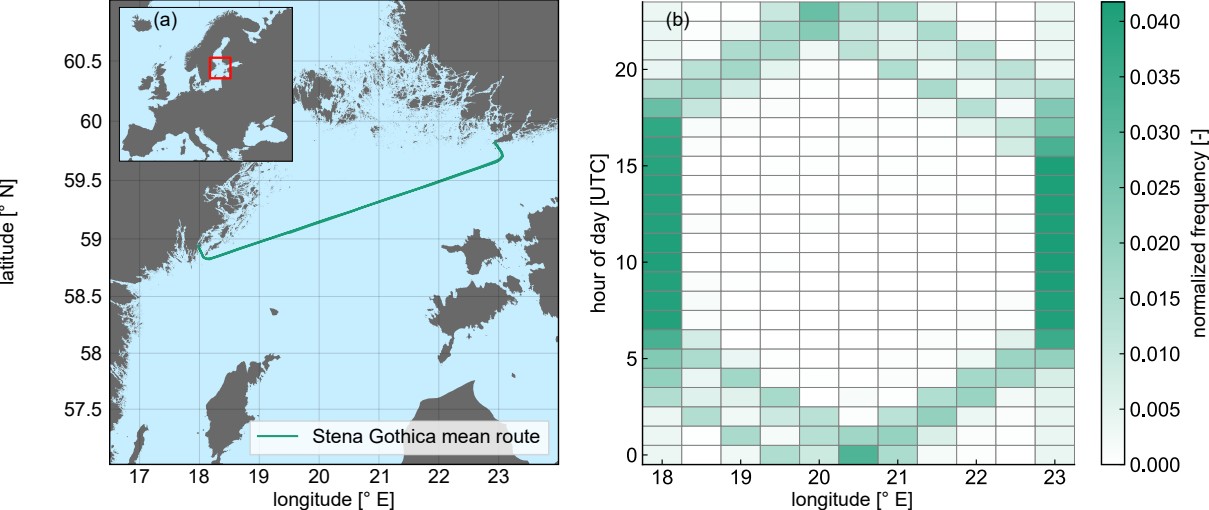

**Figure 1.** On the left panel, the mean route of the *Stena Gothica* ferry ship during the execution of the campaign. On the right, 2D histogram of the location of the ship depending on the hour of the day and the longitude of its position.

The campaign took place from 28 June 2022 to 21 February 2023 and as in Gottschall et al. (2018), the Fraunhofer IWES´s in-house developed ship-based lidar system was used. This is composed by a vertical profiling Doppler lidar WindCube WLS7

v2, from the manufacturer Vaisala, configured to measure at twelve different height levels ranging from 60 to 270 m above sea level (ASL). In addition to the lidar device, the integrated ship-based lidar system includes a motion recording unit to track the vessel motions and positions (attitude and heading reference sensor and a satellite compass) and a meteorological station to record the main meteorological parameters, including temperature, pressure, relative humidity, and precipitation.

As in previous ship-based lidar campaigns, a ship-motion compensation algorithm was implemented in order to take the motion effects out of the measurements. For this, the motion information recorded by the system is used in combination with the wind lidar measurements, applying a simplified motion correction algorithm (Wolken-Möhlmann and Gottschall, 2014). This algorithm considers the translational ship velocity and orientation, ignoring vessel tilting due to its negligible influence on the results. Additionally, lidar measurements with carrier-to-noise ratio (CNR) values below -23 dB were excluded from the final database, following the manufacturer´s recommendation to strike a balance between data availability and accuracy. Subsequently, lidar measurements and motion information (i.e. ship coordinates) were averaged into 10-minute mean values.

Figure 2 provides insights into the measured data during the campaign. In the first panel, the longitude binned wind speed at 100 m height can be observed, along with the normalized frequency of 10-minute average recordings at each longitude bin. The lowest wind speed corresponds to the longitude bin encompassing the Swedish harbour, with an average velocity of around 6.6 m s$^{-1}$. This specific location, Nynäshamn harbour, can be considered onshore due to its intricate topography, characterized by numerous small islands and hills that slow down the wind flow. In contrast, the remaining locations are characterized as offshore sites, presenting mean wind speeds above 8.5 m s$^{-1}$, with the highest mean speed observed at Hanko harbour.

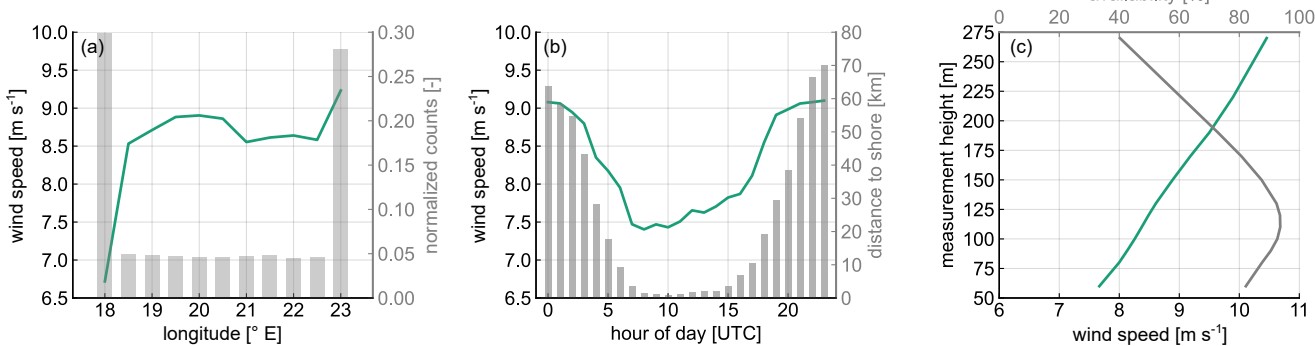

**Figure 2.** Summary of lidar measurements. (**a**) Mean wind speed at 100m height per longitude (green line) and normalized number of 10-min counts per longitude (bars). (**b**) Wind speed ship daily cycle at 100 m height (green line) and mean distance to shore per hour (bars). (**c**) Mean wind speed profile (green line) and mean availability during the campaign time extent per measurement height ASL (grey line).

Figure 2b illustrates the wind speed ship daily cycle (represented by the solid line) at 100m height and the mean distance to the shore per hour (represented by the bars). The minimum wind speeds occur during the central hours of the day, coinciding with the period when the ship is mainly located at the two harbours. Despite Hanko harbour typically presenting stronger wind speeds, the considerably lower wind velocities measured at Nynäshamn and the higher frequency of observations at this site

(refer to Fig. 2a) result in a noticeable decrease in the average wind speed during these hours. In contrast, the highest wind speeds are observed during the night and the early morning when the ship is typically in transit between the two harbours.

Finally, Fig. 2c shows the mean wind speed along the measured wind profile (green line) together with the total availability profile of the lidar over the campaign (grey line). As can be observed, there is a pronounced increase in the mean wind speed with height, going from 7.6 m s$^{-1}$ at 60 m to 10.4 m s$^{-1}$ at the top measurement height. The availability profile shows maximum values above 90 % within the range of 80 m to 130 m ASL range. Beyond 130 m, the availability drops rapidly with the height as a consequence of very clean air and low concentration of aerosols in the region and period of study. The decrease in availability at lower levels is explained by the lidar device's focus distance of around 120 m ASL. Moving further below or beyond this distance results in lower CNR values, causing measurements to be filtered out of the dataset when CNR falls below the -23 dB threshold.

## 2.2 ASCAT

The Advanced Scatterometer (ASCAT) is a space-borne remote sensing instrument that measures radar backscatter from the Earth's surface in the microwave frequency range (Martin, 2014). ASCAT was launched by the European Space Agency (ESA) onboard the Meteorological Operation (MetOp) satellites, developed and operated by the European Organization for the Exploitation of Meteorological Satellites (EUMETSAT) (Verhoef and Stoffelen, 2019). MetOp-A was the first satellite launched in October 2006, followed by MetOp-B in September 2012 and by MetOp-C in November 2018. ASCAT provides wind speed and direction measurements at 10 m above the sea surface, with a global coverage and available grid spacings of 12.5 km and 25 km (de Kloe et al., 2017). For this study, the higher spatial resolution data were selected, since it has shown better performance in previous studies when validated against in situ measurements (Verhoef and Stoffelen, 2013; Carvalho et al., 2017). This dataset is processed and distributed by EUMETSAT Ocean and Sea Ice (OSI) Satellite Application Facility (SAF) and by the Advanced Retransmission Service (EARS). Both systems are managed by the Koninklijk Nederlands Meteorologisch Instituut (KNMI) and the data were downloaded for this study using the Copernicus Marine Data Service (CMS) (product id: WIND_GLO_WIND_L3_NRT_OBSERVATIONS_012_002).

ASCAT has an effective swath width of 512.5 km with a nadir gap of 700 km, resulting in a temporal resolution of 1 to 3 overpasses daily considering both the ascending and the descending trajectories, depending on the time period and location (latitude). The number of ASCAT overpasses in the Northern Baltic Sea region during the execution of the measurement campaign is presented in Fig. 3a, whereas the diurnal distribution of the overpasses is shown in Fig.3b.

The ASCAT scatterometer is an active microwave radar that measures the backscatter power from transmitted pulses operating in the C-band frequency of 5.255 GHz. These measurements are unaffected by cloud cover and rain. The received backscatter is related to the surface roughness of the observed area. It is minimal when the surface is completely smooth, such as during calm weather conditions, and progressively increases as surface roughness intensifies. This backscatter signal is used to calculate the normalized radar cross-section (NRCS, $\sigma_0$), defined as the ratio of the received and the transmitted power, which depends on the radar settings, the atmospheric attenuation, and the ocean surface characteristics (Chelton et al., 2001). From NRCS and through the application of an empirically derived geophysical model function (GMF), the sea surface winds

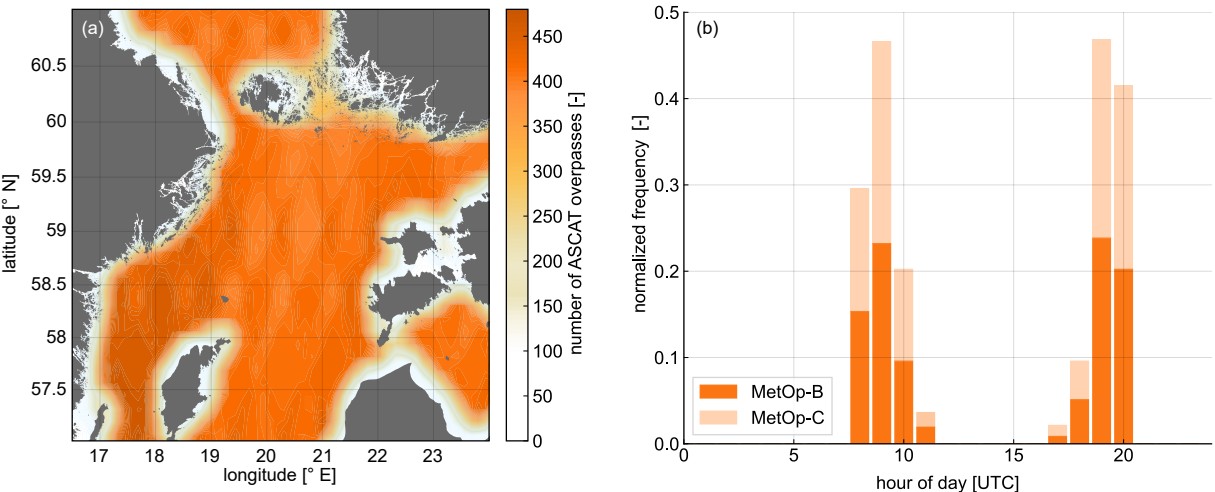

**Figure 3.** (**a**) Number of ASCAT overpasses during the duration of the campaign. (**b**) Normalized frequency of ASCAT overpasses per hour of the day.

are calculated. These empirical models are calibrated using in situ measurements of wind speed from buoys and other sources, and are validated using independent measurements from other satellite instruments and numerical models (Hersbach et al., 2007; Hersbach, 2008; Verspeek et al., 2012). The current GMF used by ASCAT is the CMOD7 (Stoffelen et al., 2017), which was developed by the ESA specifically for its use with C-band scatterometers.

The implemented ASCAT data processing for this study focused on satellite measurements retrieved during the period of the ship-based lidar measurement campaign, and included the following main steps. Firstly, a coordinate transformation was applied to transfer ASCAT coordinate points from the bottom left corner of each grid box to the centre of the box. Subsequently, a quality check was conducted by filtering out measurements based on the quality flags provided by the CMS (E.U. Copernicus Marine Service Information (CMEMS). Marine Data Store (MDS)), which account for factors such as the presence of sea

ice, extreme wind conditions (wind speeds below 3 m/s or above 30 m/s), and proximity to land. Despite the application of these quality filters, excessively high mean wind speed values were observed in ASCAT grid cells near the coast, likely due to coastal contamination effects (Stoffelen et al., 2008; Lindsley et al., 2016). To address this, an interquartile range (IQR) outlier detection method (Dekking, 2005) was employed, identifying grid boxes with unusually high wind speed values and masking them out from the analysis.

**2.3   ERA5**

ERA5 (ECMWF Reanalysis 5th generation) is the latest global atmospheric reanalysis produced by the European Centre for Medium-Range Weather Forecasts (ECMWF) (Hersbach et al., 2020). ERA5 replaces the previous reanalysis ERA-Interim (Dee et al., 2011) and is based on the latest version of the Integrated Forecasting System (IFS) model IFS Cycle 41r2. ERA5 provides hourly estimates of a wide range of atmospheric, land surface and oceanic variables with a 0.25° x 0.25° latitude-

longitude grid resolution, covering the period from 1950 to present. Additionally, ERA5 utilizes 137 model (pressure) levels extending from the surface level to the top of the atmosphere at 0.01 hPa or around 80 km height. ERA5 is produced using an assimilation scheme based on the four-dimensional variational (4D-Var) system (Bonavita et al., 2016). This method integrates modelled data from the IFS with observational data from a range of sources such as satellites (ASCAT among them), radiosondes, and aircraft widespread across the world.

For this study, the *u* and *v* wind components were downloaded for the lowest 10 model levels to calculate the horizontal wind speed and direction. Additionally, the surface sensible heat flux, air temperature at 2 m above the surface, and friction velocity parameters were also downloaded for deriving the atmospheric stability information required for ASCAT winds extrapolation (see Section 2.4). Furthermore, the ERA5 data were re-gridded to match the ASCAT wind speed maps resolution (0.125° latitude and longitude) using bilinear interpolation. It should be noted that only ERA5 data within the time frame of the measurement campaign have been used in this study.

## 2.4 Satellite vertical extrapolation

One of the main limitations of the application of satellite remote sensing measurements in the field of wind energy is that they provide wind information only at surface level. Consequently, vertical extrapolation methods must be implemented to obtain wind information at wind turbine hub heights. Several methodologies for vertical extrapolation of satellite measurements have been explored in previous literature. For example, Capps and Zender (2009, 2010) used 10-m wind measurements from QuickSCAT to estimate the global wind power potential at various vertical levels. In their approach, the Monin-Obukhov similarity theory (MOST) was implemented for the atmospheric stability correction of the vertical wind profile, using data from a global ocean-surface heat flux product and reanalysis data. Doubrawa et al. (2015) employed the equivalent neutral winds from QuickSCAT and SAR along with a neutral logarithmic profile to calculate a wind atlas in the Great Lakes region. Similarly, Badger et al. (2016) and Hasager et al. (2020) extrapolated SAR and ASCAT surface winds using the mean stability correction presented in Kelly and Gryning (2010), a method based on a probabilistic adaptation of the MOST-based wind profile. Finally, Hatfield et al. (2023) developed a machine-learning model to extrapolate ASCAT winds to wind turbine operating heights, employing 12 years of satellite wind observations in conjunction with near-surface atmospheric measurements at FINO3, and comparing the output wind profiles against both in situ measurements and numerical model data.

In this study, we employ the approach used by Badger et al. (2016) and Hasager et al. (2020) to calculate the ASCAT wind profiles. This method involves a mean correction of atmospheric stability effects, obtained from the numerical model dataset ERA5, alongside a probabilistic adaptation of the MOST-based wind profile to vertically extrapolate the satellite wind measurements. The mean stability correction factor derived from this methodology can exhibit both positive and negative stability corrections depending on the height considered, as it combines stable and unstable terms. Conversely, when applying stability correction factors to instantaneous wind speed measurements, the stable or unstable terms are applied separately.

Compared to the instantaneous stability correction approach, applying the mean stability correction avoid the need to calculate wind speeds under stability conditions and heights that fall out of the validity range of the MOST model. MOST is specifically designed to describe turbulent fluxes within the surface layer (Lange et al., 2004; Högström et al., 2006), but it has

limitations when dealing with instantaneous data analysis, particularly in stable conditions. The statistical adaptation of MOST can be effectively applied up to turbine operating heights, since the mean stability correction remains within the range where MOST is applicable. In neutral and unstable conditions, MOST can be successfully employed within the lower 200 meters of the vertical profile (Peña et al., 2008).

Additionally, employing the mean stability correction offers other potential benefits. Numerical models can accurately capture average meteorological conditions over extended periods (Peña and Hahmann, 2012), whereas the accuracy of instantaneous stability information from these datasets is questionable, introducing additional uncertainty to extrapolated profiles using this instantaneous data (Badger et al., 2012). Furthermore, while previous literature highlighted the good performance of data-based extrapolation methods (Optis et al., 2021; de Montera et al., 2022; Hatfield et al., 2023), the limited time span of the measurement campaign and the low temporal resolution of ASCAT result in an insufficient amount of data to implement these approaches in this study. Otherwise, a relevant drawback of the mean stability correction is that the information provided by the individual wind speed samples is neglected, masking the potential influence of particular mesoscale effects that modify the average wind profile.

The implementation of the mean stability correction approach is described below. This is individually executed for each of the ASCAT grid points by using the stability information from the ERA5 corresponding location. As a result of this process, one ASCAT-derived mean profile is calculated for each grid point.

The atmospheric stability can be directly accounted for by estimated the Obukhov length $L$ parameter, calculated as:

$$L = -\frac{\overline{T} u_*^3}{\kappa g \overline{w'\theta_v'}} \tag{1}$$

where $\overline{T}$ is the air temperature, $u_*$ is the friction velocity, $\kappa$ is the von Kármán constant ($\approx 0.4$), $g$ the Earth´s gravitational acceleration, $\overline{w'\theta_v'}$ the kinetic virtual heat flux, where $w'$ is the vertical component of the wind speed, and $\theta_v'$ is the virtual potential temperature. The temporal means are denoted by overbars, while fluctuations around the mean value are indicated by primes. Accurate measurements of heat and momentum fluxes require three-dimensional observations from high-frequency sonic anemometers. However, since we wish to develop an extrapolation method independent from in situ measurements, the mean temperature and heat fluxes in Eq. (1) are replaced by the ERA5 parameters air temperature at 2 m and surface sensible heat flux, respectively. Additionally, friction velocity values from ERA5 are also utilized. Positive values of the inverse Obukhov length $1/L$ denote stable atmospheric conditions, negative values indicate unstable conditions, and values around 0 indicate near-neutral stratification.

According to the formulation described in Kelly and Gryning (2010), the probability density function $P$ of $1/L$ can be estimated as:

$$P(L^{-1}) = n_\pm \frac{C_\pm}{\sigma_\pm} \frac{\exp\left[-\left(C_\pm |1/L|/\sigma_\pm\right)^{2/3}\right]}{\Gamma[1+3/2]} \tag{2}$$

where the subscripts + and - indicate the stable and unstable portions of the distribution, respectively; $n_{\pm}$ are the fractions of occurrence of each portion, $C_{\pm}$ are semi-empirical constants, and $\sigma_{\pm}$ are the scale of variations in $1/L$, based on the mean standard deviation of the surface heat flux and the average of the cube of the friction velocity, as indicated in the equation below:

$$\sigma_{\pm} = \frac{g}{\langle \overline{T} \rangle} \frac{\sqrt{\langle (\overline{w'\theta'_v} - \langle \overline{w'\theta'_v} \rangle_{\pm})^2 \rangle}}{\langle u_*^3 \rangle} \tag{3}$$

As for Eq. (1), we replace the mean temperature and heat fluxes with the corresponding parameters provided by ERA5.

In this study, the values for the $C_{\pm}$ constants have been set to 6 and 4 for the stable and unstable portions, respectively. Although previous studies focused on different datasets have used other values (e.g. both set to 3 in Badger et al. (2016); and $C_+ = 5$ and $C_- = 12$ in Optis et al. (2021)), the selection of these values for this study was based on a empirical validation, by comparing the theoretical distribution calculated from Eq. (2) against the normalized probability density (NPD) function of $1/L$ derived from ERA5. Through this process, values were chosen to ensure that the theoretical distribution closely represented the ERA5 NPD of $1/L$ across all the ASCAT grid boxes along the entire ship route. Furthermore, identical values of $C_{\pm}$ were applied to all ASCAT grid points.

Finally, the mean stability correction of the mean wind profile at a specific height $z$ is calculated as:

$$\Psi_m^* = -n_+ \frac{3\sigma_+}{C_+} b' z + n_- f_- \tag{4}$$

where $b'$ is calculated as

$$b' = \frac{b}{\Gamma[1 + 3/2]} \tag{5}$$

with $b = 4.7$ coming from the standard MOST formulation for stable conditions $\Psi_m = bz/L$ (Stull, 1988). Analogously, $f_-$ is derived from the standard MOST formulation for unstable conditions (see (Kelly and Gryning, 2010) for the exact formulation of $f_-$).

To evaluate the potential influence of the discretized temporal frequency of ASCAT overpasses, and therefore, the effect of the available stability information in the derivation of the mean stability correction factor, two different approaches have been compared. First, in the so-called collocated approach, only ERA5 stability information at times collocated with the ASCAT overpasses was considered. In the second approach, all ERA5 stability information from the entire duration of the campaign was used. The normalized probability density functions of atmospheric stability ($1/L$) derived from ERA5 at two different locations along the ship's route are shown in Fig. 4, together with the theoretical distribution calculated from Eq. (2) for the two considered approaches.

The left panels show the collocated approach, employing ERA5 stability information from timestamps coinciding with ASCAT overpasses, while the right panels depict the full campaign approach, incorporating ERA5 stability information for the

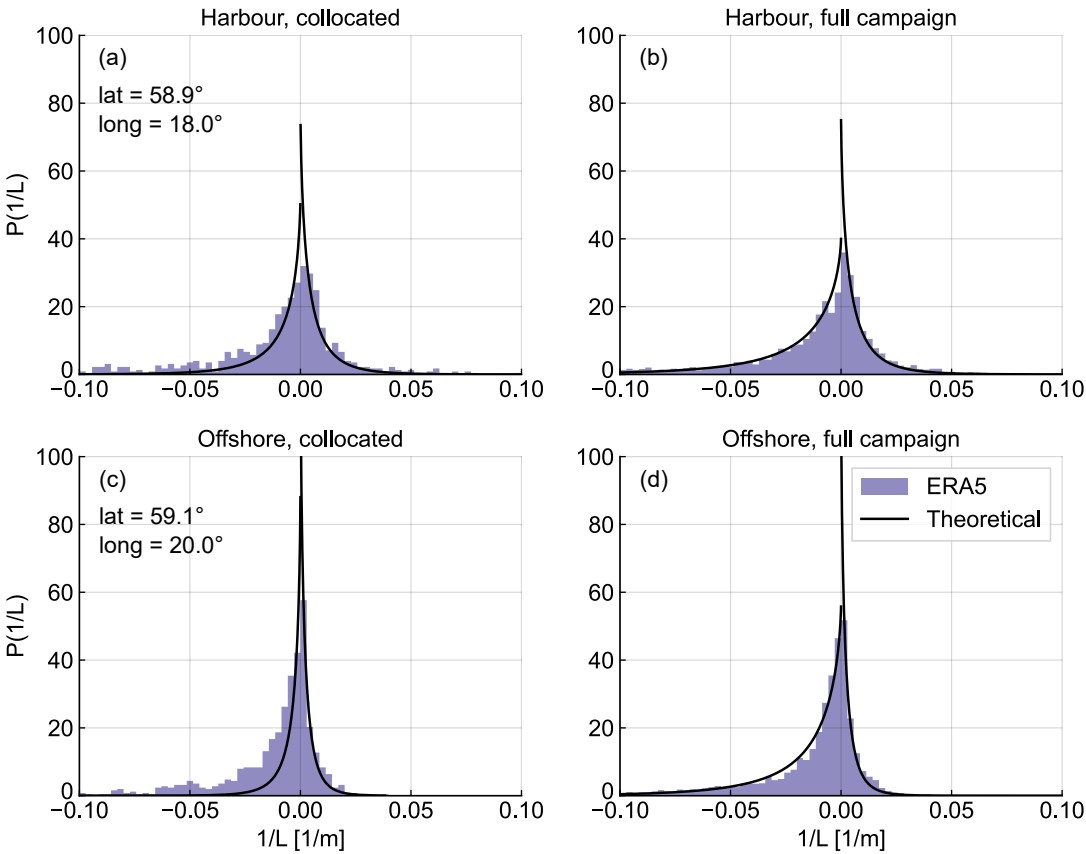

**Figure 4.** Normalized probability density functions of inverse Obukhov length $1/L$ from ERA5 and theoretical distributions calculated from Eq. (2). Left panels depict the collocated approach, employing ERA5 stability information only from timestamps coinciding with ASCAT overpasses, while right panels illustrate the full campaign approach, incorporating ERA5 stability information for the entire duration of the measurement campaign. Results are presented for two grid points: one offshore site (panels (**c**) and (**d**)) and one location near Nynäshamn harbour (panels (**a**) and (**b**)). The coordinates of these sites are indicated in panels (**a**) and (**c**), respectively.

entire duration of the measurement campaign. Results are presented for two grid points, one offshore site (panels (c) and (d)) and one location near Nynäshamn harbour (panels (a) and (b)). The coordinates of these sites are indicated in panels (a) and

275 (c), respectively.

As observed, considering the stability information from the full campaign results in a better theoretical distribution compared to the collocated approach. Although the difference is minimal at the harbour site, it is more pronounced at the offshore location, where a significant underestimation of unstable stability frequency is observed. The harbour site presents a rather symmetric distribution around zero, meaning that both unstable and stable atmospheric conditions are equally represented. However, the

offshore site exhibits a higher occurrence of unstable conditions compared to the stable side of the curve. Section 3.1 presents additional results on this matter and evaluates the differences in the obtained ASCAT wind profiles between the two approaches.

Finally, the extrapolated wind speed at any desired height $z$ can be calculated from Eq. (6) by introducing the mean stability correction $\Psi_m^*$ obtained from Eq. (4):

$$U(z) = \frac{\langle u_* \rangle}{\kappa} \left[ ln \left( \frac{z}{\langle z_0 \rangle} \right) - \Psi_m^* \right] \tag{6}$$

## 2.5  Collocation procedure

The comparison of gridded datasets (ERA5 and ASCAT) against the non-stationary measurements from the ship-based lidar system requires the implementation of a collocation methodology to ensure a fair comparison. Previous studies have already conducted comparisons between gridded data and ship-based lidar measurements (Witha et al., 2019b; Hatfield et al., 2022; Rubio et al., 2022). However, unlike previous literature that focuses on time-space collocated comparisons, in this study, ship-
based lidar measurements are compared against the mean wind profiles calculated for each of the grid points from the gridded datasets. Consequently, a novel methodology for collocating and comparing the mean gridded and lidar-measured wind profiles has been developed and is briefly introduced in this section.

After applying the coordinate transformation and re-gridding procedures explained in Sections 2.2 and 2.3, both datasets are gridded with an identical discretization, featuring a horizontal resolution of 0.125° x 0.125° and the grid points located at
the centre of the grid boxes, as shown in Fig. 5. For each grid box, the ERA5 mean profile is calculated for the period of the measurement campaign, while the mean ASCAT profile is obtained using the procedure described in Section 2.4. To obtain the mean lidar profiles for comparison, the 10-minute average ship position information is utilized to identify all the 10-minute lidar measurements captured within each grid box. Subsequently, the mean lidar profile for each grid point is calculated by averaging all the 10-minute measurements detected in the corresponding grid box. This enables the comparison of all ERA5
and ASCAT grid boxes with their respective mean wind profiles against the collocated "gridded" lidar mean profile. The collocation procedure is summarized in Fig. 5, where example ship coordinates are depicted as coloured dots, corresponding to the colour of the grid box used for deriving the mean profile.

It should be noted that grid boxes with less than 24 hours of lidar data available (equivalent to 144 10-minute samples) are excluded from the comparison. Figure 6 illustrates the count of 10-minute lidar samples considered within each ASCAT
grid box along the ship route. As observed, grid boxes corresponding to harbour locations exhibit the highest count of lidar retrievals, as the ship tends to remain stationary for longer periods in these areas. Conversely, grid boxes along the rest of the vessel route exhibit varying counts of data, ranging from 144 samples to around 400 in most of the grid boxes. Furthermore, Figs. 5 and 6 show that the surface of certain ASCAT grid boxes, particularly those at or near the two harbours, is partially covered by land. This situation may lead to coastal contamination and excessively high wind speed retrievals within these grid
boxes. The influence of this effect is discussed in the results section of this study, where its potential impact on the presented findings becomes apparent.

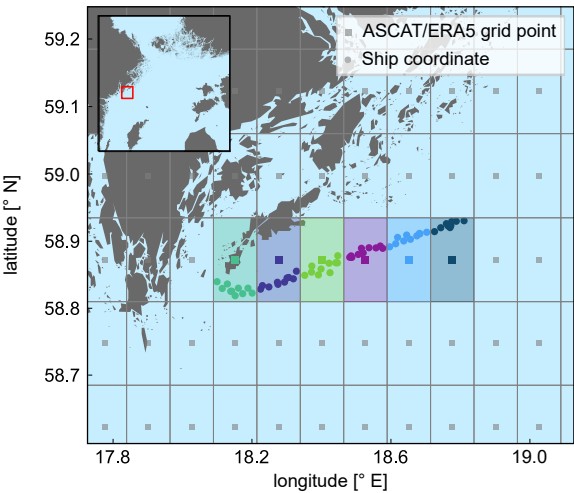

**Figure 5.** Collocation procedure sketch illustrating the comparison of lidar-measured wind profiles against ERA5 and ASCAT profiles. The grey grid represents the ASCAT and ERA5 grid boxes after the coordinate transformation of ASCAT and the ERA5 re-gridding procedure. For each coloured grid box, all lidar measurements performed within that area (depicted as dots of the corresponding colour) are averaged to calculate the corresponding lidar profile.

## 3 Results

The main results of this study are presented in this section. First, Subsection 3.1 compares the extrapolated ASCAT values obtained through the two collocation approaches employed to derive the mean stability correction factor. This analysis high-
315 lights the effect of ASCAT overpasses´ discrete temporal resolution and ERA5 coarse horizontal resolution in the derived mean stability distribution and, consequently, on the extrapolated wind speed at 100 m. Then, a comparative analysis between ERA5 and ASCAT at 10 m and 100 m heights is conducted within the Northern Baltic Sea region. Through this comparison, we evaluate the different characterization of wind speeds represented by the two datasets over the whole area, with a particular emphasis on factors such as coastal contamination and the effect of the employed extrapolation methodology on the discrepancy
between the datasets. Finally, Subsection 3.3 focuses on validating the extrapolated ASCAT and ERA5 wind speed profiles by comparing them against the reference measurements from the ship-based lidar.

In order to validate the wind profiles derived from ASCAT and ERA5 against the lidar in different locations, these comparisons are performed at six locations indicated in Fig. 7. The selection of these locations aims to represent the different wind conditions along the route, including locations near the shore as well as far offshore sites.

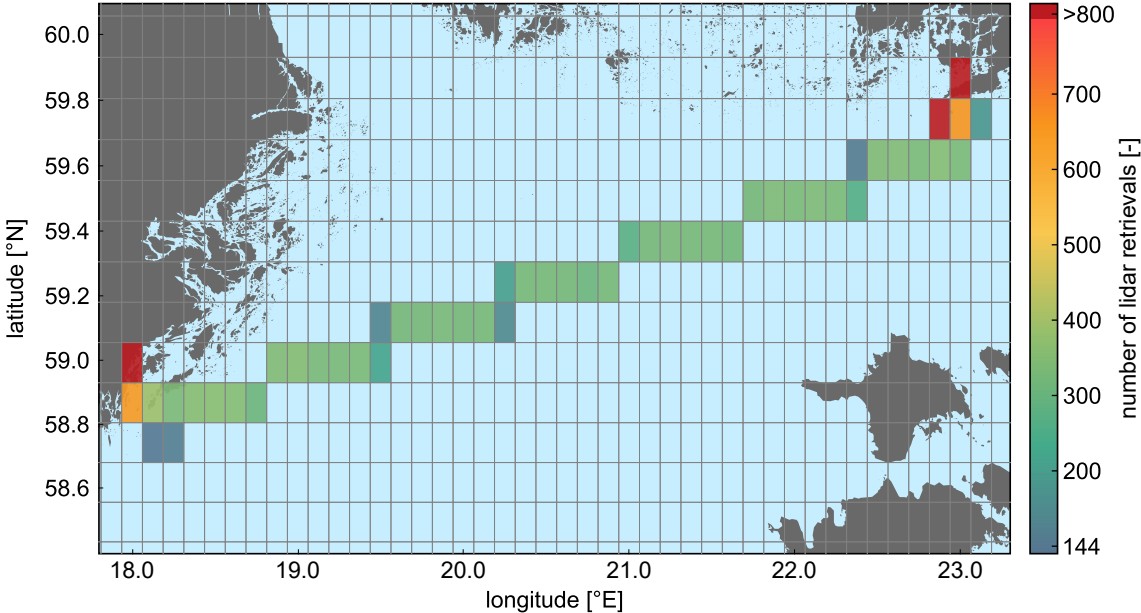

**Figure 6.** Number of 10-minute lidar samples recorded at each ASCAT grid cell. Only grid cells with more than 24 hours of lidar data (or 144 10-minute samples) are coloured, while cells with fewer data are excluded.

## 3.1 Influence of stability information in ASCAT profiles

As explained in Section 2.4, two different collocation approaches have been considered for the characterization of the stability from ERA5 parameters and the corresponding derivation of the mean stability correction. This section investigates the effects of both approaches in the obtained ASCAT wind profiles.

Figure 8 illustrates the differences in wind speed 100 m height between the collocated the full campaign approaches. Overall, the wind speed discrepancy remains minimal across the majority of the study area. In the open sea, the agreement is particularly strong, with differences typically below 0.15 m s$^{-1}$. However, more significant discrepancies are reported in areas near the shore, where wind speed biases reach up to approximately 0.4 m s$^{-1}$. Notably, the region surrounding the Swedish harbour of Nynäshamn exhibits the largest difference in wind speed.

The lower values associated with the collocated approach can be attributed to three primary factors. First, coastal contamination in nearshore areas results in the exclusion of some ASCAT overpasses for data quality reasons, thereby reducing the number of ASCAT observations in these regions. Consequently, the collocated approach in these areas may have insufficient stability information available, potentially introducing a biased representation of the theoretical stability distribution during the campaign period. Secondly, as mentioned in Section 2.4, the same values of the semi-empirical constant $C_\pm$ are assumed for the entire region, instead of using a site-specific definition of these constants. Therefore, the suitability of the selected values may not be optimal for certain locations, leading to an anomalous theoretical representation of the empirical atmospheric distribution.

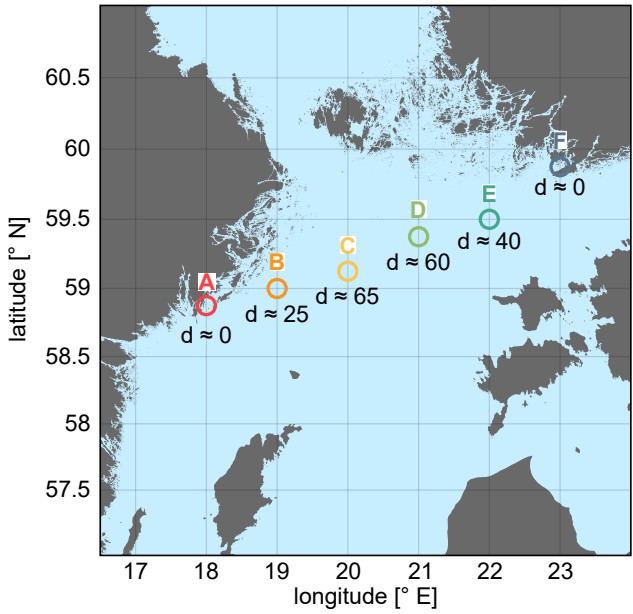

**Figure 7.** Six locations used for the comparison of the datasets. The approximate distance to the nearest shore is indicated, in km, below of each site. Location A corresponds the harbour of Nynäshamn (Sweden) and location D corresponds to Hanko harbour (Finland).

Thirdly, the temporal discretization of ASCAT overpasses, occurring at roughly the same time each day, influences the resulting mean stability distribution "seen" by the collocated approach. This is illustrated in Fig. 9, which depicts the daily cycle of the mean stability $(1/L)$ at the six locations A-F presented in Fig. 7. As can be observed, the collocated approach yields a more variable and unstable mean distribution of the stability conditions near the Nynäshamn harbour (red line), leading to a larger stability correction factor in absolute terms (despite its negative sign at this location), and consequently, to lower wind speeds compared to the full campaign approach, as derived using the equations described in Section 2.4. This instability at location A is attributed to the coarse resolution of ERA5, resulting in land contamination of the grid box at the harbour location, where the land mask covers 56% of the grid box surface. Therefore, the daily stability profile is more akin to that of an onshore site. From 5:00 to 8:00 UTC, the transition from night-time to daytime triggers a decrease in the 1/L value as the surface warms with the sunrise, fostering increased turbulence and vertical mixing in the atmosphere. The period of lowest stability then occurs around midday when the surface heating is more intense. Throughout the afternoon and evening, as surface heating decreases, so does the turbulence, developing a more stable boundary layer. As this trend persists, stability reaches its maximum in the late evening and stays relatively constant until the following morning.

In contrast, locations B to E are purely offshore (with land mask of 0%) and therefore exhibit a more stable diurnal cycle of stability and lower variations throughout the day, due to the presence of a relatively uniform water surface.This leads to

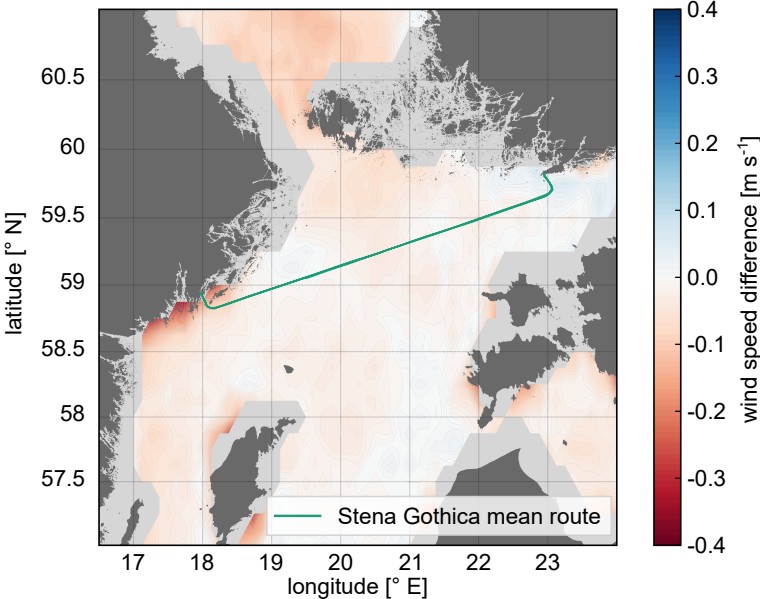

**Figure 8.** Mean wind speed difference at 100 m height, calculated as the collocated approach minus the full campaign approach.

smaller temperature variations and a more stable boundary layer. Finally, location F at Hanko harbour, with a land mask of 6%, presents slightly higher variations in stability during the day compared to the offshore sites but is still relatively steady compared to location A. Consequently, smaller differences are reported between the collocated and full campaign approaches.

Both strategies for calculating the stability correction factor and the corresponding wind profiles demonstrate a high level of agreement, except for some nearshore locations. This, together with the revealed representativeness of the theoretically derived stability distributions observed in Fig. 4, highlights the robustness of the mean stability correction approach in characterizing the atmospheric stability conditions, independently of the approach used, during the period covered by the measurement campaign, and particularly, across open sea regions. Given the minimal differences in the wind speeds at 100 m depicted in Fig.

8, and thus the similar wind profiles obtained using both approaches, subsequent sections of this paper will only consider the full campaign approach for the sake of clarity and conciseness. This approach is expected to provide more representative wind profiles along the complete ship route.

### 3.2  ASCAT-derived vs ERA5 wind speeds

The offshore mean wind speeds based on ASCAT and ERA5 in the Northern Baltic Sea region at 10 m and 100 m heights

are compared in Fig. 10. For an easier comparison, only grid points where ASCAT data is available are included and the same colour scale is used for the four plots.

As can be observed when comparing the spatial variation shown by the two datasets at 10 m, ERA5 exhibits higher mean wind speeds in the areas farthest from the shore at 10 m, with a progressive decrease as the coast is approached. However,

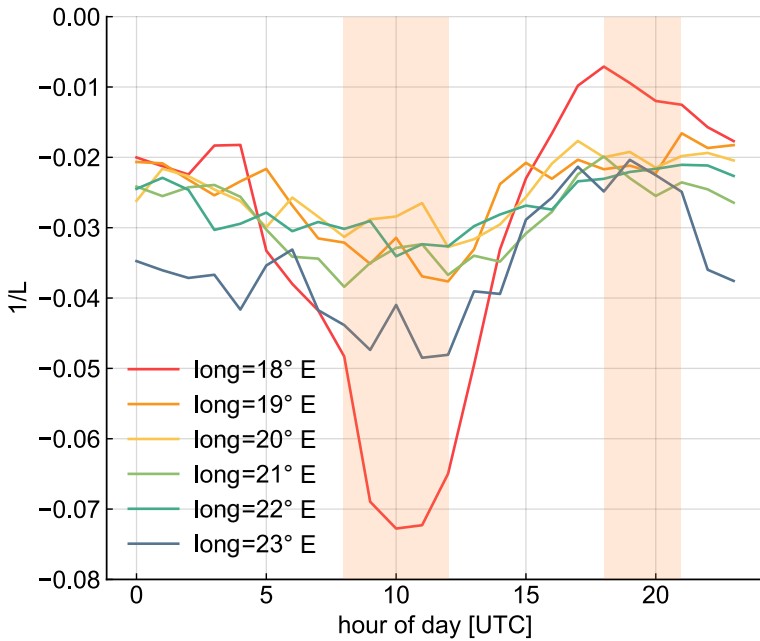

**Figure 9.** Daily cycle of the stability parameter ($1/L$) at the six evaluated locations A-F from Fig. 7. The orange shadows indicate the time periods when ASCAT overpasses are available and therefore, considered for the stability characterization in the collocation approach.

although ASCAT also shows higher wind speeds in the middle of the basin, the areas closest to shore still present considerably
higher values of wind speed compared to ERA5. This discrepancy occurs because, despite the filtering process for the ASCAT dataset, the coastal contamination still affects ASCAT measurements, leading to excessively high mean values in nearshore areas. The effect of coastal contamination in the ASCAT map is particularly visible in the 100 m height map, where the highest mean wind speeds are located along the perimeter of the region with available data.

Both datasets consistently show higher wind speeds at 100 m than at 10 m height. The overall mean wind speeds at 10
380 m are 7.61 m s$^{-1}$ and 7.15 m s$^{-1}$ for ASCAT and ERA5, respectively. However, a notable reduction in the mean deviation ($\overline{U}_{\text{ASCAT}} - \overline{U}_{\text{ERA5}}$) is observed when considering only locations distanced more than 20 km from the shore, where the overall mean deviation decreases to approximately 0.16 m s$^{-1}$. Conversely, locations within 20 km from the shore account for a total mean deviation of 0.98 m s$^{-1}$. Similar findings were reported in Duncan et al. (2019a) in their comparison of ASCAT and ERA5 wind speeds at 10 m over the North Sea and the Dutch coast, where a nearly zero deviation in far-offshore locations and
385 approximately 0.6 m s$^{-1}$ in coastal regions were reported.

At 100 m, the mean wind speed values increase to 9.31 m s$^{-1}$ for ASCAT and 8.67 m s$^{-1}$ for ERA5 if the whole area is considered, though the deviation is reduced to 0.43 m s$^{-1}$ when only far-from-shore sites are considered. The differing biases between these two datasets at the two heights levels (10 m and 100 m) can be attributed to three key factors: first, the inherent difference between the datasets at 10m, second, the mean stability correction approach used to extrapolate ASCAT; and finally,
as illustrated in Figure 8, the impact of the collocation strategy applied for the theoretical stability characterization.

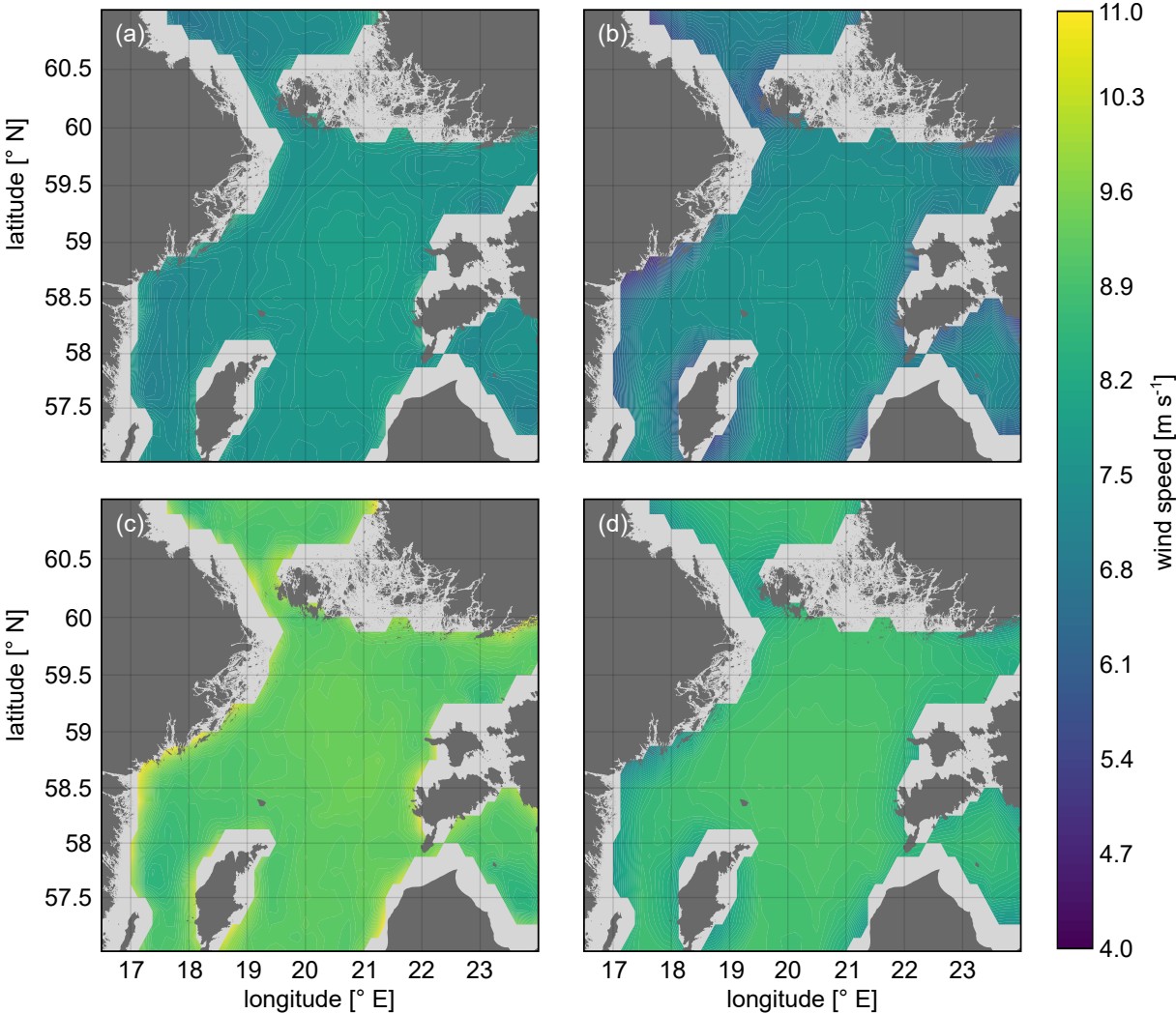

**Figure 10.** Mean wind speed for the campaign period at 10 m (upper panels) and 100 m (bottom panels) for ASCAT (left panels) and ERA5 (right panels).

Figure 11a illustrates the difference in wind speed between ASCAT and ERA5 at 10 m and 100 m, plotted as a function of the distance from the shore (calculated from the centre of each grid box). Additionally, the probability distribution of the wind speed differences for the two datasets at the aforementioned heights is presented in Fig. 11b. As can be observed, there is a clear correlation between the distance from shore and the agreement between ASCAT and ERA5 at both heights. Generally, ASCAT presents higher wind speeds than ERA5 in most grid points, with larger differences closer to the coast, which are more pronounced at the 100 m than at the 10 m level. This discrepancy in nearshore areas can be explained by the combination of excessively high wind speeds retrieved by ASCAT due to coastal contamination and ERA5's inability to properly resolve the coastal atmospheric phenomena and small-scale wind flow variations due to its coarse horizontal resolution. When moving

further offshore (more than around 40 km), this discrepancy stabilizes, converging to more consistent estimates away from the influence of land and coastal effects and reaching mean difference values of around 0.2 m s$^{-1}$ and 0.4 m s$^{-1}$ at 10 m and 100 m height, respectively.

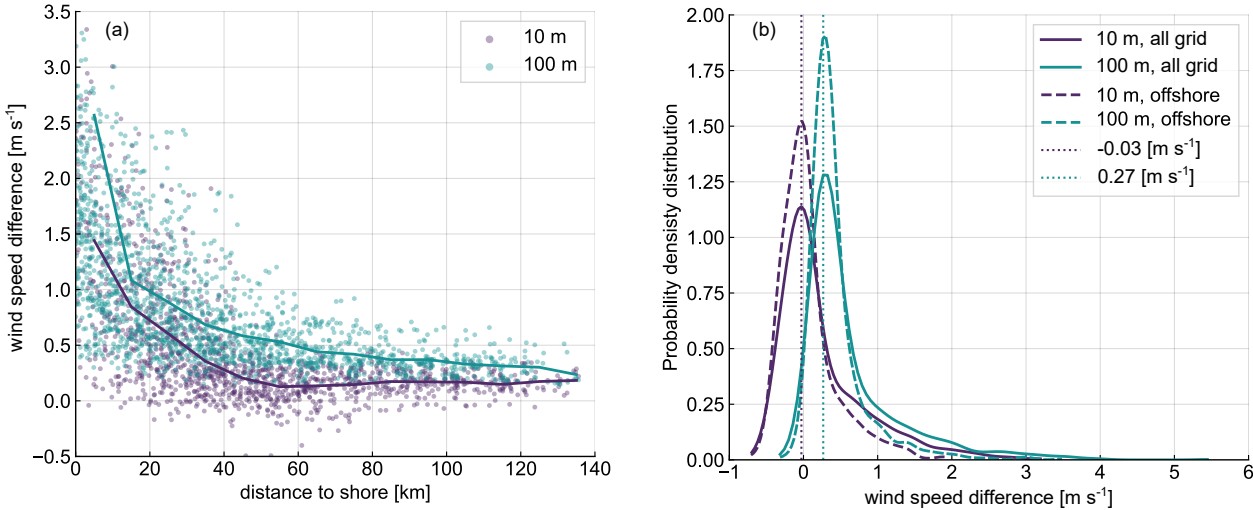

**Figure 11.** (**a**) Wind speed difference at 10 and 100 m for ASCAT minus ERA5 at 10 and 100 m as a function of the distance to the shore. (**b**) Probability density distribution of the wind speed difference at 10 and 100 m for ASCAT minus ERA5. Solid lines for the whole grid and dashed lines for grid points more than 20 km away from the shore. The dotted lines mark the maximum for each of the distribution.

As observed in Fig. 11b, the 10 m height error density distribution is approximately centred around zero bias, whereas the distribution at 100 m is slightly positively biased, highlighting the consistent larger values of extrapolated ASCAT wind speeds at this height compared to ERA5. Nonetheless, the majority of grid points exhibit wind speed differences below ±1 m s$^{-1}$. The probability density function of grid points located more than 20 km away from the shore presents a more pronounced peak near the maximum of the corresponding distribution and appears more squeezed, indicating that wind speed differences exceeding approximately 1.5 m s$^{-1}$ primarily correspond to nearshore grid points affected by coastal contamination effects. A similar error distribution was observed in Hasager et al. (2020), when comparing ASCAT and the Weather Research and Forecast (WRF) model over the European seas.

## 3.3   Comparison against ship-based lidar measurements

The overall mean profiles obtained for each of the employed datasets and averaged along the entire ship route are presented in Fig. 12a. Additionally, the mean wind profiles are shown for each of the six locations A-F defined in Fig. 7. The non-stability corrected logarithmic profiles are included for comparison (i.e. term $\Psi_m^*$ from Eq. (4) set to zero).

As observed, the accuracy of the overall mean profiles depends on the height and dataset considered. Compared to the lidar data, ERA5 consistently underestimates the wind speed by approximately 0.5 m s$^{-1}$ throughout the entire profile, which aligns with the findings of previous studies (Kalverla, 2019; Knoop et al., 2020; Rubio et al., 2022). Conversely, ASCAT's overall

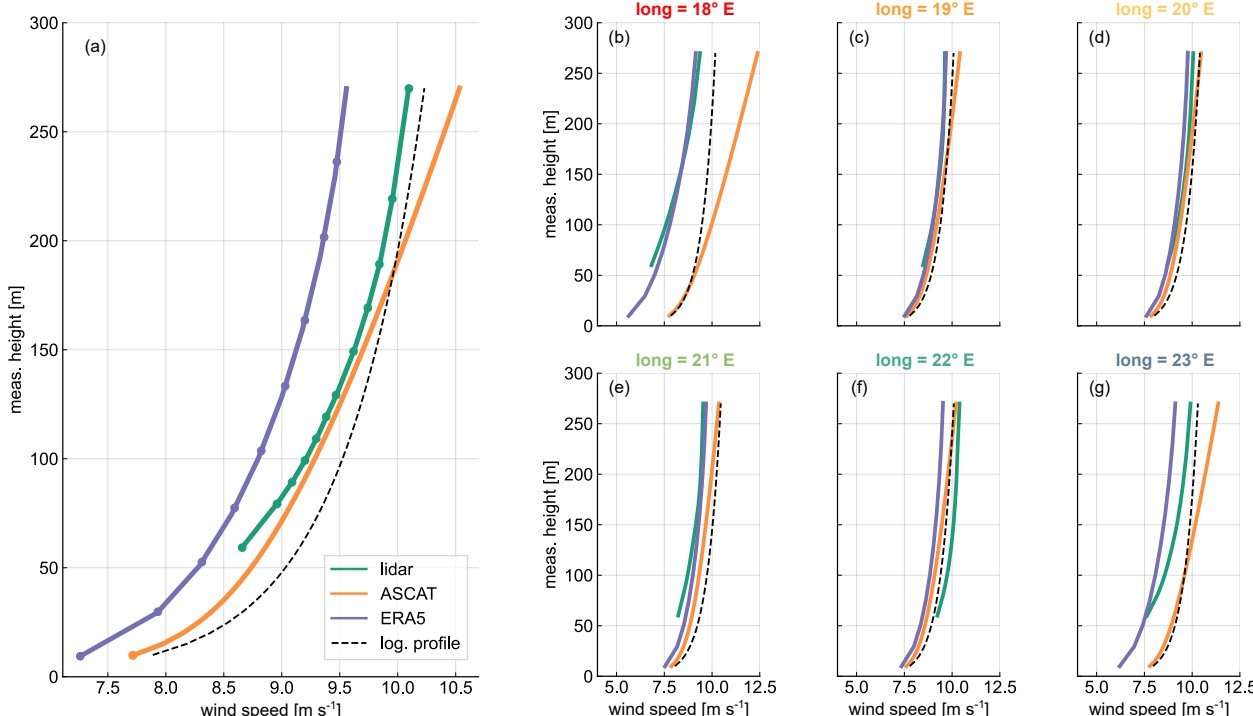

**Figure 12.** (**a**) Mean profiles for the three datasets averaged along the whole ship route. The vertical levels with available data/measurements are indicated with circular markers for each dataset. (**b** - **g**) Mean profiles for the three datasets at the six evaluated positions. In every panel, the logarithmic profile (non-stability corrected) is indicated by the black dashed line.

mean profile bias is consistently positive (indicating ASCAT overestimation regarding lidar), with the magnitude depending on the considered height. Both ERA5 and lidar profiles exhibit a similar shear within the height range covered by the lidar measurement, ranging from 8.4 m s$^{-1}$ to 9.6 m s$^{-1}$ for ERA5 and from 8.7 m s$^{-1}$ to 10.0 m s$^{-1}$ in the case of the lidar. In contrast,

the ASCAT profile struggles to characterize the shear outside the surface layer, with wind speeds ranging from 8.9 m s$^{-1}$ at 60 m height to 10.5 m s$^{-1}$ at 270 m. The ASCAT bias becomes increasingly pronounced above 200 m height; beyond this threshold, the logarithmic profile outperforms the stability corrected profile. This is because these heights exceed the range of applicability of the extrapolation methodology employed (Kelly and Gryning, 2010).

Although the ASCAT wind profiles, on average, appear to outperform ERA5 in terms of overall accuracy, Figs. 12b-g reveal

that the performance of both datasets strongly depends on the location considered. In the case of the harbour locations, ERA5 significantly outperforms ASCAT profiles, which exhibit excessively high wind values even at 10 m height, highlighting the influence of coastal contamination at these sites. Additionally, it is striking to observe the substantial deviation of the ASCAT stability corrected profiles from the logarithmic profiles, particularly at heights above 50-100 m, as a consequence of a stability distribution that is not representative enough of these specific sites. For the remaining locations, both datasets demonstrate a

comparable agreement with the lidar wind profiles.

A statistical analysis of the wind speed deviation between ASCAT and ERA5 with regard to the lidar observations ($\Delta U_{\mathrm{ASCAT}} = U_{\mathrm{ASCAT}} - U_{\mathrm{lidar}}$ and $\Delta U_{\mathrm{ERA5}} = U_{\mathrm{ERA5}} - U_{\mathrm{lidar}}$) is presented in Fig. 13 in the form of a box plot. Each box plot is calculated considering the wind speed difference of all the grid boxes with lidar data along the whole route of the ship, but grid boxes closer than 20 km away from the shore have been excluded to minimize the effect of ASCAT coastal contamination in the derived statistics. The black line corresponds to the mean, the coloured box marks the 25th and 75th percentiles, and the whiskers indicate the data extremes calculated as 1.5 times the interquartile range. Outliers outside the whiskers are hidden to maintain clarity and readability. The continuous lines represent the root mean square error (RMSE) of the wind speed difference between the gridded dataset and the lidar. To mitigate the potential effects of coastal contamination in the results presented, only grid points more than 20 km away from shore were included in the calculation of the box plot and the RMSE.

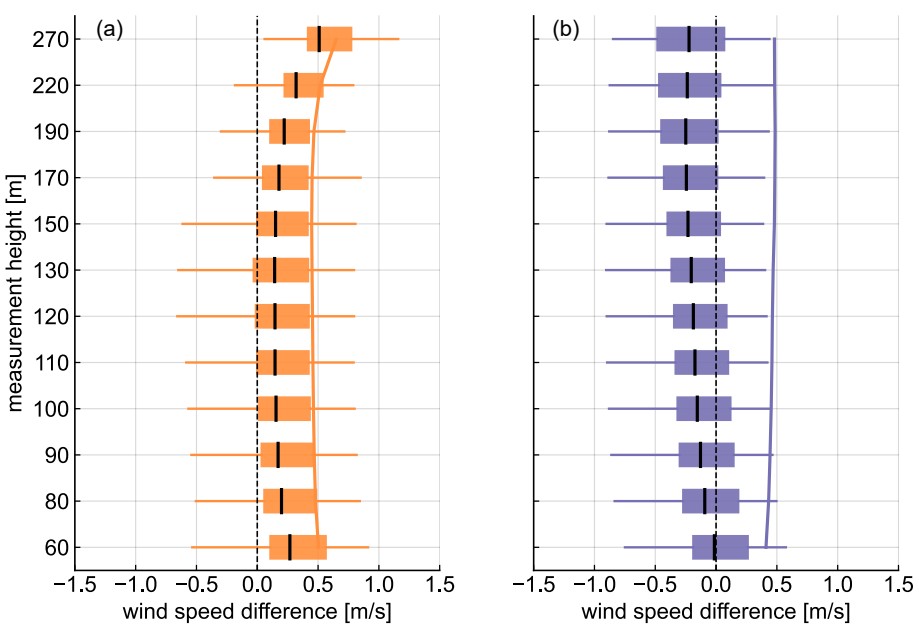

**Figure 13.** Box plots of the wind speed difference from ASCAT (**a**) and ERA5 (**b**) minus the lidar. The coloured boxes extend from the first to the third quartiles of the data and the means are indicated by black lines. The whiskers extend to the data extremes, defined as a distance of 1.5 times the interquartile range (IQR) above and below the upper and lower quartiles, respectively. The solid lines indicate the RMSE between the gridded datasets and the lidar. Only grid points more than 20 km away from shore were included.

Both datasets show a comparable absolute mean in the central part of the profile, with values of around $\pm 0.2$ m s$^{-1}$ in the height range between 90 m and 170 m height. This indicates that both ERA5 and ASCAT yield similar performance in this segment of the profile, suggesting that they are both reasonably aligned with the lidar observations in the lower to mid-altitude ranges. However, a notable difference arises when examining the overall biases. ERA5 consistently underestimates the wind speed across the entire profile, with this negative bias becoming increasingly pronounced with altitude and reaching the largest negative mean bias of around 0.2 m s$^{-1}$ at 270 m. Contrarily, ASCAT profiles exhibit a persistent overestimation of wind speed relative to the lidar across all heights. This overestimation increases significantly above 170 m. When considering variability,

as represented by the interquartile range (IQR), both datasets reveal relatively analogous patterns. For ERA5, the IQR remains fairly constant across all heights, with values around 0.5 m s$^{-1}$, suggesting a stable performance across different elevations. In the case of ASCAT, IQR displays a slight decrease with height, highlighting the larger and more consistent overestimation at higher altitudes.

The whiskers analysis provides further insights into the discrepancies between the two datasets. For ERA5, the lower whiskers extend further into negative values as altitude increases, with the larger underestimations reaching approximately -1.3 m s$^{-1}$ at 270 m. Differently, ASCAT's whiskers reveal a different pattern, particularly noteworthy are the upper (positive) whiskers that extend significantly beyond the lower whiskers at altitudes above 170 m. This observation strikes emphasises again the pronounced tendency for ASCAT to overestimate wind speeds at higher elevations.

The RMSE analysis corroborates the findings from the median and whisker assessments, revealing similar results for both datasets, with RMSE values around 0.5 m s$^{-1}$ along the profile. Nevertheless, while the RMSE remains nearly constant across the entire profile for ERA5, ASCAT's RMSE demonstrates the deteriorating performance of the employed extrapolation methodology in the upper part of the profile.

In order to evaluate the accuracy of ASCAT and ERA5 wind profiles across the different areas covered by the ship route, Fig. 14 illustrates the wind speed differences between these datasets and the lidar profiles for all the grid boxes along the ship track. As can be observed, both datasets show a better performance in regions located further away from the shore, which is evident from the concentration of outliers (points falling outside the confidence intervals) in these areas. This observation holds true for all three presented elevation levels. Notably, the western area of the ship route (longitude below 18.5 degrees) exhibits the largest errors for both ASCAT-extrapolated and ERA5 winds, with maximum differences exceeding 1 m s$^{-1}$ at all elevation levels. This indicates that wind speed estimation cannot be done accurately enough in these areas using these datasets, first, because of the poor quality of ASCAT in areas closer to the coast, and secondly, due to the insufficient ERA5 grid box sizing, which is unable to capture the small-scale wind flow variations in these complex locations and the intricate interactions in the coastal boundary layer influenced by both land and sea.

The mean differences vary depending on the dataset and elevation considered, highlighting the different shear exhibited by each of the datasets and their different representations of the wind profiles. ERA5 shows a smaller mean difference of -0.25 m s$^{-1}$ at 60 m, while reaching a maximum value of -0.5 m s$^{-1}$ at 220 m. In the case of ASCAT, the smallest mean difference happens at the intermediate height level, whereas the highest difference can be found also at 220 m height.

It can be noted that, although ERA5 usually underestimates the wind speed, this is more pronounced at higher elevations and in the western part of the ship track. In contrast, ASCAT mainly overestimates compared to the lidar measurements.

A final quantification of the accuracy of the gridded datasets compared to the lidar measurements is presented in Fig. 15. Here, the normalized root mean squared error (nRMSE) across all lidar measurement heights is calculated for each compared grid box. The calculation of the nRMSE is expressed in the equations below for ASCAT and ERA5:

$$nRMSE_{\text{ASCAT}} = \frac{\sqrt{\frac{1}{n}\sum_{i=1}^{n}(U_{\text{ASCAT,i}} - U_{\text{lidar,i}})^2}}{\bar{U}_{\text{lidar}}} \qquad (7)$$

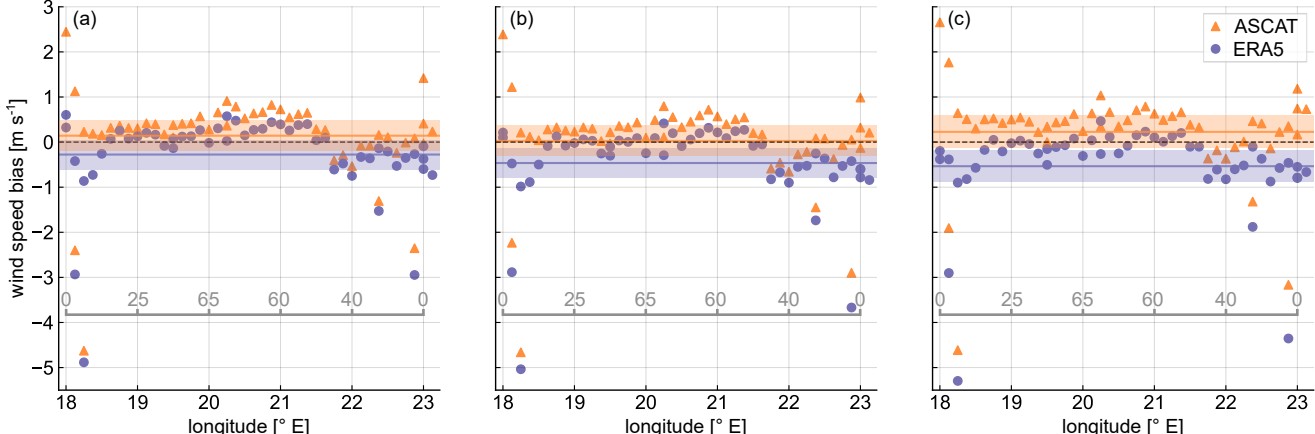

**Figure 14.** Wind speed bias ($\Delta U_{\text{ASCAT}} = U_{\text{ASCAT}} - U_{\text{lidar}}$ and $\Delta U_{\text{ERA5}} = U_{\text{ERA5}} - U_{\text{lidar}}$) along the different grid boxes depending on their longitude coordinate at 60 m (**a**), 150 m (**b**), and 220 m (**c**) height. The mean biases along the whole ship route are represented by solid lines and the 95 % confidence interval is indicated by the shadowed areas. The approximate distance to the shore, in kilometres, is indicated by the labels of the grey line.

$$480 \quad nRMSE_{\text{ERA5}} = \frac{\sqrt{\frac{1}{n}\sum_{i=1}^{n}(U_{\text{ERA5,i}} - U_{\text{lidar,i}})^2}}{\bar{U}_{\text{lidar}}} \tag{8}$$

where $n$ represents the 12 measurement levels of the lidar, $U$ corresponds to the wind speed at the $i$-th height for each dataset, and $\bar{U}_{\text{lidar}}$ is the mean lidar speed averaged across the entire profile.

As can be observed, both datasets present good agreement in the area of the basin and higher errors in the nearshore longitudes. When comparing the two datasets, ERA5 shows a smaller nRMSE in the majority of the studied region, except in the

485 eastern area near the harbour in Hanko. This may be attributed to the differing spatial resolutions of the two datasets. In the east of 22 degrees longitude, the finer resolution of ASCAT mitigates the impact of coastal contamination, enabling it to capture local conditions more effectively and consequently leading to a lower average nRMSE in this region. In contrast, the coarser resolution of ERA5 may be insufficient to adequately represent the average wind characteristics in this area. Conversely, in the western part of the studied area, with features more intricate topography and a higher density of small islets within a few tens

of kilometres from the mainland shoreline, ASCAT measurements are more susceptible to coastal contaminated. This results in excessively high wind measurements at 10 m, thereby contributing to larger nRMSE values across the whole profile. When comparing the biases between ASCAT and ERA5 against the lidar, ASCAT shows a smaller average absolute bias than ERA5 across the entire region at all three heights considered (see Fig. 14). However, as illustrated in Fig. 15, most locations exhibit a lower nRMSE for ERA5 than for ASCAT, indicating a higher precision of ERA5, which consistently underestimates the wind

profiles. In contrast, ASCAT's errors exhibit a higher variability; while most grid points overestimate the profiles, a few present

a pronounced underestimation. Furthermore, as seen in Fig. 13, including higher heights in the consideration for calculating the nRMSE heavily penalizes the performance of the satellite profiles.

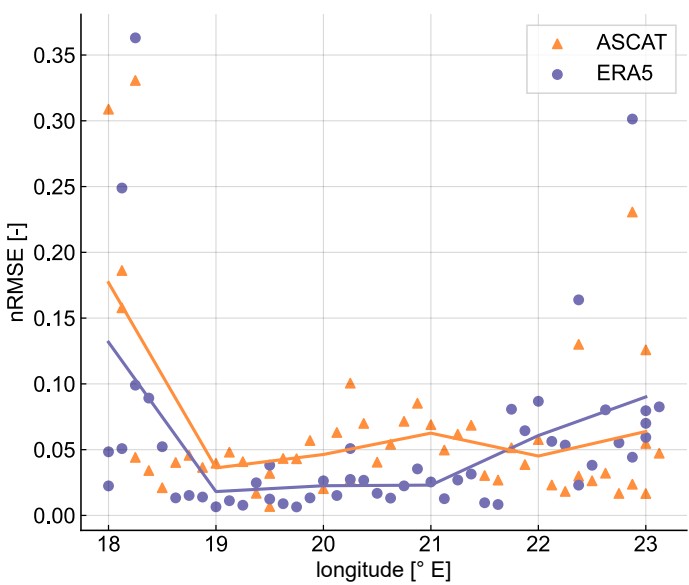

**Figure 15.** Normalized root mean squared error calculated along the whole profiles for each of the grid boxes. The solid lines represent the binned mean nRMSE calculated using longitude bins of 1°.

## 4  Discussion

The objective of this study has been to assess the accuracy of ASCAT-derived wind speed profiles for characterizing offshore winds at turbine operating heights in the Northern Baltic Sea. Initially, ASCAT winds were compared against the ERA5 reanalysis dataset, which is frequently used as a fallback for offshore wind characterization in the absence of in situ measurements. Subsequently, both gridded datasets were evaluated against reference in situ observations obtained from a novel ship-based lidar measurement campaign.

To extrapolate ASCAT wind data, the mean stability correction methodology formulated by Kelly and Gryning (2010) was employed. This method uses the mean ASCAT wind measurements at 10 m altitude during the campaign, along with atmospheric stability information derived from ERA5. It is worth noting that previous studies (Optis et al., 2021) have suggested that machine learning-based techniques for extrapolating satellite winds could outperform the mean stability correction method used in this study. However, the limited amount of data available over the campaign period hinders the implementation of such data-driven approaches.

One of the primary limitations of the mean extrapolation technique is the requisite for a comprehensive characterization of the atmospheric stability throughout the comparison period. To address this, we examined the impact of the stability information

available on ASCAT profile derivation by comparing two distinct strategies: the collocated and the full campaign approach. Both strategies demonstrated remarkable agreement across most of the examined area, resulting in very similar wind speed extrapolated at 100 m, regardless of the collocation strategy used. However, notable discrepancies were reported in coastal areas, where the collocated approach exhibited lower wind speed values, with maximum biases of up to 0.4 m s$^{-1}$ in these regions. This divergence may be due to several factors, including the greater number of ASCAT overpasses filtered out in coastal areas because of coastal contamination, and the subsequent reduction in available stability data for the collocated approach that can lead to different stability correction factors derived from the two approaches. Additionally, the temporal discretization of ASCAT overpasses also leads to variations in the predominant stability conditions observed by the two approaches. Finally, the use of generic empirical constants $C_{\pm}$, kept uniform across the entire area, may result in inaccurate theoretical distributions in some areas. Different studies have adopted varying values of these constants based on the observed stability conditions (Kelly and Gryning, 2010; Badger et al., 2016; Optis et al., 2021). Therefore, further research is necessary to establish a reliable and standardized methodology for determining optimal values of these constants according to site-specific stability conditions.

The comparison between ASCAT and ERA5 winds reveals an overall good agreement when assessing mean wind speeds across the open ocean area of the Baltic Sea. However, ASCAT consistently reports slightly higher wind speeds than ERA5, with a mean bias of approximately 0.45 m s$^{-1}$ at 10 m and 0.64 m s$^{-1}$ at 100 m. Notably, greater discrepancies are observed at near-shore grid points, where wind speed differences often exceed 1 m s$^{-1}$. When analysing mean wind speed deviations as a function of distance from the coastline, a clear reduction trend in bias emerges as this distance increases. Specifically, more consistent and lower biases are observed in grid cells beyond 40 km from the coast, with biases stabilizing at approximately 0.2 m s$^{-1}$ and 0.4 m s$^{-1}$ at 10 m and 100 m, respectively. These larger nearshore discrepancies can be attributed to the inherent limitations of both datasets. For satellite-based measurements, areas near the shorelines are susceptible to coastal contamination, especially in grid boxes partially covered by land, resulting in anomalously high wind field measurements due to factors such as wave breaking and surface slicks (Johannessen, 2005; Kudryavtsev, 2005). Meanwhile, ERA5's inaccuracy in near-shore regions stems from its relatively coarse horizontal resolution, limiting its ability to simulate coastal atmospheric dynamics and small-scale wind flow variations, particularly in areas with abundant small islands and rocky islets, which are especially common in the coastal regions analysed in this study (Dörenkämper et al., 2015; Gualtieri, 2021). The application of high-resolution satellite technologies, such as synthetic aperture radar (SAR), could enhance the resolution of coastal wind speed gradients thanks to their finer grid spacing (de Montera et al., 2022). However, in this study, SAR measurements were not contemplated due to their lower temporal resolution compared to ASCAT and the relatively short duration of the campaign. This decision was made to maximize the amount of collocated data and ensure consistency in the statistical metrics evaluated.

The discrepancies of the two datasets in the coastal regions are further emphasized when compared against the ship-based lidar measurements. Both datasets show the highest biases in the longitudes corresponding to the harbour locations, this is, in longitudes further west from 18.5° E and further east from 22.5° E. Excluding these nearshore locations, the mean nRMSE along the entire profiles is reduced from 0.07 to 0.05 for ASCAT, and from 0.06 to 0.03 for ERA5. Analogous observations were documented by Takeyama et al. (2019), in which a comparison of ASCAT data and Weather Research and Forecasting

(WRF) simulations against in situ measurements in the vicinity of the Japanese coast revealed significantly reduced errors beyond 25 km from the shore.

When comparing the overall mean wind profiles from the three datasets, ASCAT exhibited a closer similarity to the lidar wind profile than ERA5, particularly excelling at altitudes ranging from 100 to 150 m. However, a closer analysis of individual locations along the ship route (A-F) reveals that this apparent superior performance of ASCAT is a result of averaging profiles from nearshore and offshore sites. While the overall agreement between the three datasets is strong, with mean deviations from the lidar profile remaining below 0.6 m s$^{-1}$ for both datasets at the four offshore sites, ASCAT profiles at nearshore locations A and F show significant overestimations relative to lidar measurements, with mean differences of 2.5 m s$^{-1}$ and 1.1 m s$^{-1}$ at A and F, respectively. In contrast, ERA5 manifests superior performance in capturing wind shear across the profile, exhibiting a more consistent bias relative to lidar measurements. This results aligns with previous similar studies in the Southern Baltic Sea (Rubio et al., 2022). In contrast, results highlight distinct bias patterns between ERA5 and ASCAT. ERA5 consistently underestimates wind speed throughout the entire profile, with the negative bias increasing with altitude, peaking at -0.2 m s$^{-1}$ at 270 m. Conversely, ASCAT persistently overestimates wind speed relative to the lidar at all altitudes, with this overestimation rapidly increasing above 170 m and peaking at around 0.5 m s$^{-1}$ at 270 m, as evidenced in Fig.13. It is important to note that, unlike the MOST stability correction approach, the mean stability correction approach used here can be applied above the surface layer. The findings of this study indicate a good performance within the lower 170 m of the atmosphere. However, the applicability of this method depends on the specific atmospheric stability conditions at the location of interest and the duration of the comparison period.

One distinct aspect of the ship-based lidar campaign conducted onboard a ferry ship is the near-constant correlation between the ship's position and the time of day. Therefore, and similarly to the discretized temporal resolution of ASCAT observations, the derivation of a complete diurnal wind speed cycle from these measurements at the specific areas covered by the vessel route is not feasible. Consequently, the mean values derived from lidar measurements may exhibit biases that vary depending on the time slots during which measurements were acquired at particular locations. Additionally, it is acknowledged that lidar measurements, like any other observational data, are subject to inherent uncertainties that may impact the results (Duncan et al., 2019b; Rubio and Gottschall, 2022). Nevertheless, the observed deviations between the lidar measurements and both extrapolated ASCAT and ERA5 significantly exceed the magnitude of potential discrepancies attributable to floating lidar uncertainties, which can be up to approximately 2 % with mast-mounted anemometers as lower limit reference (Wolken-Möhlmann et al., 2022).

Finally, it is imperative to highlight that although the disparities in wind speeds between ASCAT and ERA5 relative to lidar are generally small in far-offshore regions, their cumulative impact over a large-scale wind energy project can still have relevant implications for energy production estimates and financial assessments. Therefore, continued efforts to refine both satellite-based measurements and numerical models are essential to enhance the accuracy of wind resource assessments for offshore wind energy applications. The diverse characteristics and insights into wind patterns derived from satellite-derived observations, numerical models, and ship-based lidar measurements suggest that an integrative approach, harnessing the collective strengths of these datasets, could yield substantial gains in the accuracy and reliability of offshore wind statistics derivation.

To this end, several studies have made inroads by generating wind atlases in other regions through the combination of these datasets (Doubrawa et al., 2015). However, the assimilation of non-stationary measurements and the incorporation of more sophisticated extrapolation methodologies, such as mean stability correction, could bring further benefits.

## 5   Conclusions

Satellite-borne scatterometers and numerical models are two additional sources of wind information for characterizing off-shore winds. This study undertakes an intercomparison of these datasets and validates them against reference ship-based lidar measurements. In this study, ASCAT satellite observations are extrapolated to turbine operation heights using a statistical adaptation of MOST, which incorporates ERA5 stability information to assess a mean profile stability correction factor.

The two proposed collocation approaches for ASCAT extrapolation show strong agreement across the open ocean area of
the Baltic Sea. However, the collocated approach generally yields slightly lower mean wind speeds, particularly in nearshore areas surrounding the Nynäshamn harbour. This discrepancy is primarily attributed to the temporal discretization of ASCAT overpasses, the limited amount of stability information in the collocated approach due to less data available in this region, and the non-site-specific definition of the $C_{\pm}$ constants, which leads to different predominant stability conditions captured by each approach.

The agreement between ERA5 and ASCAT at 10 m in far-from-shore regions shows a mean deviation of lower than 0.2 m s$^{-1}$. However, coastal contamination in ASCAT measurements and the coarse resolution of ERA5 lead to a mean deviation between these datasets of 0.98 m s$^{-1}$ in areas within 20 km from the shore. The ASCAT extrapolation results in an overall larger deviation at 100 m, with a mean bias between ASCAT and ERA5 in far-from-shore regions (more than 40 km away from shore) of approximately 0.4 m s$^{-1}$.

The comparison of extrapolated ASCAT and ERA5 against lidar wind profiles reveals a systematic underestimation of ERA5 and overestimation of ASCAT profiles by approximately 0.2 m s$^{-1}$ between 90 m and 170 m height. Although both datasets show deteriorating performance with height, this is particularly notable in ASCAT profiles, which present rapidly increasing biases above 170 m, potentially due to limitations in the applicability of the mean stability correction approach above this height, for the specific stability conditions in the area and period of study. Furthermore, the validation of these two datasets
against lidar observations also highlights the challenges when capturing wind in nearshore locations. Both datasets present notably larger biases and nRMSE values in the nearshore western and eastern sides of the area covered by the ship track.

The fundamental differences of this study from previous literature is the comparison of mean ASCAT extrapolated wind profiles against lidar measurements across a broad geographical area, within an increased vertical extension, and through the application of a novel collocating technique. This brings valuable revelations concerning the prospective applicability of
ASCAT observations within varying spatial constraints and their feasibility at higher altitudes, including turbine operational heights. Moreover, the findings highlight certain challenges when intercomparing datasets with different temporal and spatial characteristics. Such differences may culminate in potential biases amongst datasets attributed to, for instance, the temporal windows within which measurements are accessible.

In conclusion, extrapolated ASCAT wind retrievals using the mean stability correction approach may serve as a useful additional asset for portraying offshore winds at turbine operation heights, manifesting accuracy levels comparable to numerical model outputs from ERA5 in far-from-shore regions. This methodology is particularly beneficial in scenarios where more complex extrapolation methods are impractical or when in situ measurements are limited, providing an additional source of wind information and thereby improving the reliability of offshore wind characterization studies. However, the application of both ERA5 and ASCAT must be approached with caution due to their inherent characteristics, including insufficient spatial resolution and the inability to adequately capture wind farm wake effects, which limit their utility for detailed wind farm energy yield assessments. Despite this, these datasets are still valuable for other applications, such as large-scale planning of wind potential or preliminary site screening studies, helping to identify regions with promising resources. In such cases, the findings of this study provide valuable insights into the conditions under which these datasets and methodology can be applied and the level of reliability that can be expected. Nonetheless, it is crucial to also acknowledge the primary limitations of this approach, such as excessive wind speed deviations in nearshore locations and the increased expected error at higher altitudes.

Therefore, further research could explore the suitability of other satellite technologies, such as SAR measurements, with a superior spatial resolution, to mitigate the issues associated with coastal contamination. Additionally, ship-based lidar systems offer reliable wind measurements within vast areas of investigation, underscoring their potential for validating and optimizing not only satellite extrapolation methodologies, but also numerical models datasets, including regions where wind farm wake effects play a significant role.

*Data availability.* Data used for this paper were collected from the following sources. Ship-based lidar measurements were provided by Fraunhofer IWES and they are available upon request. The ERA5 data are freely available via the Copernicus Data Storage (CDS): https://cds.climate.copernicus.eu/cdsapp#!/home. ASCAT measurements were downloaded via the Copernicus Marine Data Service (CMS): https://marine.copernicus.eu/.

*Author contributions.* HR and JG designed and executed the measurement campaign. HR performed the investigation, data processing, analysis and visualization and wrote the manuscript. All authors contributed to the conceptualization and methodology and reviewed the manuscript. JG had a supervisory function.

*Competing interests.* The authors declare that they have no conflict of interest.

*Acknowledgements.* This research received funding from the European Union's Horizon 2020 research and innovation program under the Marie Skłodowska-Curie grant agreement no. 858358 (LIKE – Lidar Knowledge Europe). We would like to express our special thanks to Stena Line for providing us with the opportunity to conduct the campaign onboard the *Stena Gothica* and the Research Institutes of Sweden

RISE for their coordination of the measurement campaign. We would also like to thank the crew of the *Stena Gothica* for their invaluable support during the installation, operation, and dismantling of Fraunhofer IWES's ship-based lidar system.

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
