# Peer review of "Ship-based lidar measurements for validating ASCAT-derived and ERA5 offshore wind profiles"

_Atmospheric Measurement Techniques, 2024_

## Referee Comment (RC2)

**Ship-based lidar measurements for validating ASCAT-derived and ERA5 offshore wind profiles**

Hugo Rubio, Daniel Hatfield, Charlotte Bay Hasager, Martin Kühn, and Julia Gottschall

My main problem with this paper is that your aim is to provide a means for offshore wind resource assessment which is an alternative for ERA5, but

1. You still need ERA5 to modify ASCAT
2. You do not take into account wake and blockage effects. This is probably okay for now (there are no wind farms yet in the Baltic according to https://map.4coffshore.com/offshorewind/), but it will become a problem during the lifespan of the wind farm when more wind farms are built (the effect is already significant on the North Sea). ERA5 does not take these effects into account and ASCAT is too course to measure them (SAR can: WINS50 - Winds of the North Sea in 2050 | Publications *I Wijnant, A Stepek (2023): Fit(ch) for shipping Wind farm wake effects at 10 m height, KNMI WINS50 Report.*). The only way to predict how much power a wind farm will produce during its lifespan is to understand and properly model wind farm effects and then to use this knowledge/ these models for future wind farm scenarios. The old method for wind resource assessments (measure-correlate-predict) no longer works.
3. Mean values of the wind are not relevant if you want to predict power: you need to look at correlation on a 10 min (or hourly) basis, especially for wind speeds between cut-in and rated (power curve).

The concept of doing lidar measurements on a ferry is interesting and probably even more interesting on the North Sea where you no doubt can measure wind farm effects this way. How to compare these measurements to ERA5 is as you describe not straightforward (ship-motion compensation algorithm), comparing them to ASCAT I presume even harder. You can only do that when ASCAT is available which is around 09 and 19 UTC when the ferry is in the harbour or reasonably close to the coast and you say that there the ASCAT-signal is disturbed? Are there other measurements available near the harbours (or in the Baltic Sea) that you could use to compare the lidar measurements to?

I think that comparing ERA5 to 'ASCAT extrapolated to hub height with ERA5' is not very useful (and scientifically sound). What would be interesting is to do triple collocation with (1) ASCAT-winds extrapolated to hub height with your method (or different methods) based on ERA5, (2) ship based lidar measurements and (3) a mesoscale model in hindcast mode, for now without Wind Farm Parametrization (WRF? Unless you can get hold of COSMO-CLM or HARMONIE?). The aim of your paper then would be the best possible extrapolation of ASCAT to hub height. Why that is useful is something you will have to explain (not for wind resource assessments).

Comments more in detail:

- Bias ERA5 at hub height 0.5 m/s is also what is found on the North Sea in Characterisation of offshore winds for energy applications — Research@WUR and at Cabauw in Energies | Free Full-Text | Dutch Offshore Wind Atlas Validation against Cabauw Meteomast Wind Measurements (mdpi.com). NEWA comparable to ERA5 (at least on the North Sea). Undisturbed winds in DOWA (2008-2018) and WINS50 (2019-2021) are much better than ERA5 (including correlation) and the domain

covers most of the Baltic Sea, but hourly data unfortunately not available for 2022 and 2023 when you have the lidar measurements (Home | Dutch Offshore Wind Atlas; WINS50 - Winds of the North Sea in 2050).

- Line 52: Unclear sentence: *Lastly, the trustworthiness of satellite retrievals remains a knowledge gap, due to the lack of available in situ datasets for validation especially in deep water regions.*

- We know that the Dutch part of the North Sea (DEEZ) does not experience a trend in offshore wind speed, only an Interannual Variability (IAV) of 3.5 and 4% for sites in the northern part of the DEEZ and between 4 and 4.5% in the southern part of the DEEZ (Inter-annual wind speed variability on the North Sea | Report | KNMI Projects). Is any information like this available for the Baltic Sea? How representative is 28-6-22 until 21-2-23 for the wind climate in the Baltic Sea? This you can check e.g. with ERA5 data (compare ERA5 28-6-22 - 21-2-23 to ERA January1940-now).

- Line 143: (fig 3) maybe I missed it, but what ASCAT data did you use?

- Line 189: (typo): *Several methodologies to vertical satellite extrapolation…* not to, but for.

- Line 201: *This method involves a long-term correction of atmospheric stability effects, obtained from the numerical model dataset ERA5, along with an adaptation of the MOST to vertically extrapolate the satellite wind measurements.* What is long-term about it? Why the name '**long-term** extrapolation method'?

- Line 203-205 not clear: do you mean that a wind profile can be stable up to a certain height and above unstable and that this 'long-term extrapolation method' can handle this?

- Line 206-217 not clear: what is the difference between the 'instantaneous stability correction' and the 'long-term stability correction'?

- Line 221-268: so basically the stability correction has only 2 values for C per height which are the same for the whole Baltic Sea, one for stable and one for unstable. It does not matter how (un)stable the atmosphere is or whether the grid box is near the coast or further offshore: correct?

- Line 302: (fig 7). You compare the collocated approach (only ERA5 stability information at moments when ASCAT overpasses is considered) to the full campaign approach (all ERA5 stability information from the whole duration of the campaign is used). Both approaches do not include spring which is often the most stable period (cold sea water and warmer air above). Also, mean wind speed is not really relevant for wind resource assessments. So I do not really understand the sentence: '*This highlights the robustness of the employed methodology and indicates that the dataset size allows for an accurate characterization of atmospheric stability conditions during the campaign and along the entire ship track*'.

- Line 307-320: 'pronounced instability in the morning?' Why would ERA5 produce stronger unstable conditions (lower 1/L) in the morning at Nynashamn? What do we know about the water temperature near Nynashamn and how it is modelled by ERA5 (shallower/warmer water between Bedaron and the mainland maybe?)? ERA5 has grid boxes of 31 km$^2$ so model values are probably very land-contaminated in that area: can you show the ERA5 grid boxes near the harbours? What is the prevailing wind direction? Basically ERA5 and ASCAT are not very good in coastal area: maybe you should take them out of your analyses?

- Line 341 (fig 9 10m validation): compare to *Validation of DOWA ('undisturbed wind' = HARMONIE without WFP) with ASCAT (too coarse to measure wind farm effects)*

*at 10 m height:* TNO report - DOWA validation against ASCAT satellite winds | Report | Dutch Offshore Wind Atlas.

- Line 341 (fig 9 100m validation): so we can conclude that ERA5 is internally fairly consistent (profile depends on ERA5 stability parameters)?
- Line 354: *'... highlighting the consistent overestimation of wind speed from ASCAT at this height'.* At 100m this is not ASCAT, but ASCAT extrapolated with ERA5. And we all know that ERA5 is not unbiased at 100m (0.5 m/s underestimation) so you cannot draw this conclusion. See also Line 364/365.
- Line 364/365: Characterisation of offshore winds for energy applications — Research@WUR and Energies | Free Full-Text | Dutch Offshore Wind Atlas Validation against Cabauw Meteomast Wind Measurements (mdpi.com)
- Line (section 3.3): you need to address the uncertainty in the lidar measurements. are the differences that you find with ERA5 and/or modified ASCAT significant? Page 14: TNO report - DOWA validation against offshore mast and LiDAR measurements | Report | Dutch Offshore Wind Atlas

Other relevant literature:

- Comparing available Wind Farm Parametrisations for mesoscale models (Fitch and EWP best): Review of Mesoscale Wind-Farm Parametrizations and Their Applications | Boundary-Layer Meteorology (springer.com)
- Wind farm effects modelled with COSMO-CLM and Fitch WFP: [https://wes.copernicus.org/articles/9/697/2024/](https://wes.copernicus.org/articles/9/697/2024/)
- Quadruple collocation: KNMI Technical report - Uncertainty analysis of climatological parameters of the Dutch Offshore Wind Atlas (DOWA) | Report | Dutch Offshore Wind Atlas.
- Validation of HARMONIE+Fitch WFP with e.g. lidar measurements: A One-Year-Long Evaluation of a Wind-Farm Parameterization in HARMONIE-AROME - Stratum - 2022 - Journal of Advances in Modeling Earth Systems - Wiley Online Library
- Wake effects: [https://www.researchgate.net/publication/340838550_Long-range_modifications_of_the_wind_field_by_offshore_wind_parks_-_results_of_the_project_WIPAFF](https://www.researchgate.net/publication/340838550_Long-range_modifications_of_the_wind_field_by_offshore_wind_parks_-_results_of_the_project_WIPAFF)
- Internal boundary layer caused by change in surface roughness (coast): An effective parametrization of gust profiles during severe wind conditions - IOPscience

---

## Author Comment (AC1)

**"Ship-based lidar measurements for validating ASCAT-derived and ERA5 offshore wind profiles"**
*Rev v1*
Hugo Rubio, Daniel Hatfield, Charlotte Bay Hasager, Martin Kühn, and Julia Gottschall
**Authors response to reviewer comments**

We would like to thank the referees for their time and effort in reviewing our work. We appreciate their feedback and comments, and we have carefully considered their recommendations and requests to improve and clarify our work.

Below, we addressed all the referees' comments and reply to them point by point. First, the referee's comment is included in italics and bold font, followed by our answer, and when applicable, the new excerpt of the revised version of the manuscript (highlighted in blue).

**Anonymous Referee, Referee #1**

**Referee #1 general comments**

1) ***The mix of ASCAT and ERA5 and afterwards validating them against each other does not appear as a scientifically sound approach. ASCAT is assimilated into both NWP-models and ERA5. At the same time, sensible heat flux, 2m temperature and friction velocity from ERA5 is used in extrapolating ASCAT to 250m height and correcting for stability. Nowhere in the paper is this justified or discussed.***

The comparison between ASCAT extrapolated and ERA5 profiles is not intended as a validation against each other but rather as a means to explore similarities and differences in their retrieved values. This comparison pretends to provide some insights regarding how factors such as the temporal resolution and discretization of ASCAT, and the amount of data available may influence the final extrapolated values obtained using the statistical adaptation of MOST (Section 3.1). Additionally, this comparison aims to show the differences between these datasets along the region of study (due to factors such as coastal contamination) and before and after the application of the extrapolation methodology, i.e., at 10 m and 100 m height (Section 3.2). Finally, Section 3.3 presents the validation of extrapolated ASCAT and ERA5 values by comparing them against the reference lidar measurements. These clarifications have also been included in the manuscript for clarity (see Section 3).

It is worth noting that while ASCAT is assimilated in NWP models (and ERA5), there remains value in comparing these datasets due to their different characteristics (e.g. vertical, horizontal and temporal resolution) and the fact that the ASCAT retrievals are (direct) measurements, and not model data as ERA5, which results in different benefits and drawbacks, and therefore, different suitability for different applications. This might have been the motivation for the numerous scientific reports and peer-reviewed publications dedicated to this type of comparisons. E.g.: (Belmonte Rivas and Stoffelen 2019; Hauser et al. 2023; Duncan et al. 2019; Badger et al. 2016; Hasager et al. 2020; Hatfield et al. 2022; Takeyama et al. 2019; Takeyama et al. 2020). Some of them including the extrapolation approach also used in our study.

Finally, the utilization of the ERA5 parameters used for implementing the mean stability correction approach are listed in Section 2.4, together with the equations in which these

parameters are employed for the calculation of the stability correction factor: "…However, since we wish to develop an extrapolation method independent from in situ measurements, the mean temperature and heat fluxes in Eq. (1) are replaced by the ERA5 parameters air temperature at 2 m and surface sensible heat flux, respectively. Additionally, friction velocity values from ERA5 are also utilized…".

2) **The terms such as "ASCAT wind profiles" and "wind speed from ASCAT" when talking about wind in 100m are used in several places. Since satellites provide sea surface measurements, such terms should be defined early on (e.g. around line 60) with a short introduction to what it is.**

A clarification regarding ASCAT wind profiles has been added in Line 72: "…comparison of vertically extrapolated ASCAT wind profiles (hereafter referred to as ASCAT wind profiles) …".

Additionally, the employment of "ASCAT winds", "wind speed from ASCAT", and similar terminology have been reviewed throughout the manuscript, making necessary clarifications when referring to extrapolated or 10m measured ASCAT values.

3) **How many collocations between ship lidar and ASCAT in the open sea are there actually?**

As detailed in section 2.5, it is important to note that there is not a one-to-one collocation of ASCAT and lidar measurements, since the extrapolation methodology used for ASCAT only provides mean wind profiles within a considered time period (in this study, the duration of the measurement campaign). Therefore, there is not a specific number of collocations.

However, we have addressed this concern by incorporating a new figure to the manuscript (Figure 6 in the new version of the paper), which provides an overview of the ASCAT grid boxes used in the comparisons, as well as the amount of lidar data utilized at each of these grid boxes.

4) **The discussion of MOST in section 205-215 is confusing. A clearer overview of validity and why this can not be used, while Kelly and Gryning is valid, is needed. Later (line 371) the overestimated wind profiles based on this method get this comment: "This is due to the fact that these heights are well beyond the range of applicability of the extrapolation methodology employed (Kelly and Gryning, 2010)."**

The section has been revised with the aim of enhancing clarity and readability, while also properly highlighting the differences between the instantaneous MOST methodology for extrapolation and statistical adaptation used in this study, along with the potential advantages offered by the former approach.

5) **After seeing the poorer results near land on both sides, it is strange that these data are included further in the analysis (page 14 and figure 11). The issue is not investigated to any degree, it is only written that it is due to a number of different effects such as land contamination in ASCAT (line 335, 348, 376), coarse resolution of**

**ERA5 (line 340, 348, 378), and even wave breaking and surface slicks in the coastal zone (line 456)! But if there is land contamination in ASCAT (and it looks like it from figure 5), then the other effects are irrelevant because the measurements do not represent wind speed.**

As seen in Figure 14 of the new version of the manuscript (and Figures 13 in the old version), while nearshore grid points present higher deviations respect to lidar measurements, this is not uniformly applicable to all nearshore locations. Both ERA5 and extrapolated ASCAT show a significant agreement (even higher than those points farther away from shore) at some points very near to the harbors. Therefore, we believe including these results is pertinent, as excluding points from the comparison solely based on distance to shore may introduce a bias in the analysis. However, and in order to clarify and properly discuss these effects more comprehensively, several amendments have been made to the text:

- Page 14 of previous version of the manuscript highlights 3 main factors for the differences between the collocated and full campaign approach at near shore locations. In the new version of the manuscript, this section has been revised and we have included a detailed description of how the coarse resolution of ERA5 can lead to a different mean stability at coastal sites, resulting in different stability correction factors for the collocated and full campaign and therefore different wind speeds at 100 m (See lines 330 - 353).
- In Section 3.2, a more detailed discussion regarding the differences between ASCAT and ERA5 at 10 m has been added, including further details between the agreement of the two datasets depending on the distance to shore and therefore, isolating the higher deviation of ERA5 and ASCAT in nearshore conditions (See lines 374 – 382).
- Fig. 11 has been modified to also separate the effect of nearshore grid points in the error distribution presented.
- The discussion has been also modified to cover this topic.

Furthermore, we want to clarify that "wave breaking and surface slicks" are referred as a potential cause of coastal contamination, as evidenced in previous literature, rather than as a reasoning to explain deviations between the datasets: "…satellite measurements proximate to shorelines are susceptible to coastal contamination, occasioned by different factors such as waves breaking and surface slicks."

6) **Please revise the conclusions. At present the chapter does not summarize the main results and include statements which are not justified: "The long-term stability correction employed in this study demonstrated a strong performance for extrapolating ASCAT winds, yielding to a good agreement compared to the in situ measurements from the ship-based lidar measurements, despite the relatively constrained temporal window of the study." And "ASCAT derived wind profiles are a valuable asset for portraying offshore wind conditions at turbine operation heights, manifesting a level of accuracy similar to numerical model outputs."**

The conclusions of the manuscript have been revised.

**Referee #1 specific comments**

7) *Line 37: Why is shallow and deep water mentioned here? What is the definition of shallow and how is it used later on?*

We refer to near-shore and far-off offshore regions. The terminology has been reviewed and adjusted throughout the whole document.

8) **Line 42: "To overcome the limitations of in situ measurements and numerical models, satellite remote sensing devices have emerged as a potential alternative for characterizing ocean winds and climate over large areas, capturing the wind variability with a temporal coverage of over 15 years." It is strange to talk about a potential alternative when satellite remote sensing has been around for so long. What is meant by wind variability?**

The sentence may be misleading, and it has been revised to emphasize that satellite measurements are not a novel alternative, but they represent an additional option that can be considered when looking for diverse datasets applicable to offshore wind and climate studies.

"wind variability" refers to horizontal wind variability. This has been clarified in the manuscript.

9) **Figure 2: what height are these wind speeds recorded in?**

The wind speed at 100 m height is presented. This has been indicated in the text referring to this figure as well as in its caption.

10) **Lines 131-133: This statement sounds like a conclusion and does not belong in the data chapter.**

This sentence has been removed from the manuscript.

11) **Lines 145: please revise this sentence as it is unclear: " … being zero when having completely smooth surfaces and simultaneously increasing with the roughness".**

Sentence adjusted for clarity.

12) **Line 168: "By applying the IQR outlier detection, the impact of coastal contamination on the wind speed data is minimized, leading to more accurate and reliable results in nearshore areas." This statement sounds like a fact. Please modify and provide a reference.**

Sentence has been reformulated and a reference added as suggested.

13) **Line 186: the title of section 2.4 should include "vertical extrapolation".**

Section 2.4 title adjusted to "Satellite vertical extrapolation".

14) **Line 200: The objective is to validate this method, so naturally this method is used. However, it is not clear if it is because the authors expect that it provides better results than other methods. Why can single collocations of ERA5 and ASCAT not be used to extrapolate the values upwards? Or even the ERA5 wind profile?**

The decision to use the proposed methodology has considered few factors, rather than being solely based on the expectation of superior results compared to alternative methods. Firstly, the scarcity of available data due to the limited time extension of the measurement campaign, particularly given the coarse temporal resolution of ASCAT (about 2 measurements per day), which is not enough for the application of data-driven methodologies such as triple collocation or machine learning algorithms.

Additionally, the interest of the long-term correction approach comes from its potential better performance compared to the instantaneous correction approach. Previous studies showed that while numerical models can accurately capture average meteorological conditions over extended periods (Peña and Hahmann 2012), the accuracy of instantaneous stability information from these datasets is questionable, introducing additional uncertainty to extrapolated profiles using this instantaneous data (Badger et al. 2012). Another advantage of the long-term stability correction over the instantaneous correction is that we avoid calculating wind speed for conditions and heights outside the valid range of the MOST model. MOST is tailored to characterize turbulent fluxes within the surface boundary layer (Lange et al. 2004; Högström et al. 2006), but it has limitations when dealing with instantaneous data analysis, especially under stable conditions. The long-term adaptation of MOST, however, remains effective up to turbine operating heights, as it falls within the range where MOST is applicable.

In summary, this methodology was selected based on an evaluation of available data and the limitations of alternative approaches. Being our decision primarily driven by the pragmatic constraints of data availability and with the aim of evaluating a potentially better performing methodology for ASCAT extrapolation.

To clarify this in the manuscript, the corresponding section of the paper (Section 2.4 of the latest version) has been modified and extended for further clarification in this regard.

15) *Line 241: how were the values for C+ and C- chosen?*
The selection of values for the constants C- and C+ was based on an empirical validation, by comparing the theoretical distribution calculated from Eq. (2) (and dependent on the selected C- and C+ values) against the normalized probability density (NPD) function of the inverse Obukhov length derived from ERA5. Through this process, values were chosen to ensure that the theoretical distribution closely matched the ERA5 NPD across all the ASCAT grid boxes along the entire ship route.

At the moment, and to the authors' knowledge, no other methodology apart from this empirical validation (e.g. (Kelly and Gryning 2010)) has been presented in previous literature focusing on the application of this mean stability correction.

The corresponding section of the manuscript (Lines 345-351 of the new version) has been modified to clarify this methodology for the definition of C- and C+ values.

16) *Line 246: Remove long-term the second time in "Finally, the long-term stability correction of the mean long-term wind profile at a specific height z is calculated as:".*
Done.

17) *Line 255: The "theoretical" distribution from eq 2 is also using ERA5 data, so is there much point in comparing for the "full campaign"? As expected they are quite similar in figure 4.*
As mentioned in the response to comment 15, the theoretical distribution derived from Eq. (2) depends on the values of C± utilized. Therefore, comparing this theoretical

distribution with the NPD from ERA5 provides insight into the suitability of the selected C± for accurately representing the atmospheric stability.

Below, we have included a comparison between the theoretical and ERA5 distributions for the same location used in Figure 4 of the preprint manuscript, also considering the data from the full campaign. However, the theoretical distribution presented below has been calculated using C± values of 12 and 5, respectively, as utilized in a previous study (Optis et al. 2021). As can be observed, even with the inclusion of ERA5 data from the full campaign, the theoretical distribution struggles to closely resemble the NPD of the 1/L parameter calculated by ERA5.

In summary, we included this comparison because it shows that the data used for the derivation of the theoretical distribution, as well as the model configuration (i.e. C+- values selected), correctly represents the atmospheric stability calculated from ERA5.

[Figure]

Figure 1: Normalized probability density functions of inverse Obukhov length 1/L from ERA5 and theoretical distributions calculated from Eq. (2) using 12 and 5 for C- and C+. The same offshore location as in Figure 4d and full campaign data was considered for this plot.

18) ***Figure 4: add definition of "full campaign" and "collocated" to the figure caption.***
This clarification has been added to the figure caption.

19) ***Line 286: Please comment on the fact that there is land in two of the grid boxes and argue why they can be included in the analysis.***
A comment regarding this fact has been added in lines 312-315 of the new version of the manuscript.

20) ***Figure 6: the labels A-F are not used anywhere else.***
They are now used when commenting results from Figure 8 and Figure 11, where the differentiation of these locations is used.

21) ***Line 300: "...two different approaches…" insert collocation: "...two different collocation approaches…".***

Done.

22) **Figure 10b: Please include the pdf for values over the open sea only, so it is better documented that "... wind speed differences above this threshold correspond to those to near-shore grid points." (line 356).**
Figure has been modified.

23) **Line 390: "...ERA5 appears to outperform ASCAT …": Why "appears"?**
Adjusted to "…ERA5 outperforms ASCAT…"

24) **Figure 13: Please add a line to show the number of collocations along the track.**
A new Figure (Figure 6 in the new version of the manuscript) has been included with the amount of collocated lidar data along the ship route.

25) **Page 20: Notice that there are some typos: highes lidat . And the use of "resemble" in line 404.**
Text on this page has been reviewed and typos corrected.

26) **Lines 450-456: "The comparison between ASCAT and ERA5 winds revealed a good agreement between the two datasets.": Please be more specific. Was the agreement really good? If so, how about the mix of the two datasets as mentioned above? This long paragraph consists of hypotheses that are not investigated. Was it expected that ASCAT and ERA5 should perform well near the coast?**
This entire paragraph has been rewritten for better clarity and including a more specific discussion of the obtained results.

27) **Line 469: "...ASCAT exhibited a closer similarity to the lidar wind profile than ERA5.": where? It seems to be a coincidence that the values are close for the mean wind profile in figure 11a since it is a mix of nearshore and offshore values.**
As pointed out by the reviewer, the better agreement of ASCAT overall profile is indeed a result of averaging profiles from both onshore and offshore locations. This aspect has been clarified in the manuscript.

28) **Line 475: "... the notorious overestimation suffered by ASCAT is evident…": this doesn't fit with figure 13 where there are values below -2?**
This paragraph has been revised, and this particular assertion has been removed.

29) **Line 480: What is the value of employing ASCAT and the long-term height extrapolation using ERA5, versus just using ERA5?**
ASCAT are observations, and ERA5 is not, and only the stability information of ERA5 is assumed accurate to obtain ASCAT values at higher heights.

**Referee #2 general comments**

1) *My main problem with this paper is that your aim is to provide a means for offshore wind resource assessment which is an alternative for ERA5, but*
    a. *You still need ERA5 to modify ASCAT*

    Yes indeed – however, it is important to clarify that ASCAT provides wind field **measurements**, and **only the stability information for ERA5** is required for this methodology.
    In addition, the non-stationarity of ship-based lidar measurements leads to a comparison not focused on a single location, but within an extensive region. Consequently, using stability information from stationary measurements is unfeasible, due to the lack of an extensive enough network of measuring devices. Therefore, using ERA5 (or other potentially applicable numerical models) is essential, and a requirement imposed by the nature of ship-based lidar measurements.

    b. *You do not take into account wake and blockage effects. This is probably okay for now (there are no wind farms yet in the Baltic according to https://map.4coffshore.com/offshorewind/), but it will become a problem during the lifespan of the wind farm when more wind farms are built (the effect is already significant on the North Sea). ERA5 does not take these effects into account and ASCAT is too course to measure them (SAR can: WINS50 - Winds of the North Sea in 2050 | Publications I Wijnant, A Stepek (2023): Fit(ch) for shipping Wind farm wake effects at 10 m height, KNMI WINS50 Report.). The only way to predict how much power a wind farm will produce during its lifespan is to understand and properly model wind farm effects and then to use this knowledge/ these models for future wind farm scenarios. The old method for wind resource assessments (measure-correlate-predict) no longer works.*

    We acknowledge the referee's comment regarding wake effects. But as our study focuses on comparing ASCAT and ERA5 wind data in the Baltic Sea region, the issue of wake effects falls outside the scope of our investigation. While wake effects are indeed important considerations for wind farm development, they do not directly impact our findings, given the absence of nearby wind farms in our study area.
    Our paper does not aim to predict wind farm production, but to evaluate the performance of ASCAT and ERA5 data against reference lidar measurements, providing insights regarding their potential and limitations in characterizing offshore winds. We have tried to make clear statements about the goal of our paper in several parts of the manuscript:

    From the title: Ship-based lidar measurements for validating ASCAT-derived and ERA5 offshore wind profiles

From the abstract: …For this reason, this study presents a comprehensive comparison between wind profiles derived from the Advanced Scatterometer (ASCAT) satellite observations and the ERA5 reanalysis dataset against ship-based lidar measurements in the Northern Baltic Sea….

From the introduction: The objective of this paper is to assess the accuracy of ASCAT-derived wind speed profiles in the nearshore and offshore locations of the Northern Baltic Sea by conducting a comprehensive comparison against ship-based lidar measurements. … To the authors' knowledge, this study represents the first comprehensive comparison of vertically extrapolated ASCAT wind profiles (hereafter referred to as ASCAT wind profiles) to wind turbine operational heights against non-stationary in situ measurements, covering a wide horizontal extent that extends from nearshore to offshore locations…

From the discussion: The objective of this study has been to evaluate the accuracy of ASCAT-derived wind speed profiles for the characterization of offshore winds at turbine operating heights in the Northern Baltic Sea… Subsequently, the analysis incorporated a comparison of both gridded datasets against in situ observations obtained from a novel ship-based lidar campaign.

From the conclusion: Satellite-borne scatterometers and numerical models are two potential alternatives for characterizing offshore winds. This study undertakes an intercomparison of these datasets and validates them against reference ship-based lidar measurements.

Finally, SAR measurements were contemplated as an alternative to ASCAT due to their higher resolution and potential better performance in near-shore areas. However, given that SAR's lower temporal resolution (one overpass every couple of days) and the relatively short period of the campaign, we opted for ASCAT in order to maximize the amount of collocated data and ensure the consistency of the statistical metrics evaluated. A clarification regarding this has been added to the discussion section of the manuscript:

… However, for this study, SAR measurements were not contemplated due to their lower temporal resolution compared to ASCAT and the relatively short duration of the campaign. This decision aimed to maximize the amount of collocated data and ensure the consistency of the statistical metrics evaluated.

c. *Mean values of the wind are not relevant if you want to predict power: you need to look at correlation on a 10 min (or hourly) basis, especially for wind speeds between cut-in and rated (power curve).*

As mentioned above, our paper does not aim to assess or predict wind farm energy production, but to compare ASCAT (and ASCAT extrapolation) and ERA5, and then evaluate both of them against reference lidar measurements, providing insights regarding their potential and limitations in characterizing

offshore winds. Additionally, we must note that mean values are not the only statistical measure used in this study. Please see the results section, where additional metrics (error distributions, bias, nRMSE, confidence intervals, or box plots, to mentions some) have been applied and analyzed.

2) *The concept of doing lidar measurements on a ferry is interesting and probably even more interesting on the North Sea where you no doubt can measure wind farm effects this way. How to compare these measurements to ERA5 is as you describe not straightforward (ship-motion compensation algorithm), comparing them to ASCAT I presume even harder. You can only do that when ASCAT is available which is around 09 and 19 UTC when the ferry is in the harbour or reasonably close to the coast and you say that there the ASCAT-signal is disturbed? Are there other measurements available near the harbours (or in the Baltic Sea) that you could use to compare the lidar measurements to?*

We appreciate the referee's insight into the challenges associated with comparing ship-based lidar measurements to ERA5 and ASCAT data. For this reason, Section 2.5 includes a detailed description of the collocation methodology used in this study, which aims to address factors such as the different temporal and spatial resolution of the different datasets and the non-stationary nature of the ship, thus providing a robust comparison methodology.

In addition, to provide further information regarding the amount of collocated lidar-ASCAT data used in this comparison, we have included a new figure (Figure 6) in Section 2.5. This figure illustrates the number of lidar retrievals collocated at each ASCAT grid box along the ship route.

3) *I think that comparing ERA5 to 'ASCAT extrapolated to hub height with ERA5' is not very useful (and scientifically sound). What would be interesting is to do triple collocation with (1) ASCAT-winds extrapolated to hub height with your method (or different methods) based on ERA5, (2) ship based lidar measurements and (3) a mesoscale model in hindcast mode, for now without Wind Farm Parametrization (WRF? Unless you can get hold of COSMO-CLM or HARMONIE?). The aim of your paper then would be the best possible extrapolation of ASCAT to hub height. Why that is useful is something you will have to explain (not for wind resource assessments).*

The application of triple collocation may require a high amount of data for a successful performance. However, due the relatively short measurements campaign and the rather low temporal resolution of ASCAT, the amount of data available in this study is not sufficient for the application of triple collocation. For this reason, and although the investigation of the potential application of triple collocation is an interesting point for future work, it is not in line with the scope of this paper. This has been indicated in the manuscript:
Furthermore, and although previous literature highlighted the good performance of data-based extrapolation methods, the limited time extension of the measurement campaign results in an insufficient amount of data to implement these approaches in this study….

**Referee #2 specific comments**

4) *Bias ERA5 at hub height 0.5 m/s is also what is found on the North Sea in Characterisation of offshore winds for energy applications — Research@WUR and at Cabauw in Energies | Free Full-Text | Dutch Offshore Wind Atlas Validation against Cabauw Meteomast Wind Measurements (mdpi.com). NEWA comparable to ERA5 (at least on the North Sea). Undisturbed winds in DOWA (2008-2018) and WINS50 (2019-2021) are much better than ERA5 (including correlation) and the domain covers most of the Baltic Sea, but hourly data unfortunately not available for 2022 and 2023 when you have the lidar measurements (Home | Dutch Offshore Wind Atlas; WINS50 - Winds of the North Sea in 2050).*
The two additional references recommended by the referee have been added (Section 3.3). Regarding the additional measurement datasets suggested by the referee, as the referee rightly pointed out, they are not within the time frame of the ship-based lidar campaign and thus not applicable for this study.

5) *Line 52: Unclear sentence: Lastly, the trustworthiness of satellite retrievals remains a knowledge gap, due to the lack of available in situ datasets for validation especially in deep water regions.*
This sentence has been removed from the manuscript.

6) *We know that the Dutch part of the North Sea (DEEZ) does not experience a trend in offshore wind speed, only an Interannual Variability (IAV) of 5 and 4% for sites in the northern part of the DEEZ and between 4 and 4.5% in the southern part of the DEEZ (Inter-annual wind speed variability on the North Sea | Report | KNMI Projects). Is any information like this available for the Baltic Sea? How representative is 28-6-22 until 21-2-23 for the wind climate in the Baltic Sea? This you can check e.g. with ERA5 data (compare ERA5 28-6-22 - 21-2-23 to ERA January1940-now).*
Interannual variability has not been considered in this study since it does not have any influence on the study. The three datasets used comprise the same time frame of about 6 months, therefore no interannual changes are captured.
To avoid confusions, some clarifications have been added:
- In Section 2.2: The implemented ASCAT data processing for this study focused on satellite measurements retrieved during the period of the ship-based lidar measurement campaign, and includes …
- In Section 2.3: It must me noted that only ERA5 data within the time frame of the measurement campaign have been used in this study.
- Along the entire manuscript, we have adapted the terminology "long-term stability correction" to "mean stability correction", to avoid potential confusions regarding the time period covered by the data used in this study.

7) *Line 143: (fig 3) maybe I missed it, but what ASCAT data did you use?*
Section 2.2 has been rearranged for better clarity on this. Now it is clear that the ASCAT data used was the corresponding to the product id WIND_GLO_WIND_L3_NRT_OBSERVATIONS_012_002, downloaded from the Copernicus Marine Data Service (CMS) and corresponding to the period of the measurement campaign.

8) *Line 189: (typo): Several methodologies to vertical satellite extrapolation… not to, but for.*
Corrected.

9) *Line 201: This method involves a long-term correction of atmospheric stability effects, obtained from the numerical model dataset ERA5, along with an adaptation of the MOST to vertically extrapolate the satellite wind measurements. What is long-term about it? Why the name 'long-term extrapolation method'?*
The term "long-term" was adopted from the previous literature employing the same methodology (e.g. (Optis et al. 2021; Badger et al. 2016; Hasager et al. 2020)). However, we recognize that this may be confusing for readers, since our study only applies this methodology for the time duration of the ship-lidar measurement campaign.
To address this, we have revised the terminology "long-term stability correction" with a more appropriate term such as "mean stability correction".

10) *Lines 203-205 not clear: do you mean that a wind profile can be stable up to a certain height and above unstable and that this 'long-term extrapolation method' can handle this?*
No, we mean that the stability correction factor calculated in Eq. (4) can switch from positive to negative values with varying heights, because it combines both stable and unstable terms.
Section 2.4 has been revised for further clarity regarding the extrapolation methodology employed.

11) *Line 206-217 not clear: what is the difference between the 'instantaneous stability correction' and the 'long-term stability correction'?*
The answer to comment 14 from Referee #1 also clarifies this, highlighting some differences between these two methods, as well as some benefits of the average stability correction approach against the instantaneous. Furthermore, this part of the manuscript has been revised for further clarity.

12) *Line 221-268: so basically the stability correction has only 2 values for C per height which are the same for the whole Baltic Sea, one for stable and one for unstable. It does not matter how (un)stable the atmosphere is or whether the grid box is near the coast or further offshore: correct?*
The mean stability correction method uses a single set of C- and C+ values for each ASCAT grid point in which the extrapolation is made, with no variation based on height.
As in previous literature, the definition of C+ and C- has been done empirically, by comparing the theoretical stability distribution calculated from Eq. (2) against the normalized probability density (NPD) function of 1/L derived from ERA5. Through this method, we selected values of C- and C+ which allow a representative theoretical stability distribution at ASCAT grid points along the ship route. This is illustrated with two example sites, one near the shore and a second far from it.
Additionally, as mentioned in the manuscript, the same C- and C+ values were applied to all grid points.

This clarification has been added to the paper:

…the selection of these values for this study was based on a empirical validation, by comparing the theoretical distribution calculated from Eq. (2) against the normalized probability density (NPD) function of 1/L derived from ERA5. Through this process, values were chosen to ensure that the theoretical distribution closely represented the ERA5 NPD of 1/L across all the ASCAT grid boxes along the entire ship route. Furthermore, identical values of C± were applied to all ASCAT grid points.

Also, in section 3.1:

… Finally, as mentioned in Section 2.4, the same values of the semi-empirical constant C± are assumed for the entire region, instead of using a site-specific definition of these constants.

13) **Line 302: (fig 7). You compare the collocated approach (only ERA5 stability information at moments when ASCAT overpasses is considered) to the full campaign approach (all ERA5 stability information from the whole duration of the campaign is used). Both approaches do not include spring which is often the most stable period (cold sea water and warmer air above). Also, mean wind speed is not really relevant for wind resource assessments. So I do not really understand the sentence: 'This highlights the robustness of the employed methodology and indicates that the dataset size allows for an accurate characterization of atmospheric stability conditions during the campaign and along the entire ship track'.**

We acknowledge the concern regarding the exclusion of stability information from spring in our comparison. However, it is important to note that the stability information utilized aligns precisely with the timeframe of the lidar measurement campaign (June 2022 to February 2023). Therefore, by incorporating stability information from this period (and no further months not "seen" by the measurements), we ensure consistency in our comparison, as the extrapolation of ASCAT reflects the stability conditions during the measurement campaign. As well, as mentioned in our answers to comment 1, this paper does not aim to assess or predict wind farms energy production, but to evaluate the performance of ASCAT and ERA5 data against reference lidar measurements, providing insights regarding their potential and limitations in characterizing offshore winds.

Finally, the sentence highlighted by the referee directs to the fact that both collocation approaches yield highly similar speeds at 100 m, indicating a significant level of robustness. Therefore, the choice between the full campaign and collocation approach has a minimal impact (except near shore) on the final extrapolated ASCAT profiles. Additionally, the good agreement between the theoretical and empirical stability distributions, as illustrated in Fig. (4), further supports the consistency of the methodology. We have revised this paper excerpt to enhance the clarity of this:

Both strategies for calculating the stability correction factor and the corresponding wind profiles demonstrate a high level of agreement, except for some nearshore locations. This, together with the revealed representativeness of the theoretically derived stability distributions observed in Fig. (4) highlights the robustness of the mean stability correction approach in characterizing the atmospheric stability

conditions during the period covered by the measurement campaign and along the entire ship track.

14) **Line 307-320: 'pronounced instability in the morning?' Why would ERA5 produce stronger unstable conditions (lower 1/L) in the morning at Nynashamn? What do we know about the water temperature near Nynashamn and how it is modelled by ERA5 (shallower/warmer water between Bedaron and the mainland maybe?)? ERA5 has grid boxes of 31 km2 so model values are probably very land-contaminated in that area: can you show the ERA5 grid boxes near the harbours? What is the prevailing wind direction? Basically ERA5 and ASCAT are not very good in coastal area: maybe you should take them out of your analyses?**
The ERA5 grid box corresponding to Nynäshamn harbour has a land mask of 56%. Therefore, the stability daily cycle at this location presents higher variations regarding to other sites. This, together with the temporal discretization of ASCAT overpasses leads to a more unstable mean distribution of the stability conditions, resulting in a lower wind speed compared to the full campaign approach, as can be derived from Eq. (4).

Section 3.1 has been revised to provide a detailed explanation about this.

15) **Line 341 (fig 9 10m validation): compare to Validation of DOWA ('undisturbed wind' = HARMONIE without WFP) with ASCAT (too coarse to measure wind farm effects) at 10 m height: TNO report - DOWA validation against ASCAT satellite winds | Report | Dutch Offshore Wind Atlas.**
The results have been compared with the suggested reference (Section 3.2).
"Similar results were reported in (Duncan et al. 2019) in their comparison of ASCAT and ERA5 wind speeds at 10 m over the North Sea and the Dutch coast. Specifically, (Duncan et al. 2019) found a nearly zero bias in far-offshore locations and approximately 0.6 m s$^{-1}$ in coastal regions."

16) **Line 341 (fig 9 100m validation): so we can conclude that ERA5 is internally fairly consistent (profile depends on ERA5 stability parameters)?**
We appreciate the referee insight although we believe further analysis would be needed to derive final conclusion about the internal consistency of ERA5.

17) **Line 354: '… highlighting the consistent overestimation of wind speed from ASCAT at this height'. At 100m this is not ASCAT, but ASCAT extrapolated with ERA5. And we all know that ERA5 is not unbiased at 100m (0.5 m/s underestimation) so you cannot draw this conclusion. See also Line 364/365.**
The term "overestimation" in this context refers to a comparison between ERA5 and extrapolated ASCAT data, indicating that extrapolated ASCAT values exceed those of ERA5 at 100 m height. We have tried to clarify this in the manuscript, together with a specific mention to extrapolated ASCAT values: "overestimation of wind speed from ASCAT" has been replaced by "…overestimation of extrapolated ASCAT wind speeds at this height compared to ERA5.".

A 0.5 m/s bias is expected between ERA5 and in situ measurements, but not necessarily in comparison between ERA5 extrapolated ASCAT values. An

overestimation of extrapolated ASCAT winds relative to ERA5 at 100 m is not only plausible but a conclusion directly drawn from the results shown in Fig. 10 and 11 (from the new manuscript version).

**18) Line 364/365: Characterisation of offshore winds for energy applications — Research@WUR and Energies | Free Full-Text | Dutch Offshore Wind Atlas Validation against Cabauw Meteomast Wind Measurements (mdpi.com)**
References have been included in the new version of the manuscript.

**19) Line (section 3.3): you need to address the uncertainty in the lidar measurements. are the differences that you find with ERA5 and/or modified ASCAT significant? Page 14: TNO report - DOWA validation against offshore mast and LiDAR measurements | Report | Dutch Offshore Wind Atlas**
The discrepancies observed when comparing lidar observations against both ERA5 and extrapolated ASCAT are significantly larger than the uncertainty attributable to floating lidar measurements (below 2% according to (Wolken-Möhlmann et al. 2022)).

A clarification on this regard has been added in the discussion of the paper, including a reference to the paper suggested by the referee.

**Other relevant literature provided by the referee:**

- **Comparing available Wind Farm Parametrisations for mesoscale models (Fitch and EWP best): Review of Mesoscale Wind-Farm Parametrizations and Their Applications | Boundary-Layer Meteorology (springer.com)**
  We thank the referee for the reference suggestions; however, we think this reference is not pertinent to our study since wake effects and wind farm parameterization lie outside the scope of our research.

- **Wind farm effects modelled with COSMO-CLM and Fitch WFP:** *https://wes.copernicus.org/articles/9/697/2024/*
  We thank the referee for the reference suggestions; however, we think this reference is not pertinent to our study since wake effects and wind farm parameterization lie outside the scope of our research.

- **Quadruple collocation: KNMI Technical report - Uncertainty analysis of climatological parameters of the Dutch Offshore Wind Atlas (DOWA) | Report | Dutch Offshore Wind Atlas.**
  We thank the referee for the reference suggestions; however, we think this reference is not pertinent to our study since quadruple collocation is not discussed or covered in our paper. As indicated in our answer to comment 14 of referee #2 and the manuscript itself, these type of approaches are not suitable for this study.

- **Validation of HARMONIE+Fitch WFP with e.g. lidar measurements: A One-Year-Long Evaluation of a Wind-Farm Parameterization in HARMONIE-AROME - Stratum - 2022 - Journal of Advances in Modeling Earth Systems - Wiley Online Library**

We thank the referee for the reference suggestions; however, we think this reference is not pertinent to our study since this paper covers neither wake effects nor wind farm parameterization issues.

- ***Wake effects:*** *https://www.researchgate.net/publication/340838550_Long-range_modifications_of_the_wind_field_by_offshore_wind_parks_-_results_of_the_project_WIPAFF*
  We thank the referee for the reference suggestions; however, we think this reference is not pertinent to our study since wake effects lie outside the scope of our research.

- ***Internal boundary layer caused by change in surface roughness (coast): An effective parametrization of gust profiles during severe wind conditions – IOPscience***
  We thank the referee for the reference suggestions; however, we think this reference is not pertinent to our study since this paper.

**Publication bibliography**

Badger, Merete; Peña, Alfedo; Bredesen, Rolv Erlend; Berge, Erik; Hahmann, Andrea N; Badger, Jake et al. (2012): Bringing satellite winds to hub-height. In *Proceedings of EWEA 2012 - European Wind Energy Conference & Exhibition European Wind Energy Association (EWEA), Copenhagen, Denmark, 16-19 April 2012*.

Badger, Merete; Peña, Alfredo; Hahmann, Andrea N.; Mouche, Alexis A.; Hasager, Charlotte B. (2016): Extrapolating Satellite Winds to Turbine Operating Heights. In *Journal of Applied Meteorology and Climatology* 55 (4), pp. 975–991. DOI: 10.1175/JAMC-D-15-0197.1.

Belmonte Rivas, Maria; Stoffelen, Ad (2019): Characterizing ERA-Interim and ERA5 surface wind biases using ASCAT. In *Ocean Sci.* 15 (3), pp. 831–852. DOI: 10.5194/os-15-831-2019.

Duncan, J B; Marseille, G J; Wijnant, I L (2019): DOWA validation against ASCAT satellite winds.

Hasager, Charlotte B.; Hahmann, Andrea N.; Ahsbahs, Tobias; Karagali, Ioanna; Sile, Tija; Badger, Merete; Mann, Jakob (2020): Europe's offshore winds assessed with synthetic aperture radar, ASCAT and WRF. In *Wind Energ. Sci.* 5 (1), pp. 375–390. DOI: 10.5194/wes-5-375-2020.

Hatfield, Daniel; Hasager, Charlotte Bay; Karagali, Ioanna (2022): Comparing Offshore Ferry Lidar Measurements in the Southern Baltic Sea with ASCAT, FINO2 and WRF. In *Remote Sensing* 14 (6), p. 1427. DOI: 10.3390/rs14061427.

Hauser, Danièle; Abdalla, Saleh; Ardhuin, Fabrice; Bidlot, Jean-Raymond; Bourassa, Mark; Cotton, David et al. (2023): Satellite Remote Sensing of Surface Winds, Waves, and Currents: Where are we Now? In *Surv Geophys* 44 (5), pp. 1357–1446. DOI: 10.1007/s10712-023-09771-2.

Högström, Ulf; Smedman, Ann-Sofi; Bergström, Hans (2006): Calculation of Wind Speed Variation with Height over the Sea. In *Wind Engineering* 30 (4), pp. 269–286. DOI: 10.1260/030952406779295480.

Kelly, Mark; Gryning, Sven-Erik (2010): Long-Term Mean Wind Profiles Based on Similarity Theory. In *Boundary-Layer Meteorol* 136 (3), pp. 377–390. DOI: 10.1007/s10546-010-9509-9.

Lange, Bernhard; Larsen, Søren; Højstrup, Jørgen; Barthelmie, Rebecca (2004): The Influence of Thermal Effects on the Wind Speed Profile of the Coastal Marine Boundary Layer. In *Boundary-Layer Meteorol* 112 (3), pp. 587–617. DOI: 10.1023/B:BOUN.0000030652.20894.83.

Optis, Mike; Bodini, Nicola; Debnath, Mithu; Doubrawa, Paula (2021): New methods to improve the vertical extrapolation of near-surface offshore wind speeds. In *Wind Energ. Sci.* 6 (3), pp. 935–948. DOI: 10.5194/wes-6-935-2021.

Peña, Alfredo; Hahmann, Andrea N. (2012): Atmospheric stability and turbulence fluxes at Horns Rev-an intercomparison of sonic, bulk and WRF model data. In *Wind Energ.* 15 (5), pp. 717–731. DOI: 10.1002/we.500.

Takeyama, Yuko; Ohsawa, Teruo; Shimada, Susumu; Kozai, Katsutoshi; Kawaguchi, Koji; Kogaki, Tetsuya (2019): Assessment of the offshore wind resource in Japan with the ASCAT microwave scatterometer. In *International Journal of Remote Sensing* 40 (3), pp. 1200–1216. DOI: 10.1080/01431161.2018.1524588.

Takeyama, Yuko; Ohsawa, Teruo; Tanemoto, Jun; Shimada, Susumu; Kozai, Katsutoshi; Kogaki, Tetsuya (2020): A comparison between Advanced Scatterometer and Weather Research and Forecasting wind speeds for the Japanese offshore wind resource map. In *Wind Energ.* 23 (7), pp. 1596–1609. DOI: 10.1002/we.2503.

Wolken-Möhlmann, Gerrit; Bischoff, Oliver; Gottschall, Julia (2022): Analysis of wind speed deviations between floating lidars, fixed lidar and cup anemometry based on experimental data. In *J. Phys.: Conf. Ser.* 2362 (1), p. 12042. DOI: 10.1088/1742-6596/2362/1/012042.

---

## Referee Report (RR1)

**Ship-based lidar measurements for validating ASCAT-derived and ERA5 offshore wind profiles**

Hugo Rubio, Daniel Hatfield, Charlotte Bay Hasager, Martin Kühn, and Julia Gottschall

Despite all my comments, I still think you present promising work, worth publishing. But you need to be more serious about the feedback. Quite a lot of the comments from the previous review have not been addressed.

The added value of this work is (1) a technique to compare ship based lidar measurements to model values, (2) that we can reliably extrapolate these measurements to heights relevant for wind energy (as long as we avoid areas less than 40 km from the coast and near wind farms, the latter becoming increasingly challenging by the way) and (3) use those to validate weather models that include wind farm effects (wakes/blockage). We can maybe extend this technique to higher resolution satellite (SAR)? This work has added value for wind energy because of 2 and 3.

There are a few things that I think need to be addressed in the paper:

- Uncertainty in the lidar measurements. are the differences that you find with ERA5 and/or modified ASCAT significant? See e.g. page 14: TNO report - DOWA validation against offshore mast and LiDAR measurements | Report | Dutch Offshore Wind Atlas
- Your method is not robust with more/larger WFs (there are no wind farms yet in the Baltic according to https://map.4coffshore.com/offshorewind/, but you expect significant growth). Ship-based lidar measurements may be affected by wind farms (WF), ERA5 definitely does not take WF effects into account and ASCAT is too coarse to measure WF effects (at least in detail: then you need SAR).
- How does your method compare to assimilating ASCAT into the NWP reanalysis like it was done in DOWA (point 1 in Innovations in the DOWA project | DOWA project | Dutch Offshore Wind Atlas)?
- You basically show in your paper that ASCAT and ERA5 should not be used closer than 40 km from the coast (validation results based on ship-based lidar). That is a conclusion that I miss in your paper. As far as I know the ASCAT coastal product is only valid 15 km away from the coast and ERA5 has problems with abrupt changes in surface roughness, such as on the coast. A model (such as ERA5) assumes a grid box average surface roughness for a combination of land and water whereas the wind feels land or water. The larger the grid box size, the larger the problem (ERA5 grid box size 31 km). So basically ASCAT and ERA5 have quality issues near the coast and this is what you find confirmed in your paper.

Comments from earlier review that have not been addressed yet are e.g.: (1) Are there other measurements that you can compare to lidar measurements in harbor (where ASCAT and ERA5 are particularly inaccurate)? (2) Have you considered triple (or quadruple collocation) to assess uncertainties (there are also uncertainties in your lidar measurements! What are they)? (3) Have you considered using other wind climatology's such as NEWA GMD - The Making of the New European Wind Atlas – Part 2: Production and evaluation (copernicus.org)?

Comments more in detail:

- Line 3: typo: observations
- **Line 9/10: The comparison reveals a close agreement between ASCAT and ERA5 beyond 40 km distance from the coast.** Unclear what you mean: close agreement between two different approaches (account for stability)? At 10m height or also extrapolated to hub heights?
- Line 10/11: (Extrapolated) ASCAT tends to **significantly** overestimate the mean wind speed derived from lidar measurements, while ERA5 exhibits a consistent underestimation. I assume the difference between lidar measurement and (Extrapolated)ASCAT/ERA5 is larger than the lidar measurement uncertainty?
- Line 21: **However**, in situ …
- Line 26/27: **Floating lidar systems can be moved to different locations, but generally measure at one location for a certain period of time. With profiling lidar systems installed on cruising ships it is possible to provide reliable wind profile measurements over larger areas.**
- Line 27/30: (can be formulated shorter/clearer): **Before profiling lidar systems on cruising ships can become a generally accepted alternative for offshore met masts and floating lidar, specific challenges have to be overcome such as validation against reference data and quantifying the associated uncertainty (Rubio and Gottschall, 2022). Still, ship based lidar has already been used in different wind energy related studies.** In Wolken-Möhlmann…
- Line 34-38: However, while numerical models have demonstrated good performance in shallow-water offshore regions compared to in situ measurements (Witha et al., 2019b), they often fail to describe the spatial and temporal variability of wind with sufficient accuracy and detail. I suggest an alternative text: **Numerical weather prediction (NWP) models in re-analyses mode** are commonly used … spatial coverage. However, while numerical models have demonstrated good performance in shallow-water offshore regions compared to in situ measurements (Witha et al., 2019b; **Wijnant et al, 2019**), they have problems with areas with large changes in surface roughness, such as the coast. The larger the grid box size, the larger the problem because the model assumes a grid box average surface roughness for a larger area (whereas the wind feels land or water, not a combination). Also most re-analyses do not take into account the (changing) effect of wind farms on the atmosphere (except: https://wins50.nl/).**
- Line 38-41: This limitation arises from factors such as the inaccurate parameterization of the model variables or the insufficient temporal and spatial resolution of the models' output data. Furthermore, the lack of in situ measurements in deeper offshore regions hinders the validation of these datasets, leading to increased uncertainties in derived wind statistics for such locations. I suggest an alternative text: **Each NWP model has its own limitations (caused e.g. by grid and domain size and physical modelling and parametrisation choices). This results in uncertainties in wind statistics based on these NWP models and these uncertainties can be quantified when validation measurements (incl. measurement uncertainties) are available. This is however often a problem for hub heights, especially for far-offshore locations with deep water.**
- Line 42-44: To overcome the limitations of in situ measurements and numerical models, satellite remote sensing devices have emerged as a potential alternative for characterizing ocean winds and climate over large areas, capturing the wind

variability with a temporal coverage of over 15 years. I suggest an alternative text: **Scatterometer (wind) measurements from satellites are a welcome additional source of information in these data sparse areas.** Several studies …

- Line 47-53: Fluffy writing: does not make it clearer and there are some mistakes in it. I suggest an alternative text: **The ASCAT coastal product is available since 2007 and provides high quality offshore wind measurements on a 12.5 km grid spacing for locations further than 15 km from the coast. The ASCAT wind speed bias is less than -0.23 ms-1 in coastal areas (15- 50 km from the coast) and -0.29 ms-1 elsewhere (TNO report - DOWA validation against ASCAT satellite winds | Report | Dutch Offshore Wind Atlas ). However, ASCAT has its limitations: only available twice a day (around 09:30 and 21:30 UTC) and stability dependent assumptions have to be made to derive turbine height winds from the ASCAT 10m winds.**
- Line 54: The Baltic Sea is an area of great interests for offshore wind development…
- Line 64-71: I suggest that you change sequence of what you write to make it clearer, e.g. **To derive wind profiles from the ASCAT coastal product 10 m measurements, we employ the long-term stability correction approach presented in Kelly and Gryning (2010) and implemented in Badger et al. (2016). For this, we utilize the stability information from ECMWF Reanalysis 5th generation (ERA5)and compare two different collocating methods to evaluate the potential influence of the limited temporal resolution of satellite overpasses in the ASCAT extrapolated profiles. Not only the ASCAT derived wind profiles, but also the wind profiles from ERA5 are then compared to the lidar profiles.**
- Line 75-76: … of the reliability and accuracy of satellite measurements **derived wind statistics** for offshore wind characterization at wind energy relevant heights.
- Line 86: What is the accuracy of your lidar measurements? If you want to compare your measurements to model data, you will have to be able to tell whether the difference that you find is significant (outside the measurement uncertainty). See e.g. TNO report - DOWA validation against offshore mast and LiDAR measurements | Report | Dutch Offshore Wind Atlas
- Line 104-105: the motion (take the s out) effects
- Line 117: fig 2b is the daily cycle the ship (lidar) experiences because it is connected to the location of the ship. It is not how the wind depends on the hour in the day (which is what normally is meant by 'daily cycle'). Maybe use a different name to avoid confusion (wind speed daily cycle plots normally give highest wind speeds during the day), e.g. Wind speed ship daily cycle.
- Line 31: Therefore (?), the ….
- Line 155/156: … available **horizontal grid spacings** of 12.5 km and 25 km
- Line 159: what do you mean by Both of these (?) are implemented (?) at…
- Line 168-170: By applying the IQR outlier detection, the impact of coastal contamination on the wind speed data is minimized, leading to more accurate and reliable results in nearshore areas.
- Line 189: Several methodologies **for vertical extrapolation of satellite measurements** …
- Line 260-264: As observed, considering the stability information from the full campaign results in a better theoretical distribution compared to the collocated approach. Although the difference is minimal at the harbor site, it is more pronounced at the offshore location, where a significant underestimation of unstable stability occurrence is observed. The harbor site presents a rather symmetric distribution

**Met opmerkingen [WI(7):** Assume you used that?

**Met opmerkingen [WI(8):** Better. Someone might otherwise read this in 10 years time and think ASCAT is available since 2016

**Met opmerkingen [WI(9):** That is not the same as resolution!!! Ask Ad Stoffelen KNMI.

**Met opmerkingen [WI(10):** If I am correct: please check

**Met opmerkingen [WI(11):** You write: 'Lastly, the trustworthiness of satellite retrievals remains a knowledge gap, due
to the lack of available in situ datasets for validation especially in deep water regions'. I left this out because I think it is incorrect: ASCAT has been extensively validated (besides: its quality does not depend on water depth). Ask Ad Stoffelen KNMI.

**Met opmerkingen [WI(12):** Apparently not correct for the Baltic where you mention 1-3 times a day?

**Met opmerkingen [WI(13):** And yet: you do not mention the effect of wind farms (WF) on the atmosphere. ERA5 is without WF effects, ASCAT is too course, at least for detail (you need SAR for that), but your ship based lidar may measure the effects up to 100 (?) km from a WF. I think you should at least mention WF effects in the paper and tell what the consequences of these WF effects are for your method.

Can you quanlify what you mean with near shore (I assume > 15 km from shore otherwise ASCAT not valid)?

**Met opmerkingen [WI(14):** Not the same as resolution

**Met opmerkingen [WI(15):** I assume this is part of the ASCAT coastal product? Is nearshore more than 15 km from the coast?

around zero, meaning that both unstable and stable atmospheric conditions are equally represented. However, the offshore site exhibits a higher occurrence of unstable conditions, compared to the stable side of the curve.

- Line 192-193: …. performance at different vertical and horizontal constraints.
- Figure 6. Six locations used for the comparison of the datasets. The approximate distance to the nearest shore is indicated, in km, below of each site. Please add: **Location A is the harbour of Nynäshamn (Sweden) and location D the harbour of Hanko (Finland).**
- Line 241: In this study, the values for the C± constants have been set to 6 and 4 for the stable and unstable portions, respectively.
- Line 307-311: First, the coastal contamination of near shore areas leads to the removal of some ASCAT overpasses for data quality reasons, leading to a reduced number of ASCAT observations in **these** areas. Consequently, the insufficient number of valid wind speed measurements obtained from the collocated approach introduces a biased representation of the prevailing stability conditions during the campaign period.
- Line 313-315 (from previous review): 'pronounced instability in the morning?' Why would ERA5 produce stronger unstable conditions (lower 1/L) in the morning at Nynashamn? What do we know about the water temperature near Nynashamn and how it is modelled by ERA5 (shallower/warmer water between Bedaron and the mainland maybe?)? ERA5 has grid boxes of 31 km2 so model values are probably very land-contaminated in that area: can you make a plot of the ERA5 grid boxes near the harbours? What is the prevailing wind direction? Basically ERA5 and ASCAT are not very good in coastal area: maybe you should take them out of your analyses?
- Line 315-316: This results in a lower wind speed compared to the full campaign approach, as can be derived from Eq. 4.
- Line 316-317: In contrast, the other locations do not exhibit such pronounced daily stability cycles, and therefore, smaller differences are reported between the two approaches.
- Line 317-320: Finally, as mentioned in Section 2.4, the same values of the semi-empirical constant C± are assumed for the entire region, instead of using a site-specific definition of these constants. Therefore, the suitability of the selected values may not be optimal for certain locations, leading to an anomalous theoretical representation of the empirical atmospheric distribution.
- Line 322-323: Add names harbour to fig 7
- Line 322-324: This highlights the robustness of the employed methodology and indicates that the dataset size allows for an accurate characterization of atmospheric stability conditions during the campaign and along the entire ship track.
- Figure 9 basically shows you that ASCAT winds look unrealistic near the coast at 10 and (more so) at 100m. Especially near the Swedish coast where the wind blows predominantly from land to sea, wind near the coast should be lower than further offshore. So this figure proves that you cannot use your method near the coast for 2 reasons: (1) quality of ASCAT, (2) grid size of ERA5 (averages surface roughnesses of land and sea in grid box, therefore wrong for both wind from land and from sea). Small scale effects such as sea breeze and low level jets (you mention these in line 341) don't have a significant effect on your mean values.
- Line 341-342 (fig 9 10m validation): (from previous review) compare to Validation of DOWA ('undisturbed wind' = HARMONIE without WFP) with ASCAT (too coarse to measure wind farm effects) at 10 m height: TNO report - DOWA validation against ASCAT satellite winds | Report | Dutch Offshore Wind Atlas. Because you use ERA5

stability info to calculate ASCAT-derived wind speeds at 100m height, the difference you see at 100m should mainly be because of differences at 10m, right?

- Line 342: Figure 10a illustrates the **difference** in wind speed between ASCAT and ERA5 at 10 m and 100 m

> **Met opmerkingen [WI(24):** Wrong use of the word 'disparity' (nothing unfair about this difference).

- Lines 347-350: This discrepancy in the nearshore areas can be explained by the combination of too high wind speeds retrieved by ASCAT due to coastal contamination and ERA5's inability to properly resolve the coastal atmospheric phenomena and its coarse horizontal resolution that leads to the omission of the flow phenomena variations cause**d** by the small islands present in **these** coastal regions.

> **Met opmerkingen [WI(25):** It has nothing to do with coastal atmospheric phenomena or flow phenomena variations (do you mean sea breezes?). It has everything to do with 'land roughness contamination' of the roughness in the coastal grid cells

- Figure 10 shows you that you should not use your method within about 40 km from the coast (you should expect 31 km because of the grid size of ERA5 and what I explained earlier)
- Line 355-356: Nonetheless, the majority of grid points exhibit wind speed differences below ±1 m s-1. As previously discussed, wind speed differences above this threshold correspond to those **of** near-shore grid points.

> **Met opmerkingen [WI(26):** This big difference of 1 m/s in mean values is not the bias, but the max difference, right?

- Line 400: what do you mean with the word 'trend 'here? The word trend is used for change in time (e.g. climate change), but this is not what you mean…
- Line 400-403: Notably, the western area of the ship route **(longitude below 18.5 degrees)** exhibits the **largest** errors for both **ASCAT-derived winds (using ERA5) and ERA5 winds**, with maximum differences exceeding 3 m s-1 at all elevation levels. This indicates that wind speed estimation **cannot be done accurately enough in these areas** with **ASCAT and/or ERA5 because (1) poor quality of ASCAT coastal product closer than 15 km from the coast and (2) ERA5 grid box size (surface roughness in land-water grid boxes on the coast problematic).**

> **Met opmerkingen [WI(27):** Is it possible to add distance to the nearest coast to fig 13? In this figure we are looking at winds at 60m, 150m and 220 m, so at ASCAT derived winds (with ERA5). The ASCAT coastal product is only valid 15 km or more out of the coast as far as I know…

- Line 404-405: highlighting the different shear resemble obtained from each of the datasets and their different representation of the wind profiles

> **Met opmerkingen [IW28]:** Sentence unclear: shear resemble?

- Line 406: (mentioned in previous review: seems like a good idea to write that your results are conform what others have found): Bias ERA5 at hub height 0.5 m/s is also what is found on the North Sea in Characterisation of offshore winds for energy applications — Research@WUR and at Cabauw in Energies | Free Full-Text | Dutch Offshore Wind Atlas Validation against Cabauw Meteomast Wind Measurements (mdpi.com). NEWA comparable to ERA5 (at least on the North Sea). Undisturbed winds in DOWA (2008-2018) and WINS50 (2019-2021) are much better than ERA5 (including correlation) and the domain covers most of the Baltic Sea, but hourly data unfortunately not available for 2022 and 2023 when you have the lidar measurements (Home | Dutch Offshore Wind Atlas; WINS50 - Winds of the North Sea in 2050).
- Line 408-409: ERA5 usually underestimates the wind speed, this is more pronounced at higher elevations and in the eastern part of the ship track. In contrast, ASCAT mainly overestimates compared to the **lidar** (typo) measurements.

> **Met opmerkingen [IW29]:** If anything: more pronounced in western part of ship track (not eastern) which also makes more sense with prevailing westerly winds (land contamination ERA5 grid surface roughness)

- Line 418-419: When comparing the two datasets, ERA5 shows a smaller nRMSE in the majority of the studied region, except in the Eastern area near the harbour in Hanko. What is your explanation for this? Does it have anything to do with time of overpass ASCAT, the location characteristics?
- Line 419-421: When comparing the bias and nRMSE shown by the two datasets, the average absolute bias across the entire region is smaller for ASCAT compared to ERA5 at the three heights considered (see Fig. 13). Differently, as can be observed in Fig. 14, most of the locations reveal a smaller nRMSE for ERA5 than for ASCAT. Bit confusing. I suggest an alternative text: **So for all heights considered the bias (compared to the lidar measurements) of the ASCAT-derived wind speeds is**

**smaller than the bias of the ERA5 wind speeds (fig 13), but for most of the region (except for the eastern part of the region near the Finnish coast) the nRMSE of the ERA5 wind speeds is better (fig 14).**

- Line 427-428: The objective of this study has been to evaluate the accuracy of ASCAT-derived wind speed profiles for the characterization of offshore wind resources at turbine operating heights in the Northern Baltic Sea.
- Line 431: … obtained from **a** (typo) novel ship-based lidar campaign
- Line 435: … that machine learning-based techniques for extrapolating satellite winds could surpass the long-term correction method employed herein. Questionable English. I suggest an alternative text: … **that machine learning-based techniques for extrapolating satellite winds could work better than the long-term correction method that was used in this study.**
- Line 436-437: However, the limited amount of data available over the campaign period hinders the implementation of such data-driven approaches.
- Line 441-442: The methodology revealed a remarkable congruence between these two approaches across most of the area examined, thus underscoring the robustness of the methodology.
- Line 443-446: This divergence can be attributed to the limited availability of valid wind speed measurements in the collocated approach, the constraints of considering atmospheric conditions solely during morning and evening hours, and the generic definition of the empirical constants C± required for the calculation of the theoretical stability distributions at each site.
- Discussion: please rewrite given all comments given (running out of time to give detailed comments)
- Line 486-492: Finally, it is imperative to highlight that although the disparities in wind speeds between ASCAT and ERA5 relative to lidar are generally small in far-offshore regions, their cumulative impact over a large-scale wind energy project can still have relevant implications for energy production estimates and financial assessments. Therefore, continued efforts to refine both **satellite based** measurements and numerical models are essential to enhance the accuracy of wind resource assessments for offshore wind energy applications. The diverse characteristics and insights into wind patterns derived from satellite-derived observations, numerical models, and ship-based lidar measurements suggest that an integrative approach, harnessing the collective strengths of these datasets, could yield substantial gains in the accuracy and reliability of offshore wind statistics derivation.

**Met opmerkingen [IW30]:** Goal wind resource assessments?
As I said before, this work is interesting for wind energy, but only because we can use the ship-based lidar measurements for validation of mesoscale or LES models that include the effect of wind farms. We can then use these models with changed wind farm scenarios to predict the wind resource in the future. Bear in mind that mean values of the wind are not relevant if you want to predict power: you need to look at correlation on a 10 min (or hourly) basis, especially for wind speeds between cut-in and rated (power curve).

**Met opmerkingen [WI(31)]:** Not an ML expert, but is the fact that you have a short campaign really the limiting factor? You have ERA5 and ASCAT measurements for a much longer period, so can you not perform your long-term stability correction? What I do know is that ML cannot reproduce events that have not occurred yet (extremes).

**Met opmerkingen [WI(32)]:** Not convinced this conclusion is justified (see earlier comments).

**Met opmerkingen [WI(33)]:** Rethink this conclusion also based on earlier remarks

**Met opmerkingen [WI(34)]:** The ASCAT measurements extrapolated to 100m with ERA5 are not representative for wind in or near wind farms and therefore do not give accurate wind resource assessments (neither does ERA5 for areas with wind farms or Measure Correlate Predict for areas where the number/size of wind farms is changing). So what we need to do is further develop Numerical Weather Prediction models that include solving the effect of wind farms (for which we need measurements for validation) and run these models for current and future wind farm layouts. ML is a useful tool, but cannot be used to derive extremes in wind climate.
You should also bear in mind that there is no significant trend in the wind climate (apart from at 10m over land) but a strong Inter Annual Variability (IAV). This is the case for the North Sea, but most likely also for the Baltic? Do you know? If there is a strong IAV, then it is important to assess how representative the period you look at is for the wind climate. For the Dutch part of the North Sea (DEEZ) the IAV is 3.5 and 4% for sites in the northern part of the DEEZ and between 4 and 4.5% in the southern part of the DEEZ (Inter-annual wind speed variability on the North Sea | Report | KNMI Projects). Is any information like this available for the Baltic Sea? How representative is 28-6-22 until 21-2-23 for the wind climate in the Baltic Sea? This you can check e.g. with ERA5 data (compare ERA5 28-6-22 - 21-2-23 to ERA January1940-now). So what is the added value of having these 100m wind speeds based on ASCAT? Compared to lidar, the ASCAT derived 100m wind are maybe more accurate than those from ERA5, but only available twice a day. Should we just not assimilate ASCAT in ERA5 and focus more on how useful this ship based lidar technique is to get validation measurements for models including wind farm effects (wakes/blockage)? That is what I like about this work.

Relevant literature you should include (or at least consult):

- Characterisation of offshore winds for energy applications — Research@WUR
- Energies | Free Full-Text | Dutch Offshore Wind Atlas Validation against Cabauw Meteomast Wind Measurements (mdpi.com)
- Comparing available Wind Farm Parametrisations for mesoscale models (Fitch and EWP best): Review of Mesoscale Wind-Farm Parametrizations and Their Applications | Boundary-Layer Meteorology (springer.com)
- Wind farm effects modelled with COSMO-CLM and Fitch WFP: https://wes.copernicus.org/articles/9/697/2024/
- Quadruple collocation: KNMI Technical report - Uncertainty analysis of climatological parameters of the Dutch Offshore Wind Atlas (DOWA) | Report | Dutch Offshore Wind Atlas.
- Validation of HARMONIE+Fitch WFP with e.g. lidar measurements: A One-Year-Long Evaluation of a Wind-Farm Parameterization in HARMONIE-AROME - Stratum - 2022 - Journal of Advances in Modeling Earth Systems - Wiley Online Library
- Wake effects: https://www.researchgate.net/publication/340838550_Long-range_modifications_of_the_wind_field_by_offshore_wind_parks_-_results_of_the_project_WIPAFF
- Internal boundary layer caused by change in surface roughness (coast): An effective parametrization of gust profiles during severe wind conditions - IOPscience

---

## Referee Report (RR2)

**Ship-based lidar measurements for validating ASCAT-derived and ERA5 offshore wind profiles**

General remarks:

- The paper has improved a lot! I like your figures.
- There are however still revisions necessary I think.
- I have also corrected some of the English

**Abstract.** Text still gives the impression that this work contributes to "accurate characterization of offshore wind resources". Maybe this is true for the Baltic for now, but certainly not for places with wind farm (effects). I suggest this alternative text:

Because offshore in-situ wind measurements at turbine operating heights are scarce, ECMWF Reanalysis 5th generation (ERA5) data are often used for offshore wind resource assessments. There are however a few disadvantages of using ERA5: it has a rather course grid spacing which makes it less useful for coastal areas and it does not include wind farm effects, so it can only be used for wind resource assessments in areas without wind farms. This study presents a comprehensive comparison between wind profiles derived from the satellite-based Advanced Scatterometer (ASCAT) satellite observations and the ERA5 reanalysis dataset against ship-based lidar measurements in the Northern Baltic Sea for a period without wind farms. The aim is to investigate the applicability of ship-based lidar measurements for validating these datasets and to better understand the reliability, accuracy and limitations of ASCAT- and ERA5-derived wind statistics for offshore wind characterization at wind turbines operating heights when there are no wind farms. To extrapolate ASCAT observations at sea level to turbine rotating heights, a mean correction of atmospheric stability effects based on ERA5 and a probabilistic adaptation of the Monin-Obukhov similarity theory (MOST) was were implemented. The comparison between the two gridded.. etc

> **Met opmerkingen [WI(1):** If you use "wind resource assessment" (or "characterization of offshore wind resources") which effectively means the same) as a reason why your work is relevant, then you should add this.

**Line 44-45:** Each NWP model comes with inherent limitations due to factors like grid resolution, physical modelling, and parameterization choices (e.g. wind farm parametrisations or the lack thereof).

**Line 46-47:** However, conducting such validation is particularly challenging in deep-water offshore regions, where in situ measurements are sparse.

> **Met opmerkingen [IW2]:** There are fewer measurements at sea than on land, but I am not convinced there are fewer measurements in deep than in shallow water... what did you base this on?

**Line 77-79** (typo): To the authors' knowledge, this study represents the first comprehensive comparison of vertically extrapolated ASCAT winds  profiles (hereafter referred to as ASCAT wind profiles) from 10 m height up to wind turbine operational heights against non-stationary in situ measurements, covering locations near the coast and further offshore. a wide horizontal extent from nearshore to offshore locations.

**Line 80-82:** Therefore, this work aims to contribute significantly to a better understanding of the reliability, limitations, and accuracy of satellite measurements derived wind statistics and ERA5 wind data for offshore wind characterization at wind energy-relevant heights in areas without wind farms.

> **Met opmerkingen [IW3]:** For me the most interesting part of your work is the comparison of ship-based lidar to ASCAT-profiles. If in areas without wind farm disturbances, the ASCAT-profiles validate well against the ship-based lidars, there is reason to believe that we can do wind resource assessments with higher resolution ASCAT (KNMI now working on 5.7 km) or SAR for European waters (Copernicus Marine Service) in areas where there are wind farm(effects). And then we are talking...

**Line 87:** The discussion of these findings and the main extracted conclusions are included in Sections 4 and 5, respectively.

**2 Data and Methods**

**Line 89-91**: This section describes the three datasets used in this work. In addition, the methodology used for processing the different 90 datasets is explained in detail, as well as the methodology to extrapolate ASCAT winds and the collocation approach used for their comparison against the ship-based lidar measurements.

**Line 104**: The campaign took place from 28 June 2022  until 21 February 2023

**Line 105**: ... ship-based lidar system was used with  a vertical profiling Doppler lidar WindCube WLS7v2, ...

**2.1 Ship based lidar measurements:** I still miss info on the accuracy of the measurements from the WindCube WLS7v2 in this section. Also what you added in lines 567-573 (answer to my question 16) is not really info on accuracy. So you assume (or know? reference?) that the accuracy of a ship-based lidar with motion recorder is comparable to the accuracy of a floating lidar? Add in section 2.1: the accuracy of a ship-based lidar with motion recorder is (assumed to be) similar to the accuracy of a floating lidar. According to Dhirendra et al (2016) this is 3.1%-4.2% for heights of 92m in the wind speed range 4m/s-16m/s (pg 15 TNO report - DOWA validation against offshore mast and LiDAR measurements | Report | Dutch Offshore Wind Atlas).

[ Line 567-573: "Consequently, the mean values derived from lidar measurements may exhibit biases that vary depending on the time slots during which measurements were acquired at particular locations. Additionally, it is acknowledged that lidar measurements, like any other observational data, are subject to inherent uncertainties that may impact the results (Duncanet al., 2019b; Rubio and Gottschall, 2022). Nevertheless, the observed deviations between the lidar measurements and both extrapolated ASCAT and ERA5 significantly exceed the magnitude of potential discrepancies attributable to floating lidar uncertainties (at turbine rotor heights roughly 3-4% see section 2.1), which can be up to approximately 2 % with mast-mounted anemometers as lower limit reference (Wolken-Mohlmann et al., 2022)". ]

**2.2 ASCAT**

**Line 150:** What is a nadir gap?

**Line 170-172**: Despite the application of these quality filters, ASCAT seems to overestimate wind speeds  near the coast (as shown later in this report in fig 12), likely due to coastal contamination effects (Stoffelen et al., 2008; Lindsley et al., 2016).

**2.3 ERA5**

**Line 179-180**: provides hourly estimates of a wide range of atmospheric, land surface and oceanic variables with a 0.25° x 0.25° latitude-longitude grid resolution (31x31 km), covering the period from 1950 to present.

**2.4 Satellite vertical extrapolation**

**Line 211-212**: So there are two methods for stability correction: mean stability correction and instantaneous stability correction. Compared to the instantaneous stability correction approach, applying the mean stability correction avoids the need to calculate wind speeds under stability ...

**Line 218-221**: Another advantage of the  mean stability correction is that the numerical models used for this method can accurately capture

average meteorological conditions over extended periods (Peña and Hahmann, 2012). The stability information of data used for instantaneous stability correction is (generally?) less accurate because the measurements are for a single location or a limited time span. This introduces , whereas the accuracy of instantaneous stability information from these datasets is questionable, introducing additional uncertainty to extrapolated profiles using this instantaneous data (Badger et al., 2012)

**Line 224-225**: However, a relevant drawback of the mean stability correction is that everything gets averaged out and site- or time-specific information  information from in-situ measurements is not included.

**Line 237**: You use ERA5 to derive L and you select values of C that give NPD of 1/L closest to ERA5. Does that mean that all differences between ASCAT and ERA5 at higher levels are mainly due to differences at sea level (or 10m) because the (stability dependent) extrapolation to higher levels is equal?

**Line 250**: In this study, the values for the C± constants have been set to 6 and 4 for the stable and unstable portions, respectively. These values are the same for all ASCAT grid points (both near coast and further offshore) and for the whole period (regardless of e.g. time of day and season).

**Line 309-310**: This situation may lead to coastal contamination and excessively high wind speed retrievals within these grid boxes.

**3 Results**

**Section 3.1**

**Line 329:** Figure 8 illustrates the differences in wind speed at 100 m height between the collocated and the full campaign approache.

**Line 336-338**: Consequently, the collocated approach in these areas may have insufficient stability information available, potentially introducing a biased representation of the theoretical stability distribution during the campaign period.

**Line 344-347**: As can be observed, the more unstable conditions just before midday at Nynäshamn harbour due to land-contamination (red line) "weigh" more in the mean stability assessment if you just consider the collocated periods (orange shadows) instead of the full period.  This leads  to a larger stability correction factor in absolute terms (despite its negative sign at this location), and consequently, to lower wind speeds compared to the full campaign approach, as derived using the equations described in Section 2.4.

**Line 347-350**: This  unstability at location A is attributed to the coarse resolution of ERA5, resulting in land contamination of the grid box at the harbour location, where  land  covers 56% of the grid box surface. Therefore, the daily stability cycle  is more  similar to that of an onshore site.

**Line 352**: The period of  highest un then occurs around midday when the surface heating is most intense.

**Line 354**: Un reaches its  minimum (the negative value of 1/L closest to 0) in the late evening and stays relatively constant until the following morning.
* * *
**Met opmerkingen [WI(7):** Sentence not clear: from which datasets? You do not mean datasets from numerical models, but that is how this sentence reads. I have tried to re-write what I think you mean, but this is not my expertise.

**Met opmerkingen [WI(8):** Sentence not clear: maybe this is what you mean?

**Met opmerkingen [WI(9):** My question 27: so basically mean stability correction is based on an average (time and space) stability distribution?

**Met opmerkingen [WI(10):** Counter-intuitive: land contamination gives an overestimation of the surface roughness and an underestimation of the surface wind. So please explain.

**Met opmerkingen [IW11]:** My old question 28 has not been answered. Too few ASCAT measurements increases uncertainty, but does not necessary lead to bias. Maybe the reason is that a higher percentage of land-contaminated ERA5 data are used in the collocated stability correction than the full approach?

**Met opmerkingen [WI(12):** My old question 29 is answered: 10 UTC is around midday and if this gridpoint is very land-contimated, this explains why it is zo unstable.

**Met opmerkingen [WI(13):** My old question 30 is not sufficiently answered. I can not see in the formulas in 2.4 how a higher negative value of 1/L leads to a larger stability correction (so please explain). And how does a larger stability correction lead to lower wind speeds (and at what level)? Less unstable (more weight to surface friction effect) tends to result in a lower wind speeds at the surface: is that what you mean?

**Met opmerkingen [WI(14):** In meteorology we call it unstablity, not instability.

**Met opmerkingen [WI(15):** What you have written is wrong

**Met opmerkingen [WI(16):** Again: what you have written is wrong

**Line 355**: In contrast, locations B to E are purely offshore (with a land fraction of 0%) and therefore exhibit almost no diurnal cycle because the atmospheric stability is mainly determined by the sea water temperature. There is however a seasonal cycle that was not taken into account.

**Line 357-359**: Finally, at Hanko harbour (location F) there is more of a daily stability cycle than offshore, but a lot less than at Nynäshamn harbour (location A). There are two reasons for the difference between Nynäshamn (A) and Hanko (F): (1) The gridbox at Hanko (F) contains a significantly lower land-fraction: 6% compared to 56% at Nynäshamn (A) and (2) with predominantly W-SW winds, the wind at Hanko (F) is mostly from sea to land and at (Nynäshamn (A) is from land to sea.

**Line 364-367**: Given the minimal differences in the wind speeds at 100 m depicted in Fig. 8, and thus the similar wind profiles obtained using both approaches, subsequent sections of this paper will only consider the full campaign approach because this approach is expected to provide more representative wind profiles along the complete ship route.

**Fig 9**: Daily cycle of the stability parameter (1/L) at the six evaluated locations A-F from Fig. 7. All values of 1/L are below zero indicating an unstable atmosphere. Long(itude) 18° corresponds to the harbour of Nynäshamn (Sweden) and long(itude) 23° to the harbour of Hanko (Finland). The orange shadows indicate the time periods when ASCAT overpasses are available and are therefore the only time periods included  in the collocation approach.

**Section 3.2**

**Line 372-378**: As can be observed when comparing the spatial variation shown by the two datasets at 10 m, ERA5 exhibits higher mean wind speeds in the areas farthest from the shore, but the wind speed near the coast is lower.  This is because ERA5 has a grid-box size of 31x31km, so part of the selected grid boxes (only grid boxes with ASCAT data so at 12.5 km from the coast) are still land-contaminated in ERA5 (and assume a surface roughness that is too high and therefore a 10m wind that is too low). Again, the effect of the prevailing W-SW winds can be seen: the land affects particularly the areas where the wind predominantly blows from land to sea (Swedish coast). Similar effects can be seen at 100m height. ~~This discrepancy occurs because, despite the filtering process for the ASCAT dataset, the coastal contamination still affects ASCAT measurements, leading to excessively high mean values in nearshore areas. The effect of coastal contamination in the ASCAT map is particularly visible in the 100 m height map, where the highest mean wind speeds are located along the perimeter of the region with available data.~~

**Line 397**: As to be expected, both datasets consistently show higher wind speeds at 100 m than at 10 m height.

**Line 380-383**: For 10 m height, t wind speed averaged over all included gridpoints is  7.61 m s⁻¹ (ASCAT) and 7.15 m s⁻¹  (ERA5)which means that  the difference (U$_{ASCAT}$ − U$_{ERA5}$) is 0.46 m s⁻¹. When only locations  more than 20 km from the shore are included, this difference reduces to

approximately 0.16 m s$^{-1}$. However, only including locations within 20 km from the shore increases the difference to 0.98 m s$^{-1}$.

**Line 386-390**: For 100 m height, the wind speed averaged over all included gridpoints is , 9.31 m s$^{-1}$ ASCAT) and 8.67 m s$^{-1}$ ERA5) and the difference 0.64 m s$^{-1}$. If only more than 20 km from shore locations are included, the difference is only slightly reduced to 0.43 m s$^{-1}$.  So land-contamination in ERA5 is less relevant at 100 m height than at 10 m height, which is what we expect (surface roughness affects wind at lower levels more than at higher levels). The differences between ASCAT and ERA5 at 10 and 100m can be attributed to  two (or one?) key factors: first, the inherent difference between the datasets at 10m (e.g. the gridbox sizes: ERA5 still land-contaminated near the coast, ASCAT not), second, the mean stability correction approach used to extrapolate ASCAT

**Line 396-398**: This  difference between in nearshore areas can be explained by the combination of excessively high wind speeds retrieved by ASCAT due to coastal contamination and ERA5's inability to properly resolve the coastal atmospheric phenomena and small-scale wind flow variations due to its coarse horizontal resolution.

**Line 398-401**: The differences become smaller moving further offshore and almost negligible at distances further than 40 km from the shore: around 0.2 m s$^{-1}$ at 10 m height and 0.4 m s$^{-1}$ at 100 m height

**Question 37**: Figure 10 shows you that you should not use your method within about 40 km from the coast (you should expect 31 km because of the grid size of ERA5 and what I explained earlier) Results presented in Figure 11 (Figure 10 in first submitted version of the manuscript) show higher discrepancies between ERA5 and ASCAT at both 10m and 100m (within these 40km distance to shore). However, since the extrapolation methodology used in this study does not affect the data at 10m, we cannot conclude that the method itself should not be used within 40 km from the coast. Rather, we believe a more accurate conclusion is that, within this region, higher uncertainty is expected in both ERA5 and ASCAT values, as evidenced by the larger differences observed due to the limitations of these datasets (e.g. ERA5 grid size as mentioned by referee). Therefore, we want to highlight that this is not due to a limitation of the method, but a limitation of these datasets. This has been discussed in the manuscript. While the extrapolation may contribute to some additional uncertainty, as seen by the consistently larger bias at 100m compared to 10m (also explicitly mentioned in the manuscript), the key limitations regarding the applicability closer or further away from the shore lie in the datasets themselves, not in the methodology employed in this study.

**Section 3.3**

**Line 417**: mean profile bias is consistently positive (indicating ASCAT overestimation compared to the lidar measurements), with the magnitude depending

**Line 426**: significantly outperforms ASCAT profiles, which overestimates the wind speed even at 10 m height, highlighting the

**Met opmerkingen [IW24]:** My old question 34 was not answered: Because you use ERA5 stability info to calculate ASCAT-derived wind speeds at 100m height, the difference you see at 100m should mainly be because of differences at 10m, right?

**Met opmerkingen [IW25]:** No: you only used the full dataset for figure 10 (assuming this is still about figure 10).

**Met opmerkingen [IW26]:** Again: ?

**Met opmerkingen [IW27]:** See my old remark 36: I do not think "coastal atmospheric phenomena" (sea breezes?) and small scale wind variations (low level jets?) affect your mean values. This is all because of ERA5 land-contamination (which indirectly affects ASCAT winds extrapolated to 100m). Please rethink your conclusions.

**Met opmerkingen [IW28]:** Part of the method is that you use ERA5 data for the extrapolation of ASCAT to 100m, so I do not agree with your answer.

**Line 427-429**: Additionally, it is striking to observe the substantial deviation of the ASCAT stability corrected profiles from the logarithmic profiles, particularly at heights above 50-100 m, as a consequence of a stability distribution that is not representative enough of these specific sites.

**Line 431:** A statistical analysis of the wind speed deviation between ASCAT and ERA5 compared to the lidar observations

**Line 432-435:** Each box plot is calculated considering the wind speed difference of all the grid boxes with lidar data along the whole route of the ship, but grid boxes closer than 20 km away from the shore have been excluded to minimize the effect of ASCAT coastal contamination in the derived statistics.

**Line 441-443:** This indicates that both ERA5 and ASCAT are probably within measurement uncertainty of the lidar measurements for these heights.

**Line 443-445**: ERA5 consistently underestimates the wind speed across the entire profile, with this negative bias becoming increasingly pronounced with altitude and reaching the largest negative mean bias of around 0.2 m s$^{-1}$ at 270 m, which is (probably) still an insignificant difference with the lidar measurements if you take into account the accuracy of the lidar measurement itself

**Line 445-446:** As opposed to ERA5, ASCAT profiles exhibit a persistent overestimation of wind speed relative to the lidar across all heights. This overestimation increases significantly above 170 m.

**Line 447-450:** For ERA5, the IQR is almost the same for  all heights, with values around 0.5 m s$^{-1}$, suggesting the quality of ERA5 wind speeds does not depend on height. In the case of ASCAT, IQR displays a slight decrease with height, highlighting the larger and more consistent overestimation at higher altitudes.

**Line 451-455:** The whiskers analysis provides further insights into the discrepancies between the two datasets. For ERA5, the lower whiskers extend further into negative values as altitude increases, with the larger underestimations reaching approximately 0.8 m s$^{-1}$ at 270 m.  ASCAT's whiskers reveal a different pattern particularly noteworthy are the upper (positive) whiskers that extend significantly beyond the lower whiskers at 270 m, illustrating once again the  tendency for ASCAT to specifically overestimate wind speeds at greater heights.

**Line 464-469:** Notably, the western area of the ship route (longitude below 18.5 degrees) exhibits the largest errors for both ASCAT-extrapolated and ERA5 winds, with maximum differences up to about 5  m s$^{-1}$ at all elevation levels. In the eastern area of the ship route, there are maximum differences up to about 4 m s$^{-1}$. This indicates that wind speed estimation cannot be done accurately enough in  coastal areas using these datasets, first, because of the poor quality of ASCAT in areas closer to the coast, and secondly, due to the  ERA5 grid box size of 31 km, which means that for distances closer than 31 km to the coast the surface roughness in ERA5 gridboxes is overestimated because of land-contamination. This effect will be larger near the harbour of Nynäshamn in Sweden (longitude 18°) than near the harbour of Hanko in Finland (longitude 23°) because with a prevailing W-SW'ly winds, the wind at Nynäshamn blows mostly from land to sea, advecting 'land surface roughness contamination' to sea grid points (at Hanko where the wind mostly blows from sea to land, 'water surface roughness contamination' is advected to land grid points).

**Met opmerkingen [IW29]:** I do not understand want you want to say here. The logaritmic profile represents a wind profile for neutral atmospheric stability. Do you mean that at the harbour sites the ASCAT-profile seems to follow a logaritmic profile up to 50-150m (so no stability correction occurs). What does that mean?

**Met opmerkingen [IW30]:** Why not 30 km to eliminate the land contamination in ERA5 (that you have used for stability correction)? Line 438-439 can go: you already mentioned that you did not use grid points closer than 20k from the coast.

**Met opmerkingen [IW31]:** This is what you have to assess. No measurement is without uncertainty.

**Met opmerkingen [IW32]:** But at what level does the difference become significant (bigger than lidar measurement uncertainty)?

**Met opmerkingen [IW33]:** Correct?

**Met opmerkingen [IW34]:** Nowhere in fig 13b so the whiskers reach -1.3 m/s?

**Met opmerkingen [IW35]:** Fig 13a: only at 270m height.

**Met opmerkingen [IW36]:** This is what I see in fig 14.

**Met opmerkingen [IW37]:** This is what I see in fig 14.

**Met opmerkingen [IW38]:** There are no ASCAT-values less than 12.5 km from the coast, so what are those "ASCAT" values in fig 14 for < 12.5 km distance from the coast? See also earlier remark about quality of ASCAT near coast (after quality control and just looking at sites > 12.5 km from the coast).

Also, a fairly course model like ERA5 is  unable to capture the small-scale wind flow variations in these complex locations and the intricate interactions in the coastal boundary layer influenced by both land and sea.

**Line 474-475**: It can be noted that, although ERA5 usually underestimates the wind speed, this is more pronounced at higher elevations and in the western part of the ship track.

**Line 484-490**: When comparing the two datasets, ERA5 shows a smaller nRMSE in the majority of the studied region, except in the 485 eastern area near the harbour in Hanko. This may be attributed to the differing spatial resolutions of the two datasets. In the east of 22 degrees longitude, the finer resolution of ASCAT mitigates the impact of coastal contamination, enabling it to capture local conditions more effectively and consequently leading to a lower average nRMSE in this region. In contrast, the coarser resolution of ERA5 may be insufficient to adequately represent the average wind characteristics in this area. Conversely, in the western part of the studied area, with features more intricate topography and a higher density of small islets within a few tens of kilometres from the mainland shoreline, ASCAT measurements are more susceptible to coastal contaminated.

**4. Discussion**

**Line 510-513:** For the mean stability correction methodology, we had to decide whether we would use the collocated or the full dataset. A disadvantage of the collocated dataset is that the stability information may be biased because ASCAT overpasses only twice a day at roughly the same time.  The collocated and full dataset  strategies demonstrated remarkable agreement across most of the examined area, resulting in very similar wind speed

**5 Conclusion**

**Met opmerkingen [IW39]:** Basically at a coast where there is an abrupt change of the surface roughness an internal boundary layer (IBL) is formed where the flow adjusts to the new surface roughness. This is what affects your results in coastal areas, mainly near the Swedish harbour where the wind mstly blows from land to sea. So why is the underestimation of the wind speed more pronounced at higher levels? I think because the Internal Boundary Layer (IBL) has not reached these heights yet. So basically the wind profile has not adapted to the surface roughness of the sea.
Internal Boundary Layer (IBL): Internal boundary layer growth following a step change in surface roughness | Boundary-Layer Meteorology . The height of the IBL grows the father away you are from the place where the surface roughness changes (coast). So at the Swedish harbour with a W-SW wind, the wind speed adapts to the lower sea surface roughness in the IBL. The further away from the coast, the higher this IBL.

**Met opmerkingen [IW40]:** See reasoning above.

**Met opmerkingen [IW41]:** How significant are the differences that you find in figure 15.
Also, I do not think you should put all heights together. At 10m ASCAT is ASCAT, at other heights ASCAT is not ASCAT, but you have used ERA5 data for stability correction. So you can only compare spatial resolutions at 10 m height. Due to land contamination, ERA5 (31 km) will always lose from ASAT (12.5 km) in joined gridpoints closer than 31 km from the coast. At other heights than 10m you include ERA5 data, so affectively make the spatial resolution courser.
The only reason why one location might on average be more land-contaminated than another is prevailing wind direction (if wind blows mostly from sea to land, there is less land-contamination).
Please rewrite this (and the rest of this) section or leave it out. You use the grid box land fraction argument to say that ASCAT is better than ERA5 near the Finnish harbour (I get that) and ERA5 is better than ASCAT near the Swedish harbour ( that I think is wrong)???

**Met opmerkingen [IW42]:** Please look again at the discussion with all the remarks that I made earlier. I have not continued to check this section in detail (too much repetition of what has been said before), but my general remarks are: (1) This is not really a discussion, but more a conclusion (or summary?). Does this paper need a discussion section (discussion is included in the results section)? (2) Do not repeat (literary) what you have written earlier in the report and (3) Write clear sentences, Do not try to impress with long vague sentences.

**Met opmerkingen [IW43]:** Not checked. Needs to be adapted after revisions.

---

## Referee Report (RR3)

Review of "Ship-based lidar measurements for validating ASCAT-derived and ERA5 offshore wind profiles" by Hugo Rubio, Daniel Hatfield, Charlotte Bay Hasager, Martin Kühn, and Julia Gottschall

**General comments**

The authors conducted a comprehensive comparison of wind profiles derived 5 from the Advanced Scatterometer (ASCAT) satellite observations and the ERA5 reanalysis dataset against ship-based lidar measurements in the Northern Baltic Sea. Their analysis revealed a consistent underestimation in ERA5 profiles and an overestimation in ASCAT profiles when compared to ship-based lidar measurements. Additionally, they reported that the accuracy of both datasets deteriorates with height, a trend that is particularly pronounced in ASCAT profiles due to the limitations of the extrapolating methodology. The study is well conducted, and the methods used are appropriate. In particular, the use of ship-based lidar measurements to validate satellite-based and reanalysis datasets is a valuable approach, as their accuracy can be dependent on the distance from the coast.

These findings will be of interest to researchers in wind energy meteorology, as well as to engineers assessing offshore wind energy resources. However, the manuscript still has room for improvement before publication, as it lacks a positive message highlighting the significance of their findings through the manuscript. I have following concerns regarding the manuscript:

- Page 5, Figure 2: Although a height of 100 m is used as the reference height in this study, the hub height of recent wind turbines often exceeds 100 m. It would be helpful to explain why you chose 100 m as the reference height.

- Page 20, Figure 20: Since the ERA5 profiles above the surface layer are calculated using the PBL scheme, they exhibited more natural profiles than ASCAT-based dataset profiles as shown in Figure 12. Rather than using surface parameters from ERA5 for the vertical extrapolation of ASCAT, a simpler and potentially better approach might be to combine the surface wind field from ASCAT with wind profiles from ERA5.

- Additionally, the abstract currently concludes by highlighting the issues with the extrapolation method for ASCAT surface wind speeds. However, it would be more effective to end on a positive description, emphasizing the advantages of ASCAT-based offshore wind resource assessment.

- Page 25, Line 528–533: It is stated that there is a negative impact up to 40 km from the coast, but this seems rather extensive given the ASCAT data resolution of 12.5 km. This could lead to the impression that this dataset is unusable in near-shore areas. Considering the potential application of ASCAT data, it might be beneficial to examine this issue more carefully.

- Page 25, Line 524–540: It would be beneficial to describe the relationship between the

validation results of ASCAT and ERA5 in this study and previous research. Additionally, instead of listing numerical values in the main text, presenting them in tables would make the information clearer and easier to understand.

**Minor comments**

- Page 5, Line 103–104: In Section 2.1, it would be beneficial to include information on the accuracy of the ship-based LiDAR, particularly regarding whether there is any difference in accuracy compared to fixed LiDAR systems.
- Page 5, Figure 2: It would be more informative if Figure 2 were improved by using a geo-bubble chart or other visualization methods to plot the data on the map.
- Page 6, Line 138–149: It would be helpful to include a comparison table of ASCAT and ERA5 specifications in the Data and Methods section (Section 2).
- Page 15, Figures 7 and 9: In Figure 7, points A–F are indicated, but longitude is used in the figure's labeling. It would be better to ensure consistency in notation.

---

## Author Response (AR2)

**"Ship-based lidar measurements for validating ASCAT-derived and ERA5 offshore wind profiles"**
*Rev v2*
Hugo Rubio, Daniel Hatfield, Charlotte Bay Hasager, Martin Kühn, and Julia Gottschall
**Authors response to reviewer comments**

We would like to thank the referees once more for their time and effort in reviewing our work. We appreciate their feedback and comments. We have carefully considered their recommendations and concrete suggestions to enhance and clarify our work.

Below, we addressed the additional referees' comments and reply to them point by point. First, the referee's comment is included in italics and bold font, followed by our answer, and when applicable, the new excerpt of the revised version of the manuscript (highlighted in blue).

**Anonymous Referee, Referee #1**

**Referee #1 general comments**

1) ***First of all, it is still difficult to understand why the grid points with land contamination are included. To remove them would make the rest of the results much more useful. But it has helped that the results are separated into coastal and offshore areas.***
We decided to include all grid points, including those with land contamination, to provide a more comprehensive view of how both ERA5 and ASCAT perform in different regions of the study area, especially when compared against the reference ship-based lidar measurements. One of the primary advantages of using non-stationary, ship-borne lidar measurements is that they allow for validation of dataset performance across a variety of regions. By including these points, we aim to show the performance variability both nearshore and offshore, offering a complete picture of the datasets under test accuracy in different environments.

However, we agree that isolating the effect of land contamination is important to ensure clearer conclusions, particularly since the statistics can differ significantly between coastal and offshore regions. In response to this concern, we made an effort to modify several plots during the previous review round, which helped differentiate the performance of ASCAT and ERA5 in both areas more distinctly. We have also addressed these differences in the results and conclusions. As an additional effort in this direction, we have updated Fig. 13 to exclude nearshore grid points (within 20 km of the coastline), removing their influence from the overall results. The associated text and discussion have been adjusted accordingly to reflect these changes.

We believe that for the remaining figures, further differentiation is not required, as the effect of coastal grid points is already effectively separated. Additionally, we want to emphasize the challenge of clearly defining a threshold for nearshore versus offshore regions in this particular study area. The region is densely populated with small islets,

which, while not always considered true "land" regions, may still impact the performance of both ASCAT and ERA5.

2) ***The conclusion has improved, but it could still be more balanced on the usefulness of the approach. Secondly, it is true that satellite data are valuable measurements as an independent data source. However, when they are mixed with model data and the result, as shown here, is less precise than the model, then it is difficult to see the benefit. It is relatively easy to correct the consistent bias in the ERA5 profile, while the ASCAT-extrapolated wind profiles seem to add noise.***
We have further revised the conclusion to emphasize the main findings of this study, as well as to clarify the potential applications and limitations of the methodology used.

We also want to highlight that drawing conclusions such as "… when they are mixed with model data and the result, as shown here, is less precise than the model…" requires results supporting them. The results presented on this study quantify the performance of ASCAT-derived and ERA5 wind profiles, thus providing insights into the limitations and potential of these datasets and proving the usefulness of the study.

Furthermore, we do not completely agree with the assertion that ASCAT adds only noise, as the results show that both ASCAT and ERA5 exhibit comparable agreement with ship-based lidar measurements, although depending on the location and height under consideration. In fact, the inclusion of ASCAT data may offer a valuable supplementary dataset in cases where model-based outputs alone do not suffice for offshore wind characterization. The uncertainties inherent in numerical models can be mitigated by using multiple datasets, such as combining extrapolated ASCAT and ERA5, to examine the variability of the estimated mean wind speeds across different sources, and therefore helping to, for instance, quantify the risk of wind farm development prior to any available in-situ dataset. The results of this study also provide clear guidance on the limitations to be considered when using ASCAT data, enabling a more informed application of this dataset in future studies.

**The following numbering follows the numbering used by the referee in their report, which follow the first author's response (submitted on 14/05/2024):**

1  ***Regarding comparison of vertical wind profiles from ERA5, lidar and ASCAT-extrapolation: While comparing with ERA5 is not intended as a validation, there is still a discussion on what is more correct against lidar data in line 430 and 460 and onward which I find problematic in the form it is now, through the whole paper. "Overestimate", "underestimate" and "outperform" implies that a "truth" is known which is not the case in all places where these formulations are used.***
The manuscript has been reviewed to reformulate this wording, and maintaining terms such as "overestimate", "underestimate" and "outperform" in comparisons involving the lidar reference. In fact, we consider the ship lidar data as reference (as they come with an – in principle – well defined uncertainty).

***Also: line 12: "...ERA5 displays a consistent bias of approximately 0.5 m s-1 along the profile, whereas ASCAT exhibits a smaller bias within the lower 200 m of the profile." please add a comment about the variability, as this is quite important.***

The abstract in general, and this part in particular have been revised.

*line 169: Please add: "It should be noted that ASCAT winds are used in data assimilation in ERA5." or something similar.*
Included in line 184 of the new version of the manuscript.

**5** **Separating onshore and offshore in Figure 11 is helpful. But rather than more text on problems of ERA5 in a complex coastal site it would be nice to make sure that land contamination is avoided altogether. The justification from the authors of including the coast-affected data is that some of the points "show a significant agreement". Is that a good argument when the mix of surfaces within the scatterometer footprint may add up to values that are in the range of backscatter from water? This way it is not known what is measured and thus we don't know why there is an agreement.**
We have addressed some of the concerns regarding coastal contamination in our response to Comment 1, and we fully acknowledge the referee's point on this challenge.

While we agree that coastal contamination can introduce uncertainty, the aim of our study is to compare the performance of ERA5 and ASCAT under different environments. Leveraging non-stationary ship-based lidar measurements allows us to assess performance both in far-from-shore and nearshore regions, which is a unique strength of this dataset. However, fully investigating the underlying reasons for ASCAT's limitations in coastal regions falls outside the scope of this paper, as such an investigation would require a much more in-depth study into the generation and processing of ASCAT wind fields.

We recognize that removing points potentially affected by coastal contamination would eliminate the discussion of these issues altogether. However, we believe that doing so would not substantially affect the primary discussion of far-from-shore regions, as we have clearly differentiated between coastal and offshore effects in our analysis. Additionally, discussing coastal contamination does not overshadow the other key findings of the study. On the contrary, we believe that highlighting these limitations contributes to a more comprehensive understanding of the dataset's capabilities and shortcomings in various environments.

*Also: line 169: "By applying the IQR outlier detection, the impact of coastal contamination on the wind speed data is mitigated by removing unusually high wind speed retrievals in nearshore areas." -> "By applying the IQR outlier detection, the impact of coastal contamination on the wind speed data is mitigated." Only removing the high values through filtering may not remove all land effects. Is there a land flag in the ASCAT data set and did you use it? Please add information in the text about this.*
Sentence rewritten for clarity and further context added on this regard.

*line 369: "The reason for this is that, despite the filtering process for the ASCAT dataset, the coastal contamination still affects ASCAT measurements,..." and "The stronger impact of coastal contamination at 100 m can be attributed to the inaccurate characterization of stability conditions by ERA5 in nearshore locations due to its coarse horizontal resolution and limited ability to resolve fine-scale atmospheric features in these regions.": In my view the problem of ERA5 in coastal areas is a secondary problem if ASCAT is measuring some land instead of only wind. Please remove the last sentence.*

We appreciate the reviewer's comment and agree that the primary cause of the stronger deviations in coastal areas is due to ASCAT's coastal contamination. The intent of this part of the manuscript was not to compare the overall differences between ASCAT and ERA5, but rather to explain why these differences are more pronounced at 100 m compared to 10 m. That said, we understand the referee's concern, and a more detailed discussion of the potential reasons for the variation in ASCAT and ERA5 differences with height is provided a few lines below (lines 390–394). Therefore, as suggested, we have removed the second sentence to avoid confusion and to highlight the primary issue.

19 **Line 286: From first review: "Please comment on the fact that there is land in two of the grid boxes and argue why they can be included in the analysis."**
**Authors response: "A comment regarding this fact has been added in lines 312-315 of the new version of the manuscript."**
**This review: Please add the comment in the text when describing figure 5, e.g in line 297, and explain why you choose to keep them (see also point 5 above).**
We added a comment on this specifically referring to Figure 5 and Figure 6. However, we did refer to the wrong lines. This is within lines 308 and 312: "Furthermore, Figs. 5 and 6 show that the surface of certain ASCAT grid boxes, particularly those at or near the two harbours, is partially covered by land. This situation may lead to coastal contamination and excessively high wind speed retrievals within these grid boxes. The influence of this effect is discussed in the results section of this study, where its potential impact on the presented findings becomes apparent."

**Other comments:**

A. *line 308-316: This part is a description of what is coming and can be removed or moved to the section at the end of the introduction.*
We acknowledge the referee's comment. While we have included a concise summary of the manuscript's structure at the end of the introduction, we believe that the additional details provided in these lines enhance the reader's understanding of the distinct subsections within the Results section. This text serves to clarify the specific analyses and comparisons that follow, offering essential context for the findings presented. Furthermore, given its brevity, we believe that this excerpt does not significantly lengthen the manuscript, yet it does provide valuable guidance for the reader navigating through the results.

B. *Figure 8: please specify what is subtracted from what in the figure caption.*
Clarification added.

C. *line 330: How can we conclude that the collocated approach is underestimating? Should it just be "The lower values associated with …"?*
Yes, this has been corrected according to referee comment.

D. *line 370: "The stronger impact of coastal contamination…" -> "An additional source of uncertainty leading to stronger impact of the coastal contamination…"*
As suggested by the referee in comment 5, the sentence referred to in this comment has been removed from the manuscript.

E. *line 374-383: This paragraph is a mix of results and discussion. It would be better to move it to chapter 4.*
In this paragraph, we highlight the key conclusion derived from results presented in Figure 10, and as suggested by Referee #2, briefly compare these findings with previous literature. We believe this brief discussion helps interpret the results and fits well within the flow of this section.

F. *line 423: In my opinion, "Excellent" is a too strong word for this visual comparison. Please modify, fex: "For the remaining locations, both datasets demonstrate comparable agreement with the lidar wind profile."*
Modified according to referee suggestion.

G. *The paragraph starting at line 475: The machine-learning approach was already mentioned in the section 2.4, there is no need to repeat it here since it was not applied.*
ML approaches are mentioned twice in the manuscript—once in Section 2.4 and once in the Discussion (as referred to by the reviewer). However, the context and focus of these mentions differ. In Section 2.4, the mention is brief and serves as part of a general review of satellite wind extrapolation methods used in prior literature. In contrast, the mention in the Discussion highlights the potential advantages of ML approaches, suggesting that they may offer improved results over the mean stability correction method applied in this study. However, we also clarify that ML approaches are not feasible in this case due to the limited data available for training such models.
Given that both mentions address distinct aspects of ML methods—one as part of a broader literature review, and the other as a reflection on the study's limitations and potential—we believe that both references are relevant and valuable to the manuscript. Therefore, we kindly suggest retaining this mention in the Discussion.

H. *line 494: Please replace "observed region" with "open ocean" or "far from the coast" in the sentence: "The comparison between ASCAT and ERA5 winds reveals an overall good agreement when assessing the mean wind speed values across the observed region of the Baltic Sea".*
Done.

I. *line: 210: Here it is stated that the statistical MOST (applied in the paper) can be applied up to turbine operating heights, but in line 415 it is said that the heights above 200m are well beyond the range of applicability. Please write more clearly what range the method should be valid in.*
To the authors' knowledge, there is no explicitly defined height limit for the applicability of the extrapolation methodology used in this study. However, as explained in the paper, it is expected to perform better than the instantaneous MOST at heights above the surface layer. There is currently no literature that defines or investigates a specific height for this methodology. In fact, this study represents the first comparison of ASCAT-extrapolated profiles using this methodology against in situ measurements at heights above 200 m.

Given that most wind turbines operate below the 200 m threshold, we believe that stating the methodology is applicable "up to turbine operating heights" is effectively very similar as saying "up to below 200 m.". This aligns with the results presented, which show that extrapolated ASCAT profiles demonstrate a decreased performance above 170 m, and this

is mentioned in the discussion presented results. However, establishing a more precise and definitive threshold for the validity of this methodology would require additional analyses.

*J.* ***There are typos throughout the manuscript.***
The full manuscript has been reviewed to correct typos and enhance its clarity and readiness.

Before addressing Referee´s #2 comments, we would like to note that we believe Referee #2 may have reviewed the wrong version of the manuscript in this revision round. This is evidenced by comments referring to content that was either altered or removed in the latest version. Below are some specific examples:

- **Line 31**: Referee 2 suggested removing the word "therefore." However, this paragraph was removed entirely during the previous (first) review round. The comment corresponds to Line 31 in the first submitted version of the manuscript, where the word "therefore" was present.
- **Line 241**: Referee 2's comment discusses the "C± constants" being set to 6 and 4 for stable and unstable portions, respectively. This text is found in the first submitted version of the manuscript, but the corresponding section was revised in the second version (Line 241) submitted for review.
- **Figure 6**: Referee 2 mentions "Six locations used for the comparison of the datasets." This refers to Figure 6 in the first submitted version. After a new figure was added during the first reviewing round, this content now corresponds to Figure 7 in the updated version.

These examples suggest that Referee 2 may have reviewed the first submitted version of the manuscript, which is inconsistent with the version we intended for that review. As a result, some comments may no longer be relevant or applicable.

In response to such comments, we have highlighted where they may no longer apply to the current version of the manuscript or have already been addressed. However, most comments are still relevant, and we have worked to address them in the newly revised manuscript submitted after the second (current) review round to improve the paper.

**Referee #2 general comments**

1) *There are a few things that I think need to be addressed in the paper:*
   a. *Uncertainty in the lidar measurements. are the differences that you find with ERA5 and/or modified ASCAT significant? See e.g. page 14: [TNO report - DOWA validation against offshore mast and LiDAR measurements | Report | Dutch Offshore Wind Atlas](TNO report - DOWA validation against offshore mast and LiDAR measurements | Report | Dutch Offshore Wind Atlas)*
      This was already addressed during the first review round, in which we added some information on this regard in the Discussion section: *"…*Additionally, it is acknowledged that lidar measurements, like any other observational data, are subject to inherent uncertainties that may impact the results (Duncanet al., 2019b; Rubio and Gottschall, 2022). Nevertheless, the observed deviations between the lidar measurements and both extrapolated ASCAT and ERA5 significantly exceed the magnitude of potential discrepancies attributable to floating lidar uncertainties, which can be up to approximately 2 % with mast-mounted anemometers as lower limit reference (Wolken-Mohlmann et al., 2022)*".*

   b. *Your method is not robust with more/larger WFs (there are no wind farms yet in the Baltic according to [https://map.4coffshore.com/offshorewind/](https://map.4coffshore.com/offshorewind/),*

*but you expect significant growth). Ship-based lidar measurements may be affected by wind farms (WF), ERA5 definitely does not take WF effects into account and ASCAT is too coarse to measure WF effects (at least in detail: then you need SAR).*

Certainly, wind farm wake losses are a relevant consideration when investigating the potential energy yield of a site. However, as noted in our previous response to the referee, the primary focus of this study is not on evaluating the potential of wind farms, but rather on validating ASCAT and ERA5 datasets against reference lidar measurements, with special attention to wind speeds at turbine operating heights, providing insights regarding their potential and limitations in characterizing offshore winds.

While, in some parts of the paper (in the Discussion), we mention the potential application of paper's findings to wind resource assessment, this is not an objective of the paper. Instead, we believe the validation of these datasets helps to clarify their limitations and accuracy, offering insight into the contexts in which they can be effectively used or not. Although they are not usable for highly detailed and advanced wind resource assessments, these datasets are still useful for other applications, such as large-scale planning of wind (or energy) potential or preliminary site screening studies helping to identify regions with promising resources, particularly in areas with limited or no in-situ measurements. It is in these cases that we believe, these datasets can be used, and where the methodology and findings presented in our study are relevant, meaningful and applicable.

Lastly, as the referee correctly pointed out, wind farm wake losses do not influence the results presented here. First, no nearby wind farms currently exist in the study area to induce wake effects. Second, ERA5 does not account for wake losses of any type, and neither ERA5 nor ASCAT have the spatial resolution required to reliably capture internal wind farm wake effects.

However, in order to address referee's concern, we have revised the Conclusions section to clarify potential applications of the finding from this study and to highlight the limitations of ERA5 and ASCAT for characterizing wake effects: "…This methodology is particularly beneficial in scenarios where more complex extrapolation methods are impractical or when in situ measurements are limited, providing an additional source of wind information and thereby improving the reliability of offshore wind characterization studies. However, the application of both ERA5 and ASCAT must be approached with caution due to their inherent characteristics, including insufficient spatial resolution and the inability to adequately capture wind farm wake effects, which limit their utility for detailed wind farm energy yield assessments. Despite this, these datasets are still valuable for other applications, such as large-scale planning of wind potential or preliminary site screening studies, helping to identify regions with promising resources. In such cases, the findings of this study provide valuable insights into the conditions under which these datasets and methodology can be applied and the level of reliability that can be expected. Nonetheless, it is crucial to also acknowledge the primary

limitations of this approach, such as excessive wind speed deviations in nearshore locations and the increased expected error at higher altitudes."

c. **How does your method compare to assimilating ASCAT into the NWP reanalysis like it was done in DOWA (point 1 in** [Innovations in the DOWA project | DOWA project | Dutch Offshore Wind Atlas](...))**?**

Data assimilation methodologies are out of the scope of this paper and therefore, these methods have not been investigated or discussed in this work.

d. **You basically show in your paper that ASCAT and ERA5 should not be used closer than 40 km from the coast (validation results based on ship-based lidar). That is a conclusion that I miss in your paper. As far as I know the ASCAT coastal product is only valid 15 km away from the coast and ERA5 has problems with abrupt changes in surface roughness, such as on the coast. A model (such as ERA5) assumes a grid box average surface roughness for a combination of land and water whereas the wind feels land or water. The larger the grid box size, the larger the problem (ERA5 grid box size 31 km). So basically ASCAT and ERA5 have quality issues near the coast and this is what you find confirmed in your paper.**

We acknowledge the referee's reflection. However, a key conclusion we draw from the study is that gridded datasets, like ASCAT and ERA5, are not really comparable to microscale measurements systems, such as, in this case, a floating lidar system. Consequently, the application of these datasets requires caution, especially in nearshore areas. It is important to manage expectations regarding their accuracy and reliability in these regions. Additionally, we did not define a strict threshold of 40 km, as doing so is challenging with the available data. The study area is densely populated with small islands and islets, which, while not always classified as "land" regions, can still influence the performance of both ASCAT and ERA5.

In any case, the limitations of both ASCAT and ERA5 in nearshore regions have been highlighted throughout the entire manuscript, as this is a key conclusion shown by the presented results, and explicitly mentioned in the conclusion of the manuscript.

2) **Comments from earlier review that have not been addressed yet are:**

a. **Are there other measurements that you can compare to lidar measurements in harbor (where ASCAT and ERA5 are particularly inaccurate)?**

To the authors' knowledge, no additional measurements were available near the ship's track during the measurement campaign.

b. **Have you considered triple (or quadruple collocation) to assess uncertainties (there are also uncertainties in your lidar measurements! What are they)?**

Triple collocation requires a large amount of data to effectively derive the uncertainty levels of ship-based lidar measurements. Unfortunately, due to the relatively short duration of the campaign and the limited availability of

ASCAT data during this period, insufficient data was collected to implement triple collocation.

In addition, while the application of triple collocation is indeed an interesting approach for further investigating the uncertainties in lidar measurements, it falls outside the scope of this paper and would be more suitable for future research.

c. *Have you considered using other wind climatology's such as NEWA [GMD - The Making of the New European Wind Atlas – Part 2: Production and evaluation (copernicus.org)](...)?*

The temporal coverage of the NEWA does not align with the period of the ship-based lidar measurement campaign utilized in this study. Since our analysis and results are specific to this time period, incorporating NEWA would not have been applicable nor comparable. Therefore, we chose not to use it in this analysis.

**Referee #2 specific comments**

The referee has included additional comments in a bundled format within a PDF file. These "subcomments" are presented below each corresponding main comment in grey font, accompanied by a grey highlight in the main comment to indicate the specific text to which the subcomment refers, consistent with the referee's formatting in her reply.

3) *Line 3: typo: observations*
Corrected.

4) *Line 9/10: The comparison reveals a close agreement between ASCAT and ERA5 beyond 40 km distance from the coast. Unclear what you mean: close agreement between two different approaches (account for stability)? At 10m height or also extrapolated to hub heights?*
This part of the abstract has been modified for clarity, as requested by the referee.

5) *Line 10/11: (Extrapolated) ASCAT tends to significantly overestimate the mean wind speed derived from lidar measurements, while ERA5 exhibits a consistent underestimation. I assume the difference between lidar measurement and (Extrapolated)ASCAT/ERA5 is larger than the lidar measurement uncertainty?*
A comment regarding the uncertainty level expected for the floating ship-based lidar system has been included in the Discussion section of the paper.

6) *Line 21: However, in situ …*
Corrected.

7) *Line 26/27: Floating lidar systems can be moved to different locations, but generally measure at one location for a certain period of time. With profiling lidar systems installed on cruising ships it is possible to provide reliable wind profile measurements over larger areas.*
Manuscript modified according to referee suggestion.

8) *Line 27/30: (can be formulated shorter/clearer): Before profiling lidar systems on cruising ships can become a generally accepted alternative for offshore met masts and floating lidar, specific challenges have to be overcome such as validation against reference data and quantifying the associated uncertainty (Rubio and Gottschall, 2022). Still, ship based lidar has already been used in different wind energy related studies. In Wolken-Möhlmann…*

    i. You used industry standard, but industry standard for what? These measurements are only useful for validation of reanalyses which can be used wind resource assessments if the growing effect of wind farms is accounted for.

Manuscript modified according to referee suggestion.

9) *Line 34-38: However, while numerical models have demonstrated good performance in shallow-water offshore regions compared to in situ measurements (Witha et al., 2019b), they often fail to describe the spatial and temporal variability of wind with*

*sufficient accuracy and detail. I suggest an alternative text: Numerical weather prediction (NWP) models in re-analyses mode are commonly used … spatial coverage. However, while numerical models have demonstrated good performance in shallow-water offshore regions compared to in situ measurements (Witha et al., 2019b; Wijnant et al, 2019), they have problems with areas with large changes in surface roughness, such as the coast. The larger the grid box size, the larger the problem because the model assumes a grid box average surface roughness for a larger area (whereas the wind feels land or water, not a combination). Also most re-analyses do not take into account the (changing) effect of wind farms on the atmosphere (except: https://wins50.nl/*

    i.    '… they often fail to describe the spatial and temporal variability of wind with sufficient accuracy and detail' = very general conclusion which I do not agree with. The spatial and temporal variability is pretty well captured in the DOWA/WINS50 re-analyses (see validation section of KNMI Technical report - The Dutch Offshore Wind Atlas (DOWA): description of the dataset | Report | Dutch Offshore Wind Atlas ). Witha just used one weather model (WRF). Abbreviate 'Numerical weather prediction models' to NWP models (not numerical models).

    ii.    KNMI Technical report - The Dutch Offshore Wind Atlas (DOWA): description of the dataset | Report | Dutch Offshore Wind Atlas

Manuscript modified according to referee suggestion and reference suggested added.

10) *Line 38-41: This limitation arises from factors such as the inaccurate parameterization of the model variables or the insufficient temporal and spatial resolution of the models' output data. Furthermore, the lack of in situ measurements in deeper offshore regions hinders the validation of these datasets, leading to increased uncertainties in derived wind statistics for such locations. I suggest an alternative text: Each NWP model has its own limitations (caused e.g. by grid and domain size and physical modelling and parametrisation choices). This results in uncertainties in wind statistics based on these NWP models and these uncertainties can be quantified when validation measurements (incl. measurement uncertainties) are available. This is however often a problem for hub heights, especially for far-offshore locations with deep water.*

    i.    Again a bit over simplified and I do not agree with what you write. I do not think that you can say 'inaccurate parametrisation of model variables'. Choices in NWP models always are a trade-off (optimise which forecast lead-time? Optimise which parameters?).

Text modified according to referee suggestion.

11) *Line 42-44: To overcome the limitations of in situ measurements and numerical models, satellite remote sensing devices have emerged as a potential alternative for characterizing ocean winds and climate over large areas, capturing the wind variability with a temporal coverage of over 15 years. I suggest an alternative text: Scatterometer (wind) measurements from satellites are a welcome additional source of information in these data sparse areas. Several studies …*

    i.    I would leave this 'model/measurement limitations' out because ASCAT has its own limitations. Basically you have scatterometer data from satellites as additional information in data sparse areas (it is no replacement for models

or other measurements). Note that another way of using ASCAT is data-assimilation in the NWP model, like in DOWA/WINS50 (see **TNO report - DOWA validation against ASCAT satellite winds | Report | Dutch Offshore Wind Atlas** ).

The manuscript has been updated according to the referee's suggestion.

12) *Line 47-53: Fluffy writing: does not make it clearer and there are some mistakes in it. I suggest an alternative text: The ASCAT coastal product is available since 2007 and provides high quality offshore wind measurements* on a 12.5 km grid spacing for locations further than 15 km from the coast. The ASCAT wind speed bias is less than -0.23 ms-1 in coastal areas (15- 50 km from the coast) and -0.29 ms-1 elsewhere (*TNO report - DOWA validation against ASCAT satellite winds | Report | Dutch Offshore Wind Atlas* ). However, ASCAT has its limitations: only available twice a day (around 09:30 and 21:30 UTC) and stability dependent assumptions have to be made to derive turbine height winds from the ASCAT 10m winds.

    i.    Assume you used that?
    ii.    Better. Someone might otherwise read this in 10 years time and think ASCAT is available since 2016
    iii.    That is not the same as resolution!!! Ask Ad Stoffelen KNMI.
    iv.    If I am correct: please check
    v.    You write: 'Lastly, the trustworthiness of satellite retrievals remains a knowledge gap, due to the lack of available in situ datasets for validation especially in deep water regions'. I left this out because I think it is incorrect: ASCAT has been extensively validated (besides: its quality does not depend on water depth). Ask Ad Stoffelen KNMI.

As suggested by the referee, we have modified "12.5 km resolution" by "12.5 km grid spacing" across the manuscript. However, we have retained other details as originally presented since additional information regarding ASCAT—such as the year it became available, the specific product utilized in this study, and other relevant details—can be found in Section 2.2 of the manuscript. Furthermore, the text excerpt referenced by the referee in sub-comment v. was removed during the previous review round and was no longer present in the latest submitted version of the manuscript.

13) *Line 54: The Baltic Sea is an area of great interests for offshore wind development…*
    i.    And yet: you do not mention the effect of wind farms (WF) on the atmosphere. ERA5 is without WF effects, ASCAT is too course, at least for detail (you need SAR for that), but your ship based lidar may measure the effects up to 100 (?) km from a WF. I think you should at least mention WF effects in the paper and tell what the consequences of these WF effects are for your method.
    Can you quanlify what you mean with near shore (I assume > 15 km from shore otherwise ASCAT not valid)?

In response to comment 1b, we have already argument why we believe wake effects are not relevant in this study. First, datasets used are not suitable/capable of capturing wake effects. Secondly, due to the absence of nearby wind farms in the period considered in this study, their effects in the results and methodology presented are

none. The fact that there is interest in the development of wind farms in the Baltic Sea does not imply that our study needs to contemplate those. Especially, when at the time this study is realized, there are none. Furthermore, we believe that the inability to characterize wake effects is not a limitation of the methods applied in this study, but rather a limitation inherent to the nature of ERA5 and ASCAT themselves. Thus, such considerations are beyond the scope of this paper.

The quantification of nearshore regions is addressed in several sections of the paper, particularly during the discussion of the results. Specifically, we refer to two distinct thresholds (20 km and 40 km), as significant changes in the observed biases are noted when applying these distance values.

14) **Line 64-71: I suggest that you change sequence of what you write to make it clearer, e.g.** To derive wind profiles from the ASCAT coastal product 10 m measurements, we employ the long-term stability correction approach presented in Kelly and Gryning (2010) and implemented in Badger et al. (2016). For this, we utilize the stability information from ECMWF Reanalysis 5th generation (ERA5)and compare two different collocating methods to evaluate the potential influence of the limited temporal resolution of satellite overpasses in the ASCAT extrapolated profiles. Not only the ASCAT derived wind profiles, but also the wind profiles from ERA5 are then compared to the lidar profiles.
Manuscript modified according to referee suggestion.

15) **Line 75-76: … of the reliability and accuracy of satellite measurements derived wind statistics for offshore wind characterization at wind energy relevant heights.**
Corrected according to referee comment.

16) **Line 86: What is the accuracy of your lidar measurements? If you want to compare your measurements to model data, you will have to be able to tell whether the difference that you find is significant (outside the measurement uncertainty). See e.g. TNO report - DOWA validation against offshore mast and LiDAR measurements | Report | Dutch Offshore Wind Atlas**
This was already addressed during the first review round, including a citation to the suggested reference:
"Consequently, the mean values derived from lidar measurements may exhibit biases that vary depending on the time slots during which measurements were acquired at particular locations. Additionally, it is acknowledged that lidar measurements, like any other observational data, are subject to inherent uncertainties that may impact the results (Duncanet al., 2019b; Rubio and Gottschall, 2022). Nevertheless, the observed deviations between the lidar measurements and both extrapolated ASCAT and ERA5 significantly exceed the magnitude of potential discrepancies attributable to floating lidar uncertainties, which can be up to approximately 2 % with mast-mounted anemometers as lower limit reference (Wolken-Mohlmann et al., 2022)".

17) **Line 104-105: the motion (take the s out) effects**
Corrected.

18) **Line 117: fig 2b is the daily cycle the ship (lidar) experiences because it is connected to the location of the ship. It is not how the wind depends on the hour in the day (which is what normally is meant by 'daily cycle'). Maybe use a different name to avoid confusion (wind speed daily cycle plots normally give highest wind speeds during the day), e.g. Wind speed ship daily cycle.**

Corrected as suggested by referee using "Wind speed ship daily cycle", both in figure caption and main text.

19) **Line 31: Therefore (?), the ….**

The paragraph containing the "Therefore" mentioned by the referee was removed from the manuscript during the previous review round, as we felt it would be more appropriately placed in the conclusion or discussion section.

20) **Line 155/156: … available horizontal grid spacings of 12.5 km and 25 km**
    i. **Not the same as resolution**

Corrected according to referee comment.

21) **Line 159: what do you mean by Both of these (?) are implemented (?) at…**

We are referring to the two services mentioned in the sentence before. This has been clarified in the manuscript.

22) **Line 168-170: By applying the IQR outlier detection, the impact of coastal contamination on the wind speed data is minimized, leading to more accurate and reliable results in nearshore areas.**
    i. **I assume this is part of the ASCAT coastal product? Is nearshore more than 15 km from the coast?**

The IQR outlier detection was specifically implemented in our study to filter out 10m ASCAT mean wind speeds with values excessively high near the coast. The relevant section of the manuscript has been revised to clarify this.

23) **Line 189: Several methodologies for vertical extrapolation of satellite measurements …**

Corrected according to referee comment.

24) **Line 260-264: As observed, considering the stability information from the full campaign results in a better theoretical distribution compared to the collocated approach. Although the difference is minimal at the harbor site, it is more pronounced at the offshore location, where a significant underestimation of unstable stability occurrence is observed. The harbor site presents a rather symmetric distribution around zero, meaning that both unstable and stable atmospheric conditions are equally represented. However, the offshore site exhibits a higher occurrence of unstable conditions, compared to the stable side of the curve.**
    i. **There are a few things to consider here: (1) at the harbors you have on average 'land behaviour' of stability which means a daily cycle with more stable in the night and more unstable during the day; offshore there is no such daily cycle (stability depends more on the season); (2) I assume that in your collocated harbor graphs you look at around 09:30 UTC and in your collocated offshore graphs at around 21:30 UTC. I assume for the whole**

measuring campaign you looked at day and night for the period 28 June 2022 to 21 February 2023?

Harbor: in the collocated set (around 09:30 UTC), you would expect unstable to be slightly underrepresented if it is day time, but you see the opposite (more unstable than the theoretical line which is derived for all stability classes). A lot depends on what is really day time off course (in winter well after 07:00). If you take the whole dataset you have apparently a higher percentage stable (?) and a better match with the theoretical line.

Offshore: there is no daily cycle effect on stability here. The only thing that affects stability is that you miss 3 months of the year where the sea is particularly cold (but that is the case for both the collocated and whole dataset). So that probably explains the overestimation of unstable. But why do we only see that for collocated, not for the whole campaign?

Section 3.1 of the manuscript was modified during the first reviewing round in order to better explain how the stability daily cycle may influence the differences between the collocated and full campaign approaches. Therefore, we would like to refer the referee to the latest version of the manuscript, since this issue is properly discussed there.

**25) Line 192-193: …. performance at different vertical and horizontal constraints.**

    i.    **What do you mean?**

This sentence was removed from the manuscript during the first review round.

**26) Figure 6. Six locations used for the comparison of the datasets. The approximate distance to the nearest shore is indicated, in km, below of each site. Please add: Location A is the harbour of Nynäshamn (Sweden) and location D the harbour of Hanko (Finland).**

We assume the referee is referring to Figure 7 of the new version of the manuscript, since Figure 6 was added after the first reviewing round and shows number of lidar samples recorded at each ASCAT grid cell.

The suggested text has been added to Figure 7 caption.

**27) Line 241: In this study, the values for the C± constants have been set to 6 and 4 for the stable and unstable portions, respectively.**

    i.    **So basically the stability correction has only 2 values for C which are the same for the whole Baltic Sea and all heights, one for stable and one for unstable: correct? Did you consider other values, like the ones from literature? Can you show why these are the best values? Does it not depend on the season which values are…**

Correct, as specified in the manuscript: "…C± constants have been set to 6 and 4 for the stable and unstable portions, respectively. … identical values of C± were applied to all ASCAT grid points."

As an example, the comparison between the ERA5 NPD of 1/L and the theoretical distribution obtained with values of C+=5 and C-=12 (used in (Optis et al. 2021)) is shown below. As can be observed, even with the inclusion of ERA5 data from the full campaign, the theoretical distribution struggles to closely resemble the NPD of the 1/L parameter calculated by ERA5.

[Figure]

*Figure 1: Normalized probability density function of 1/L from ERA5 and theoretical distributions calculated from Eq. (2) using 5 and 12 for C+ and C-, respectively. The same offshore location as in Figure 4d and full campaign data was considered for this example.*

**28) Line 307-311: _First, the coastal contamination of near shore areas leads to the removal of some ASCAT overpasses for data quality reasons, leading to a reduced number of ASCAT observations in these areas. Consequently, the insufficient number of valid wind speed measurements obtained from the collocated approach introduces a biased representation of the prevailing stability conditions during the campaign period._**

    i.    ASCAT and ERA5 both have problems in coastal areas (see general remarks). So the uncertainty in the wind data that you use in fig 7 is large in these areas (larger than further offshore). Uncertainties in ERA5 are probably larger near the Swedish coast with prevailing W-SW winds. Insufficient number of measurements does not necessarily have to lead to a biased...

We would like to refer the referee to the new amendments made in Section 3.1 during the first reviewing round.

**29) Line 313-315 (from previous review): _'pronounced instability in the morning?' Why would ERA5 produce stronger unstable conditions (lower 1/L) in the morning at Nynashamn? What do we know about the water temperature near Nynashamn and how it is modelled by ERA5 (shallower/warmer water between Bedaron and the mainland maybe?)? ERA5 has grid boxes of 31 km2 so model values are probably very land-contaminated in that area: can you make a plot of the ERA5 grid boxes near the harbours? What is the prevailing wind direction? Basically ERA5 and ASCAT are not very good in coastal area: maybe you should take them out of your analyses?_**

As in the previous comment, we would like to refer the referee to the amendments made in Section 3.1 during the first review round. These revisions now explicitly

discuss the daily stability cycle at the different locations considered and highlight the significant role that the ERA5 land mask (or land contamination) plays in this context.

30) **Line 315-316: *This results in a lower wind speed compared to the full campaign approach, as can be derived from Eq. 4.***

   i. So what you say is that the wind at 100m height is lower in more unstable conditions? If that is what you mean, it is wrong.

This sentence has been modified for clarity and precision.

31) **Line 316-317: *In contrast, the other locations do not exhibit such pronounced daily stability cycles, and therefore, smaller differences are reported between the two approaches.***

   i. I explained why this is the case for offshore. For the Finnish harbour the prevailing W-SW'ly wind is the reason for a reduced daily stability cycle. Please add the why to your paper.

We would like to refer the referee to the new amendments made in Section 3.1 during the first reviewing round.

32) **Line 317-320: *Finally, as mentioned in Section 2.4, the same values of the semi-empirical constant C± are assumed for the entire region, instead of using a site-specific definition of these constants. Therefore, the suitability of the selected values may not be optimal for certain locations, leading to an anomalous theoretical representation of the empirical atmospheric distribution.***

   i. I think your conclusion should be different. You can conclude that your method works well for offshore, but not near the coast (so not for the entire ship track) because of poor quality ASCAT and ERA5 near the coast (less than 31 km from the coast actually).

This section was modified during the first reviewing round, so we would like to refer the referee to the new section. In the new version of the manuscript the text excerpt is not used as any conclusion, but as a reason that potentially contributes to explaining the differences between the collocated and full campaign approach.

33) **Figure 9 basically shows you that ASCAT winds look unrealistic near the coast at 10 and (more so) at 100m. Especially near the Swedish coast where the wind blows predominantly from land to sea, wind near the coast should be lower than further offshore. So this figure proves that you cannot use your method near the coast for 2 reasons: (1) quality of ASCAT, (2) grid size of ERA5 (averages surface roughnesses of land and sea in grid box, therefore wrong for both wind from land and from sea). Small scale effects such as sea breeze and low level jets (you mention these in line 341) don't have a significant effect on your mean values.**
The limitations of both ASCAT and ERA5 in coastal regions due to coastal contamination and coarse grid resolution are now explicitly addressed in the manuscript: "The effect of coastal contamination in the ASCAT map is particularly visible in the 100 m height map, where the highest mean wind speeds are located along the perimeter of the region with available data. An additional source of uncertainty contributing to the stronger impact of the coastal contamination at 100 m is the inaccurate characterization of stability conditions by ERA5 in nearshore locations

due to its coarse horizontal resolution and limited ability to resolve fine-scale atmospheric features in these regions."

In response to the comment on sea breezes and low-level jets, these are now only mentioned in the introduction (line 62).

Regarding the statement that "*this figure proves that you cannot use your method near the coast*", it is important to clarify that this limitation stems from inherent characteristics of the ASCAT and ERA5 datasets, rather than from the methodology employed in this study.

**34) Line 341-342 (fig 9 10m validation): (from previous review) compare to Validation of DOWA ('undisturbed wind' = HARMONIE without WFP) with ASCAT (too coarse to measure wind farm effects) at 10 m height: TNO report - DOWA validation against ASCAT satellite winds | Report | Dutch Offshore Wind Atlas. Because you use ERA5 stability info to calculate ASCAT-derived wind speeds at 100m height, the difference you see at 100m should mainly be because of differences at 10m, right?**

In the first review round, a comparison of the results presented in Figure 10 (previously Figure 9 in the initially submitted version of the manuscript) with the reference provided by the referee was already included, and the corresponding reference was incorporated into the manuscript at that time.

The fact that biases between the two datasets at 10 m and 100 m differ are basically explained by three main factors, which have been now added in the manuscript: "The differing biases between these two datasets at the two heights levels (10 m and 100 m) can be attributed to three key factors: first, the inherent difference between the datasets at 10m, second, the mean stability correction approach used to extrapolate ASCAT; and finally, as illustrated in Figure 8, the impact of the collocation strategy applied for the theoretical stability characterization".

**35) Line 342: Figure 10a illustrates the _difference_ in wind speed between ASCAT and ERA5 at 10 m and 100 m**

   i.      Wrong use of the word 'disparity' (nothing unfair about this difference).

Corrected.

**36) Lines 347-350: This discrepancy in the nearshore areas can be explained by the combination of too high wind speeds retrieved by ASCAT due to coastal contamination and ERA5's inability to properly resolve the coastal atmospheric phenomena and its coarse horizontal resolution that leads to the omission of the flow phenomena variations caused by the small islands present in these coastal regions.**

   i.      It has nothing to do with coastal atmospheric phenomena or flow phenomena variations (do you mean sea breezes?). It has everything to do with 'land roughness contamination' of the roughness in the coastal grid cells

Sentence rewritten for clarity: "…ERA5's inability to properly resolve the coastal atmospheric phenomena and small-scale wind flow variations due to its coarse horizontal resolution."

**37) Figure 10 shows you that you should not use your method within about 40 km from the coast (you should expect 31 km because of the grid size of ERA5 and what I explained earlier)**

Results presented in Figure 11 (Figure 10 in first submitted version of the manuscript) show higher discrepancies between ERA5 and ASCAT at both 10m and 100m (within these 40km distance to shore). However, since the extrapolation methodology used in this study does not affect the data at 10m, we cannot conclude that the method itself should not be used within 40 km from the coast. Rather, we believe a more accurate conclusion is that, within this region, higher uncertainty is expected in both ERA5 and ASCAT values, as evidenced by the larger differences observed due to the limitations of these datasets (e.g. ERA5 grid size as mentioned by referee). Therefore, we want to highlight that this is not due to a limitation of the method, but a limitation of these datasets. This has been discussed in the manuscript.

While the extrapolation may contribute to some additional uncertainty, as seen by the consistently larger bias at 100m compared to 10m (also explicitly mentioned in the manuscript), the key limitations regarding the applicability closer or further away from the shore lie in the datasets themselves, not in the methodology employed in this study.

**38) Line 355-356: Nonetheless, the majority of grid points exhibit wind speed differences below ±1 m s-1. As previously discussed, wind speed differences above this threshold correspond to those of near-shore grid points.**

    i.    This big difference of 1 m/s in mean values is not the bias, but the max difference, right?

This sentence was revised during the first review round.

To clarify, each point in Figure 11a represents a grid point within the study area, and the y-axis shows the difference between the wind speed obtained from ASCAT and ERA5 (i.e., ASCAT minus ERA5). This difference represents the bias between the two datasets, not the maximum difference.

As a reminder, as explained in Section 2.4, the extrapolation methodology used only provides mean wind speed values. Thus, for each ASCAT grid point, we have a mean wind speed for the entire study period (the duration of the campaign). These mean values are then compared against the corresponding mean wind speeds provided by ERA5, with one mean value per grid point as well. Consequently, the figure illustrates the bias in mean wind speeds between the two datasets, rather than instantaneous (or max or min) differences.

**39) Line 400: what do you mean with the word 'trend 'here? The word trend is used for change in time (e.g. climate change), but this is not what you mean…**

The second part of the sentence has been omitted, since it does not add any meaningful information: This observation holds true for all three presented elevation levels. Notably, the western area…

**40) Line 400-403: Notably, the western area of the ship route (longitude below 18.5 degrees) exhibits the largest errors for both ASCAT-derived winds (using ERA5) and ERA5 winds, with maximum differences exceeding 3 m s-1 at all elevation levels. This indicates that wind speed estimation cannot be done accurately enough in these**

*areas* with *ASCAT and/or ERA5 because (1) poor quality of ASCAT coastal product closer than 15 km from the coast and (2) ERA5 grid box size (surface roughness in land-water grid boxes on the coast problematic).*

    i.     **Is it possible to add distance to the nearest coast to fig 13? In this figure we are looking at winds at 60m, 150m and 220 m, so at ASCAT derived winds (with ERA5). The ASCAT coastal product is only valid 15 km or more out of the coast as far as I know...**

Manuscript modified according to referee suggestion.

As suggested by the referee, the approximate distance to shore is now indicated in Figure 14 (Figure 13 in first submitted version of the manuscript).

41) *Line 404-405: highlighting the different shear resemble obtained from each of the datasets and their different representation of the wind profiles .*

    i.     **Sentence unclear: shear resemble?**

This sentence was already modified during the first reviewing round: "highlighting the different shear exhibited by each of the datasets and their different representations of the wind profiles".

42) *Line 406: (mentioned in previous review: seems like a good idea to write that your results are conform what others have found): Bias ERA5 at hub height 0.5 m/s is also what is found on the North Sea in Characterisation of offshore winds for energy applications — Research@WUR and at Cabauw in Energies | Free Full-Text | Dutch Offshore Wind Atlas Validation against Cabauw Meteomast Wind Measurements (mdpi.com). NEWA comparable to ERA5 (at least on the North Sea). Undisturbed winds in DOWA (2008-2018) and WINS50 (2019-2021) are much better than ERA5 (including correlation) and the domain covers most of the Baltic Sea, but hourly data unfortunately not available for 2022 and 2023 when you have the lidar measurements (Home | Dutch Offshore Wind Atlas; WINS50 - Winds of the North Sea in 2050).*

Both references were added, as suggested by the referee, during the first reviewing round. "…ERA5 consistently underestimates the wind speed by approximately 0.5 m s-1 throughout the entire profile, which aligns with the findings of previous studies (Kalverla, 2019; Knoop et al., 2020; Rubio et al., 2022)".

43) *Line 408-409: ERA5 usually underestimates the wind speed, this is more pronounced at higher elevations and in the eastern part of the ship track. In contrast, ASCAT mainly overestimates compared to the lidar (typo) measurements.*

    i.     **If anything: more pronounced in western part of ship track (not eastern) which also makes more sense with prevailing westerly winds (land contamination ERA5 grid surface roughness)**

The mentioned typo was already corrected during the first reviewing round.

It has been corrected though, that the more pronounced underestimation mentioned in the manuscript is visible in the western part (not in the eastern).

44) *Line 418-419: When comparing the two datasets, ERA5 shows a smaller nRMSE in the majority of the studied region, except in the Eastern area near the harbour in Hanko.*

***What is your explanation for this? Does it have anything to do with time of overpass ASCAT, the location characteristics?***

An explanation on this has been added in the Section 2.2 of the manuscript: " *…except in the eastern area near the harbour in Hanko. This may be attributed to the differing spatial resolutions of the two datasets. In the east of 22 degrees longitude, the finer resolution of ASCAT mitigates the impact of coastal contamination, enabling it to capture local conditions more effectively and consequently leading to a lower average nRMSE in this region. In contrast, the coarser resolution of ERA5 may be insufficient to adequately represent the average wind characteristics in this area. Conversely, in the western part of the studied area, with features more intricate topography and a higher density of small islets within a few tens of kilometres from the mainland shoreline, ASCAT measurements are more susceptible to coastal contaminated. This results in excessively high wind measurements at 10 m, thereby contributing to larger nRMSE values across the whole profile*.".

45) ***Line 419-421: When comparing the bias and nRMSE shown by the two datasets, the average absolute bias across the entire region is smaller for ASCAT compared to ERA5 at the three heights considered (see Fig. 13). Differently, as can be observed in Fig. 14, most of the locations reveal a smaller nRMSE for ERA5 than for ASCAT. Bit confusing. I suggest an alternative text: So for all heights considered the bias (compared to the lidar measurements) of the ASCAT-derived wind speeds is smaller than the bias of the ERA5 wind speeds (fig 13), but for most of the region (except for the eastern part of the region near the Finnish coast) the nRMSE of the ERA5 wind speeds is better (fig 14).***

Text excerpt modified for clarity.

46) ***Line 427-428:*** ***The objective of this study has been to evaluate the accuracy of ASCAT-derived wind speed profiles for the characterization of offshore wind resources at turbine operating heights in the Northern Baltic Sea.***

    i.    **Goal wind resource assessments?**

        **As I said before, this work is interesting for wind energy, but only because we can use the ship-based lidar measurements for validation of mesoscale or LES models that include the effect of wind farms. We can then use these models with changed wind farm scenarios to predict the wind resource in the future. Bear in mind that mean values of the wind are not relevant if you want to predict power: you need to look at correlation on a 10 min (or hourly) basis, especially for wind speeds between cut-in and rated (power curve).**

No, the goal of this paper is not wind resource assessment, understood as the characterization of wind farms potential. This is discussed in several sections of the manuscript, and detailly explained in response to comment 1b.

47) ***Line 431: … obtained from a (typo) novel ship-based lidar campaign***

This typo was already corrected during the first reviewing round.

48) ***Line 435: … that machine learning-based techniques for extrapolating satellite winds could surpass the long-term correction method employed herein. Questionable English. I suggest an alternative text: … that machine learning-based techniques for***

*extrapolating satellite winds could work better than the long-term correction method that was used in this study.*

Corrected as suggested by referee.

**49) Line 436-437: *However, the limited amount of data available over the campaign period hinders the implementation of such data-driven approaches.***

  i.  Not an ML expert, but is the fact that you have a short campaign really the limiting factor? You have ERA5 and ASCAT measurements for a much longer period, so can you not perform your long-term stability correction? What I do know is that ML cannot reproduce events that have not occurred yet (extremes).

The comparison presented in this study is specifically focused on the campaign period, as the mean values from the lidar measurements are compared against the corresponding ERA5 mean values and the extrapolated ASCAT wind speeds—both derived for the same time frame. The extrapolation of ASCAT data is based on mean 10 m wind speeds and ERA5 stability information during the campaign period. Therefore, even though the total temporal extension of ERA5 and ASCAT is larger, it is not directly applicable to this study, since the average stability conditions are during this longer period would be different to those present specifically during the campaign.

We believe that the limited campaign duration does indeed present a significant constraint, since ML models are data-based models that require large amounts of data to effectively capture the complex relationships between input and output variables. Additionally, the methodology employed in this study to extrapolate ASCAT wind speeds produces only mean values, without providing a one-to-one, time-correlated comparison between the three datasets. This further reduces the available comparable data, making it even more challenging to develop or apply ML models for this purpose (or triple collocation, because of the same reason).

That being said, there are studies with different scopes and data availability that focus on the development and evaluation of ML algorithms for ASCAT extrapolation, such as (Optis et al. 2021; Hatfield et al. 2023). But ML is out of the scope of this study, since an alternative approach is presented and employed.

**50) Line 441-442: *The methodology revealed a remarkable congruence between these two approaches across most of the area examined, thus underscoring the robustness of the methodology.***

  i.  Not convinced this conclusion is justified (see earlier comments).

Sentence clarified: "These two approaches shown a remarkable agreement across most of the area examined, highlighting the robustness of the mean stability correction approach independently of the strategy selected".

**51) Line 443-446: *This divergence can be attributed to the limited availability of valid wind speed measurements in the collocated approach, the constraints of considering atmospheric conditions solely during morning and evening hours, and the generic definition of the empirical constants C± required for the calculation of the theoretical stability distributions at each site.***

i.     Rethink this conclusion also based on earlier remarks
We believe, as explained in Section 3.1 that the main factors to explain the differences between the collocated and full campaign approach are those mention here. The sentence has been rewritten for conciseness and clarity:

"This divergence can be attributed to the limited availability of valid wind speed measurements in the collocated approach, which slightly affects the 10 m ASCAT mean values extrapolated, differences in the predominant stability conditions between the two approaches due to the temporal discretization of ASCAT overpasses, and the generic definition of the empirical constants $C\pm$ for calculating the theoretical stability distributions at each site."

**52) Discussion: please rewrite given all comments given (running out of time to give detailed comments)**
Discussion revised.

**53) Line 486-492: Finally, it is imperative to highlight that although the disparities in wind speeds between ASCAT and ERA5 relative to lidar are generally small in far-offshore regions, their cumulative impact over a large-scale wind energy project can still have relevant implications for energy production estimates and financial assessments. Therefore, continued efforts to refine both satellite based measurements and numerical models are essential to enhance the accuracy of wind resource assessments for offshore wind energy applications. The diverse characteristics and insights into wind patterns derived from satellite-derived observations, numerical models, and ship-based lidar measurements suggest that an integrative approach, harnessing the collective strengths of these datasets, could yield substantial gains in the accuracy and reliability of offshore wind statistics derivation.**

ii.     The ASCAT measurements extrapolated to 100m with ERA5 are not representative for wind in or near wind farms and therefore do not give accurate wind resource assessments (neither does ERA5 for areas with wind farms or Measure Correlate Predict for areas where the number/size of wind farms is changing). So what we need to do is further develop Numerical Weather Prediction models that include solving the effect of wind farms (for which we need measurements for validation) and run these models for current and future wind farm layouts. ML is a useful tool, but cannot be used to derive extremes in wind climate.

You should also bear in mind that there is no significant trend in the wind climate (apart from at 10m over land) but a strong Inter Annual Variability (IAV). This is the case for the North Sea, but most likely also for the Baltic? Do you know? If there is a strong IAV, then it is important to assess how representative the period you look at is for the wind climate. For the Dutch part of the North Sea (DEEZ) the IAV is 3.5 and 4% for sites in the northern part of the DEEZ and between 4 and 4.5% in the southern part of the DEEZ (Inter-annual wind speed variability on the North Sea | Report | KNMI Projects). Is any information like this available for the Baltic Sea? How representative is 28-6-22 until 21-2-23 for the wind climate in the Baltic Sea? This you can check e.g. with ERA5 data (compare ERA5 28-6-22 - 21-2-23 to ERA January1940-now).

*So what is the added value of having these 100m wind speeds based on ASCAT? Compared to lidar, the ASCAT derived 100m wind are maybe more accurate than those from ERA5, but only available twice a day. Should we just not assimilate ASCAT in ERA5 and focus more on how useful this ship based lidar technique is to get validation measurements for models including wind farm effects (wakes/blockage)? That is what I like about this work.*

The conclusion regarding whether ASCAT and ERA5 are representative of wind conditions in or near wind farms cannot be drawn from the results of this study, as the data presented do not include nearshore wind farms. Therefore, this is a consideration that must be addressed separately when using datasets affected by phenomena like wind farm wake effects. However, while ERA5 and ASCAT do not directly account for such effects, it is still possible to estimate them using physical or numerical models, such as eddy viscosity or machine learning (ML) models, in combinations with these datasets. Although the accuracy of this methodology remains relatively limited, they still provide valuable insight when no other data are available, especially in the absence of in situ measurements. This suggests that while it may not be the ideal or preferred method, it can still serve as a viable alternative in certain situations. Nevertheless, wake effects lie beyond the scope of this study.

As previously mentioned in our response to the referee, inter-annual variability is not a relevant factor in this study, as the analysis only covers data from the same time frame (the measurement campaign period) across all three datasets. We have not conducted any assessment of IAV trends for this specific period or region, as it falls outside the intended focus of this work and does not play any role in the findings presented.

**Publication bibliography**

Hatfield, Daniel; Hasager, Charlotte Bay; Karagali, Ioanna (2023): Vertical extrapolation of Advanced Scatterometer (ASCAT) ocean surface winds using machine-learning techniques. In *Wind Energ. Sci.* 8 (4), pp. 621–637. DOI: 10.5194/wes-8-621-2023.

Optis, Mike; Bodini, Nicola; Debnath, Mithu; Doubrawa, Paula (2021): New methods to improve the vertical extrapolation of near-surface offshore wind speeds. In *Wind Energ. Sci.* 6 (3), pp. 935–948. DOI: 10.5194/wes-6-935-2021.

---

## Author Response (AR3)

**"Ship-based lidar measurements for validating ASCAT-derived and ERA5 offshore wind profiles"**
*Rev v3*
Hugo Rubio, Daniel Hatfield, Charlotte Bay Hasager, Martin Kühn, and Julia Gottschall
**Authors response to reviewer comments**

We would like to thank the referees once again for their time and effort in reviewing our work. We appreciate their feedback and comments, and we have carefully considered their recommendations and concrete suggestions to enhance and clarify our work.

Below, we addressed the additional referees' comments and reply to them point by point. First, the referee's comment is included in italics and bold font, followed by our answer, and when applicable, we have included an excerpt of the revised version of the manuscript (highlighted in blue).

Ine Wijnant, Referee #2

1. ***Abstract: Text still gives the impression that this work contributes to "accurate characterization of offshore wind resources". Maybe this is true for the Baltic for now, but certainly not for places with wind farm (effects). I suggest this alternative text:*** *Because offshore in-situ wind measurements at turbine operating heights are scarce, ECMWF Reanalysis 5th generation (ERA5) data are often used for offshore wind resource assessments. There are however a few disadvantages of using ERA5: it has a rather course grid spacing which makes it less useful for coastal areas and it does not include wind farm effects, so it can only be used for wind resource assessments in areas without wind farms.* ***This study presents a comprehensive comparison between wind profiles derived from the*** *satellite-based* ***Advanced Scatterometer (ASCAT)*** ** ***and*** ** ***ERA5*** ** ***against ship-based lidar measurements in the Northern Baltic Sea*** *for a period without wind farms.* ***The aim is to investigate the applicability of ship-based lidar measurements for validating these datasets and to better understand the reliability, accuracy and limitations of ASCAT-and ERA5-derived wind statistics for offshore wind characterization at wind turbines operating heights*** *when there are no wind farms.* ***To extrapolate ASCAT observations*** *at sea level to turbine rotating heights,* ***a*** *mean correction of atmospheric stability effects based on ERA5 and a* ***probabilistic adaptation of the Monin-Obukhov similarity theory (MOST)*** * were* ***implemented. The comparison between the two gridded... etc***

   The aim of the paper is clearly stated in the abstract already "The aim is to investigate the applicability of ship-based lidar measurements for validating these datasets and to better understand the reliability, accuracy and limitations of ASCAT- and ERA5-derived wind statistics for offshore wind characterisation at wind turbines operating heights."

   Additionally, as mentioned in previous review rounds, noted in earlier review rounds, wake effects are not relevant to the objectives or findings of this study. We fully agree with the referee that the validity of our results applies to regions without wind farms, and we have made this clear in the revised manuscript for instance, in the Introduction ("…offshore wind characterisation at wind energy-relevant heights in areas without

wind farms.") and in the Concluding discussion ("…both ERA5 and ASCAT must be approached with caution due to their inherent characteristics, including insufficient spatial resolution and the inability to adequately capture wind farm wake effects…").
In our view, repeating this clarification 3 times more again in the abstract is unnecessary, as wake effects do not factor into the presented results. Moreover, the known limitations of ERA5 in modeling wake losses are already well-known in the literature and do not directly influence the findings in this context. Including several times an explicit statement in the abstract regarding wake effects may suggest they were a central element of the study, although they are not.
The rest of the abstract has been adjusted according to referee suggestions.

2. *Line 44-45: Each NWP model comes with inherent limitations due to factors like grid resolution, physical modelling, and parameterization choices (e.g. wind farm parametrisations or the lack thereof).*
Corrected.

3. *Line 46-47: "However, conducting such validation is particularly challenging in deep-water offshore regions, where in situ measurements are sparse.". There are fewer measurements at sea than on land, but I am not convinced there are fewer measurements in deep than in shallow water… what did you base this on?*
We have clarified this in the manuscript. We are referring to the lack of measurements at turbine operating heights in deeper waters, given the high costs and logistical difficulties of deploying mast- and buoy-based systems in these areas: …challenging in deep-water offshore regions, where in situ measurements at wind turbine operating heights are sparse.

4. *Line 77-79(typo): To the authors' knowledge, this study represents the first comprehensive comparison of vertically extrapolated ASCAT wind (hereafter referred to as ASCAT wind profiles)  to wind turbine operational heights against non-stationary in situ measurements, covering locations near the coast and further offshore. *
Corrected.

5. *Line 80-82: Therefore, this work aims to contribute significantly to a better understanding of the reliability, limitations, and accuracy of satellite measurements derived wind statistics and ERA5 wind data for offshore wind characterization at wind energy-relevant heights in areas without wind farms.*
Corrected.

6. *Line 87: The discussion of these findings and the main  conclusions are included in Sections 4 and 5, respectively.*
Corrected.

7. *Line 89-91: This section describes the three datasets used in this work. In addition, the methodology used for processing the different 90 datasets is explained in detail , as well as the methodology to extrapolate ASCAT winds and the collocation approach used for their comparison against the ship-based lidar measurements.*

Corrected.

8. **Line 104: The campaign took place from 28 June 2022  until 21 February 2023**
Corrected.

9. **Line 105: … ship-based lidar system was used with  a vertical profiling Doppler lidar WindCube WLS7v2, …**
Corrected.

10. **2.1 Ship based lidar measurements: I still miss info on the accuracy of the measurements from the WindCube WLS7v2 in this section. Also what you added in lines 567-573(answer to my question 16) is not really info on accuracy. So you assume (or know? reference?) that the accuracy of a ship-based lidar with motion recorder is comparable to the accuracy of a floating lidar? Add in section 2.1: the accuracy of a ship-based lidar with motion recorder is (assumed to be) similar to the accuracy of a floating lidar. According to Dhirendra et al (2016) this is 3.1%-4.2% for heights of 92m in the wind speed range 4m/s-16m/s (pg 15 TNO report -DOWA validation against offshore mast and LiDAR measurements | Report | Dutch Offshore Wind Atlas).**
It is difficult to include specific uncertainty values to ship-based lidar systems, as their non-stationary nature hinders direct comparisons with reference measurements typically used for uncertainty quantification (as stated in the Introduction …before ship-mounted profiling lidar systems can become a generally accepted alternative …, specific challenges need to be overcome, such as the validation of these data against reference measurements and the quantification of the associated uncertainty). The main sources of uncertainty in ship-based lidar measurements are (1) the intrinsic lidar measurement uncertainty, (2) the effect of tilting, and (3) the translational motion of the ship. While the first two components are well studied in the context of floating lidar validation, the third is, to our knowledge, not yet fully quantified in the literature, but we expect will not have a great impact. Therefore, we assume that the overall uncertainty of ship-based lidar equipped with a motion compensation system is similar to that of a floating lidar. This assumption is supported by previous studies showing good agreement between ship-based lidar measurements and met mast data from FINO1 (G. Wolken-Möhlmann et al. 2014), as well as by other comparisons between ship-based lidar measurements and independent datasets (Zentek et al. 2018; Rubio et al. 2022; Zhai et al. 2018).
We have clarified this in the conclusion section of the manuscript (rather than in Section 2.1, since the uncertainty of ship-based lidars is discussed in this section). We have also included a citation to the recommended reference (Dhirendra et al. 2016) to provide a quantitative benchmark for floating lidar uncertainty.

11. **[ Line567-573: "Consequently, the mean values derived from lidar measurements may exhibit biases that vary depending on the time slots during which measurements were acquired at particular locations. Additionally, it is acknowledged that lidar measurements, like any other observational data, are subject to inherent uncertainties that may impact the results (Duncanet al., 2019b; Rubio and Gottschall, 2022). Nevertheless, the observed deviations between the lidar measurements and both extrapolated ASCAT and ERA5 significantly exceed the**

*magnitude of potential discrepancies attributable to floating lidar uncertainties (at turbine rotor heights roughly 3-4% see section 2.1), which can be up to approximately 2 % with mast-mounted anemometers as lower limit reference (Wolken-Mohlmann et al., 2022)".]*

We have added the suggested reference from the reviewer and added the corresponding reference (Dhirendra et al. 2016).

12. **Line 150: What is a nadir gap?**
    The nadir gap is the area directly beneath the satellite's path where no data is collected (see: The Xynthia Storm by ASCAT, Scatterometry | Learning Weather at Penn State Meteorology).

13. **Line 170-172: Despite the application of these quality filters, *ASCAT seems to overestimate wind speeds*  *near the coast (as shown later in this report in fig 12)*, likely due to coastal contamination effects (Stoffelen et al., 2008; Lindsley et al., 2016).**
    Corrected.

14. **Line 179-180: provides hourly estimates of a wide range of atmospheric, land surface and oceanic variables with a 0.25° x 0.25° latitude-longitude grid resolution *(31x31 km)*, covering the period from 1950 to present.**
    A resolution of 0.25° × 0.25° does not necessarily correspond to 31 × 31 km, as it depends on latitude. In fact, in the study area, this 31 × 31 km measurement would be significantly smaller. We have, however, included the specific ERA5 resolution in kilometers in the newly added Table 1.

15. **Line 211-212: *So there are two methods for stability correction: mean stability correction and instantaneous stability correction*. Compared to the instantaneous stability correction approach, applying the mean stability correction avoid*s* the need to calculate wind speeds under stability …**
    Corrected.

16. **Line 218-221: *Another advantage of the*  mean stability correction *is that the* numerical models *used for this method* can accurately capture average meteorological conditions over extended periods (Peña and Hahmann, 2012). *The stability information of data used for instantaneous stability correction is (generally?) less accurate because the measurements are for a single location or a limited time span. This introduces ,*  additional uncertainty to extrapolated profiles using this instantaneous data (Badger et al., 2012)**
    Corrected.

17. **Line 224-225: However, a relevant drawback of the mean stability correction is that *everything gets averaged out and site-or time-specific information the* information from*in-situ measurements is not included***

Corrected.

18. **Line 237: You use ERA5 to derive L and you select values of C that give NPD of 1/L closest to ERA5. Does that mean that all differences between ASCAT and ERA5 at higher levels are mainly due to differences at sea level (or 10m) because the (stability dependent) extrapolation to higher levels is equal?**
No, because ASCAT values at higher altitudes are derived using the mean stability correction method presented in the paper, while ERA5 values are obtained directly from the model output. Therefore, the differences at higher level will be due to the differences at 10 m plus the methodology to obtain mean values at higher heights, in the case of ASCAT, the mean stability correction approach. This is outlined in Section 3.2 of the paper: The difference in bias observed between 100 m and 10 m in ASCAT and ERA5 can be attributed to two key factors: first, the inherent differences between the datasets at 10 m, and second, the mean stability correction approach applied to extrapolate ASCAT.

19. **Line 250: In this study, the values for the C± constants have been set to 6 and 4 for the stable and unstable portions, respectively. These values are the same for all ASCAT grid points (both near coast and further offshore) and for the whole period (regardless of e.g. time of day and season).**
It is clearly stated in this paragraph (couple of lines below) that identical values of C± were applied across all ASCAT grid points: … Furthermore, identical values of C± were applied to all ASCAT grid points.
Additionally, we believe it is unnecessary to specify that these values were applied throughout the entire study period, as this is an inherent characteristic of the methodology used. The approach involves averaging wind profiles and stability conditions over the study period, meaning that explicitly stating this information does not add further clarity or information.

20. **Line 309-310: "This situation may lead to coastal contamination and excessively high wind speed retrievals within these grid boxes." Counter-intuitive: land contamination gives an overestimation of the surface roughness and an underestimation of the surface wind. So please explain.**
This sentence has been removed from the manuscript.

21. **Line 329: Figure 8 illustrates the differences in wind speed at 100 m height between the collocated and the full campaign approaches.**
Corrected.

22. **Line 336-338: "Consequently, the collocated approach in these areas may have insufficient stability information available, potentially introducing a biased representation of the theoretical stability distribution during the campaign period". My old question 28 has not been answered. Too few ASCAT measurements increases uncertainty but does not necessary lead to bias. Maybe the reason is that a higher percentage of land-contaminated ERA5 data are used in the collocated stability correction than the full approach?**
We have removed this sentence from the manuscript, as we believe the difference in the theoretical distribution derived from the two approaches (collocated and full

campaign) is relation with the ERA5 stability information used is better explained in the lines below: Secondly, the temporal discretization of ASCAT overpasses, occurring at roughly the same time each day, influences the resulting mean stability distribution "seen" by the collocated approach. This is…

23. **Line 344-347: As can be observed, the more unstable conditions just beforemiddayat Nynäshamn harbour due to land-contamination (red line)"weigh"more in themean stability assessmentif you just consider the collocated periods (orange shadows) instead ofthe full period.  This leads  to a larger stability correction factor in absolute terms (despite its negative sign at this location), and consequently, to lower wind speeds compared to the full campaign approach, as derived using the equations described in Section 2.4.**
Text corrected according to referee suggestion.

*My old question 30 is not sufficiently answered. I can not see in the formulas in 2.4 how a higher negative value of 1/L leads to a larger stability correction (so please explain). And how does a larger stability correction lead to lower wind speeds (and at what level)? Less unstable (more weight to surface friction effect) tends to result in a lower wind speeds at the surface: is that what you mean?*
In locations where unstable conditions occur more frequently ($n_- > n_+$), the mean stability correction factor ($\Psi_m^*$) will be greater than zero. The higher is the prevalence of unstable conditions, the larger is $n_-$, which in turn leads to an increase in $\Psi_m^*$. According to Eq. 6, for a given grid point, $\Psi_m^*$ is the only variable that changes, and therefore, the larger frequence of unstable conditions in the correlated approach leads to slightly lower values of the calculated wind speed ($U$).

The manuscript has been modified for further clarity on this point.

24. **Line 347-350: This  unstability at location A is attributed to the coarse resolution of ERA5, resulting in land contamination of the grid box at the harbour location, where  land  covers 56% of the grid box surface. Therefore, the daily stability cycle  is more  similar to that of an onshore site.**
Corrected.

25. **Line 352: The period of  highest unstability then occurs around midday when the surface heating is mostintense.**
Corrected.

26. **Line 354: Unstability reaches its  minimum (the negative value of 1/L closest to 0) in the late evening and stays relatively constant until the following morning.**
Corrected.

27. **Line 355: In contrast, locations B to E are purely offshore (with a land fraction  of 0%) and therefore exhibit almost no diurnal cycle because the atmospheric stability is mainly determined by the sea water temperature. There is however a seasonal cycle that was not taken into account. **

*stability and lower variations throughout the day, due to the presence of a relatively uniform water surface. This leads to*

Corrected.
However, we have omitted the sentence "*There is however a seasonal cycle that was not taken into account.*" since the seasonal cycle occurring outside the temporal scope of the campaign is not relevant to this study. The mean stability correction approach applied here considers only the stability conditions within the study period, which corresponds to the measurement campaign. Consequently, stability variations outside this period are neglectable, as they do not affect the in-situ measurements nor influence the mean stability conditions calculated for the extrapolation of ASCAT.

28. **Line 357-359: Finally,  at Hanko harbour (location F) there is more of a daily stability cycle than offshore, but a lot less than at Nynäshamn harbour (location A). There are two reasons for the difference between Nynäshamn (A) and Hanko (F): (1) The gridbox at Hanko (F) contains a significantly lower land-fraction: 6% compared to 56% at Nynäshamn (A)and (2) with predominantly W-SW winds, the wind at Hanko (F) is mostly from sea to land and at(Nynäshamn(A) is from land to sea. **
Suggestion from referee included.

29. **Line 364-367: Given the minimal differences in the wind speeds at 100 m depicted in Fig. 8, and thus the similar wind profiles obtained using both approaches, subsequent sections of this paper will only consider the full campaign approach because t~~his approach is expected to provide more representative wind profiles along the complete ship route.**
Rewritten for clarity.
Differences between the collocated and full campaign approaches are minimal, as demonstrated by the results. Therefore, for the sake of clarity and conciseness, we have chosen to present only one of them, as comparing both would not yield sufficiently significant differences. This does not imply that no differences exist, but rather that they are small enough to make including both approaches in the subsequent results unnecessary.

30. **Fig 9: Daily cycle of the stability parameter (1/L) at the six evaluated locations A-F from Fig. 7. All values of 1/L are below zero indicating an unstable atmosphere. Long(itude)18°corresponds to the harbour of Nynäshamn (Sweden)and long(itude) 23°to the harbour of Hanko (Finland). The orange shadows indicate the time periods when ASCAT overpasses are available and are therefore the only time periodsincluded  in the collocation approach.**
Corrected.

31. **Line 372-378: As can be observed when comparing the spatial variation shown by the two datasets at 10 m, ERA5 exhibits higher mean wind speeds in the areas farthest from the shore , but the wind speed near the coast is lower. **

*present considerably higher values of wind speed compared to ERA5. This is because ERA5 has a grid-box size of 31x31km, so part of the selected grid boxes (only grid boxeswith ASCAT data so at 12.5 km from the coast) are still land-contaminated in ERA5(and assume a surface roughness that is toohigh and therefore a 10m wind that is too low).Again, the effect of the prevailing W-SW winds can be seen: the land affects particularly the areas where the wind predominantly blows from land to sea(Swedish coast).Similar effects can be seen at 100m height. ~~This discrepancy occurs because, despite the filtering process for the ASCAT dataset, the coastal contamination still affects ASCAT measurements, leading to excessively high mean values in nearshore areas. The effect of coastal contamination in the ASCAT map is particularly visible in the 100 m height map, where the highest mean wind speeds are located along the perimeter of the region with available data.~~*

We have adjusted the manuscript as suggested by the referee. However, we have retained and revised the discussion regarding the ASCAT results. As can be seen in the figure below, ASCAT does exhibit higher 10 m wind speed values near the shore compared to offshore areas. While a detailed evaluation of ASCAT performance is beyond the scope of this study, we believe this pattern may be related to the high density of small inlets in these coastal regions, which could increase water surface roughness and therefore, increase the 10 m ASCAT wind retrievals. At 100 m, this nearshore overestimation suffered by ASCAT is more evident.

We have added and clarified this in the manuscript.

[Figure]

*Figure 1: mean ASCAT and ERA5 wind speeds at 10 m depending on distance to shore. Each scatter point represents a point of the ASCAT/ERA5 grid.*

**32. Line 397: Both datasets consistently show higher wind speeds at 100 m than at 10 m height.**

Corrected.

33. *Line 380-383: For 10m height, t  wind speed averaged over all included gridpoints is s at 10 m are 7.61 m s-1 (ASCAT) and 7.15 m s-1  (ERA5)which means that  the  difference(UASCAT− UERA5) is 0.46 ms-1. When only locations  more than 20 km from the shore are included, this difference reduces  to approximately 0.16 m s-1. However , only including locations within 20 km from the shore increases the  difference  to 0.98 m s-1.*
Corrected.

34. *Line 386-390: For  100 m height, the wind speed averaged over all included gridpoints is ,  9.31 m s-1 ( ASCAT)and 8.67 m s-1 ( ERA5) and the difference 0.64 m s-1. If only more than 20 km from shore locations are included, the difference is only slightly  reduced to 0.43 m s-1. So land-contamination in ERA5 is less relevant at 100 m height than at10 m height, which is what we expect (surface roughness affects wind at lower levels more than at higher levels). The differences between ASCAT and ERA5 at 10 and 100m  can be attributed to  two (or one?) key factors: first, the inherent difference between the datasets at 10m (e.g. the gridbox sizes: ERA5 still land-contaminated near the coast, ASCAT not), second, the mean stability correction approach used to extrapolate ASCAT; *
Corrected.

35. *Line 396-398: This  difference between in nearshore areas can be explained by the combination of excessively high wind speeds retrieved by ASCAT due to coastal contamination and ERA5's inability to properly resolve the coastal atmospheric phenomena and small-scale wind flow variations due to its coarse horizontal resolution.*
Corrected.

36. *Line 398-401: The differences become smaller  moving further offshore and almost negligible at distances further than 40 km from the shore:  around 0.2 m s-1 at 10 m height and 0.4 m s-1  at 100 m height, .*
Corrected.

37. *Line 417: mean profile bias is consistently positive (indicating ASCAT overestimation compared to the  lidar measurements), with the magnitude depending*
Corrected.

38. *Line 426: significantly outperforms ASCAT profiles, which overestimates the wind speed  even at 10 m height, highlighting the*
Corrected.

39. *Line 427-429: "Additionally, it is striking to observe the substantial deviation of the ASCAT stability corrected profiles from the logarithmic profiles, particularly at heights above 50-100 m, as a consequence of a stability distribution that is not representative enough of these specific sites".*
*I do not understand want you want to say here. The logarithmic profile represents a wind profile for neutral atmospheric stability. Do you mean that at the harbour sites the ASCAT-profile seems to follow a logarithmic profile up to 50-150m (so no stability correction occurs). What does that mean?*
Rewritten for clarity: Furthermore, a significant difference is observed between the ASCAT stability-corrected and logarithmic profiles at harbour locations, particularly above 50-100 m, mainly driven by the introduction of the stability correction factor that leads to higher wind speed estimates compared to the logarithmic profile.

40. *Line 431: A statistical analysis of the wind speed deviation between ASCAT and ERA5  compared to the lidar observations*
Corrected.

41. *Line 432-435: "Each box plot is calculated considering the wind speed difference of all the grid boxes with lidar data along the whole route of the ship, but grid boxes closer than 20 km away from the shore have been excluded to minimize the effect of ASCAT coastal contamination in the derived statistics". Why not 30 km to eliminate the land contamination in ERA5 (that you have used for stability correction)? Line 438-439 can go: you already mentioned that you did not use grid points closer than 20k from the coast.*
We have applied a 20 km threshold to ensure consistency with previously presented results, such as those in Fig. 11. Additionally, as observed in Fig. 11, the agreement between ERA5 and ASCAT at 10 m improves significantly after applying the 20 km threshold, indicating that the effects of coastal contamination are largely mitigated (ERA5 grid boxes in the Baltic are about 17 x 31 km). Nevertheless, we have compared the results shown in Fig. 13 using both 20 km and 30 km thresholds. As illustrated in the figure below, the exclusion of additional grid points (from 20 to 30 km distance from coast) has a minimal impact on the statistical comparison presented, probably likely due to their limited number and reduced coastal contamination influence.

[Figure]

*Figure 2: Fig. 13 of manuscript but filtering out grid points closer than 30 km from the coast.*

Line 438-439 has been removed as suggested.

42. ***Line 441-443: This indicates that both ERA5 and ASCAT*** ***are probably within measurement uncertainty of the lidar measurements for these heights.***
    As already mentioned, we are not able to fully quantify the uncertainty of ship-based lidar measurements, and therefore, we cannot confirm if ERA5 and ASCAT deviations are within that range. Instead, the phrasing "reasonably aligned with the lidar observations" was chosen to reflect a qualitative comparison based on observed agreement of the ERA5 and ASCAT with the considered reference dataset, the ship-based lidar, shown in Fig. 13. Additionally, this boxplot includes data from different locations along the ship route, introducing additional variability that likely exceeds the magnitude of the uncertainty attributable to the measurement system.
    We have further discussed this issue in the discussion of the paper, where we provide more context on the implications of ship-based lidar uncertainty.

43. ***Line 443-445: ERA5 consistently underestimates the wind speed across the entire profile, with this negative bias becoming increasingly pronounced with altitude and reaching the largest negative mean bias of around 0.2 m s-1 at 270 m,*** ***which is (probably) still an insignificant difference with the lidar measurements if you take into account the accuracy of the lidar measurement itself.***
    The lidar measurement uncertainty is not the only factor influencing the results presented in Fig. 13. In fact, the spatial variability introduced in the boxplot (as data across different locations along the vessel's route are included) likely exceeds the inherent uncertainty of the lidar system itself. Therefore, we do not consider a mean bias of 0.2 m/s to be negligible, as the total uncertainty in this context is considerably larger due to these additional variability sources.
    As already mentioned too (here and in the manuscript), the uncertainty associated with ship-based lidar measurements cannot be fully quantified, and thus it is not

possible to determine whether such a bias falls within the bounds of measurement uncertainty. We believe the current wording accurately emphasizes the consistent underestimation observed throughout the vertical profile without overinterpreting its statistical significance.

44. **Line 445-446: As opposed to ERA5 , ASCAT profiles exhibit a persistent overestimation of wind speed relative to the lidar across all heights.This overestimation increases significantly above 170 m.**
Corrected.

45. **Line 447-450: For ERA5, the IQR is almost the same for  all heights, with values around 0.5 m s-1, suggesting the quality of ERA5 wind speeds does not depend on height . In the case of ASCAT, IQR displays a slight decrease with height, highlighting the larger and more consistent overestimation at higher altitudes.**
Corrected.

46. **Line 451-455:The whiskers analysis provides further insights into the discrepancies between the two datasets. For ERA5, the lower whiskers extend further into negative values as altitude increases, with the larger underestimations reaching approximately -0.8m s-1at 270 m. , ASCAT's whiskers reveal a different pattern:, particularly noteworthy are the upper (positive) whiskers that extend significantly beyond the lower whiskers at 270m, illustrating once again the .  tendency for ASCAT to specifically overestimate wind speeds at  greater heights.**
Corrected.

47. **Line464-469: Notably, the western area of the ship route (longitude below 18.5 degrees) exhibits the largest errors for both ASCAT-extrapolated and ERA5 winds, with maximum differences up to about 5  m s-1 at all elevation levels. In the eastern area of the ship route, there are maximum differences up to about 4 ms-1. This indicates that wind speed estimation cannot be done accurately enough in  coastal areas using these datasets, first, because of the poor quality of ASCAT in areas closer to the coast, and secondly, due to the  ERA5 grid box size of 31 km, which means that for distances closer than 31 km to the coast the surface roughness in ERA5 gridboxes is overestimated because of land-contamination. This effect will be larger near the harbour of Nynäshamn in Sweden (longitude18°) than near the harbour of Hanko in Finland (longitude23°) because with a prevailing W-SW'lywinds, the wind at Nynäshamn blows mostly from land to sea, advecting 'landsurface roughness contamination'to sea grid points (at Hanko where the wind mostly blows from sea to land, 'watersurface roughness contamination'is advected to land grid points). Also, a fairly course model like ERA5 is  unable to capture the small-scale wind flow variations in these complex locations and the intricate interactions in the coastal boundary layer influenced by both land and sea.**
Manuscript adjusted according to referee suggestions.

*There are no ASCAT-values less than 12.5 km from the coast, so what are those "ASCAT" values in fig 14 for < 12.5 km distance from the coast? See also earlier remark about quality of ASCAT near coast (after quality control and just looking at sites > 12.5 km from the coast).*

The distances shown in Fig. 14 are approximate, and the minimum distance from the centre of the grid cells to nearest shoreline is roughly 5 km. Despite being less than 12.5 km, most of the grid box remains over water, which might explain the existence of ASCAT retrievals at these grid points. For improved clarity, we have revised the figure and its label.

48. *Line 474-475: "It can be noted that, although ERA5 usually underestimates the wind speed, this is more pronounced at higher elevations and in the western part of the ship track".*
    *Basically at a coast where there is an abrupt change of the surface roughness an internal boundary layer (IBL) is formed where the flow adjusts to the new surface roughness. This is what affects your results in coastal areas, mainly near the Swedish harbour where the wind mostly blows from land to sea. So why is the underestimation of the wind speed more pronounced at higher levels? I think because the Internal Boundary Layer (IBL) has not reached these heights yet. So basically the wind profile has not adapted to the surface roughness of the sea. Internal Boundary Layer (IBL): Internal boundary layer growth following a step change in surface roughness | Boundary-Layer Meteorology . The height of the IBL grows the father away you are from the place where the surface roughness changes (coast). So at the Swedish harbour with a W-SW wind, the wind speed adapts to the lower sea surface roughness in the IBL. The further away from the coast, the higher this IBL.*

    Thanks for the clear explanation. This has been now included in the manuscript.

49. *Line 484-490: "When comparing the two datasets, ERA5 shows a smaller nRMSE in the majority of the studied region, except in the eastern area near the harbour in Hanko. This may be attributed to the differing spatial resolutions of the two datasets. In the east of 22 degrees longitude, the finer resolution of ASCAT mitigates the impact of coastal contamination, enabling it to capture local conditions more effectively and consequently leading to a lower average nRMSE in this region. In contrast, the coarser resolution of ERA5 may be insufficient to adequately represent the average wind characteristics in this area. Conversely, in the western part of the studied area, with features more intricate topography and a higher density of small islets within a few tens of kilometres from the mainland shoreline, ASCAT measurements are more susceptible to coastal contaminated."*
    *How significant are the differences that you find in figure 15. Also, I do not think you should put all heights together. At 10m ASCAT is ASCAT, at other heights ASCAT is not ASCAT, but you have used ERA5 data for stability correction. So you can only compare spatial resolutions at 10 m height. Due to land contamination, ERA5 (31 km) will always lose from ASAT (12.5 km) in joined grid points closer than 31 km from the coast. At other heights than 10m you include ERA5 data, so affectively make the spatial resolution courser. The only reason why one location might on average be more land-contaminated than another is prevailing wind direction (if wind blows mostly from sea to land, there is less land-contamination). Please rewrite this (and*

*the rest of this) section or leave it out. You use the grid box land fraction argument to say that ASCAT is better than ERA5 near the Finnish harbour (I get that) and ERA5 is better than ASCAT near the Swedish harbour ( that I think is wrong)???*

As suggested by referee, this part has been removed from the manuscript.

50. **Line 510-513: *For the mean stability correction methodology, we had to decide whether we would use the collocated or the full dataset. A disadvantage of the collocated dataset is that the stability information may be biased because ASCAT overpasses only twice a day at roughly the same time.***  ***The collocated and full dataset***  ***strategies demonstrated remarkable agreement across most of the examined area, resulting in very similar wind speed***

Referee suggestion included, and the rest of the Discussion has been also updated according to referee previous suggestions.

The referee also questions the necessity of having a Discussion section, so we have decided to revise this, and merge both the Conclusion and Discussion into a final section named Concluding discussion, as done in other manuscript published in this journal.

51. *Conclusion: Not checked. Needs to be adapted after revisions.*

Conclusion revised and merged with Discussion into a single section: Concluding discussion.

1. *Page 5, Figure 2: Although a height of 100 m is used as the reference height in this study, the hub height of recent wind turbines often exceeds 100 m. It would be helpful to explain why you chose 100 m as the reference height.*
   Thank you for the comment. We have chosen 100m as reference for several reasons. First, 100 m is a widely used standard in wind energy research, primarily due to the height limitations of existing meteorological masts commonly used for validating numerical models and other measurement techniques. Therefore, using this height helps to be consistent with previous studies and facilitates the comparison of our results with existing literature.
   Secondly, while it is true that some recent offshore wind projects have hub heights slightly above 100 m, the difference is not substantial. For example, the global capacity-weighted average hub height in 2023 was approximately 124 m, according to the NREL Offshore Wind Market Report: 2024 Edition (see Offshore Wind Market Report: 2024 Edition). Therefore, 100 m remains within the typical hub height range of many operational offshore turbines and is still representative for slightly higher turbines.
   Finally, we note that the results at 100 m are also representative of nearby heights. As shown in Section 3.3, there are no significant changes across the intermediate height levels, and the trends discussed in Section 3.2 are consistent throughout the vertical profile. While some quantitative differences would be expected between, for example, 100 m and 130 m, the qualitative conclusions remain unchanged.

2. *Page 20, Figure 20: Since the ERA5 profiles above the surface layer are calculated using the PBL scheme, they exhibited more natural profiles than ASCAT-based dataset profiles as shown in Figure 12. Rather than using surface parameters from ERA5 for the vertical extrapolation of ASCAT, a simpler and potentially better approach might be to combine the surface wind field from ASCAT with wind profiles from ERA5.*
   We appreciate referee suggested approach and recognize it could be a valuable alternative. However, as any other alternative, this approach also introduces additional assumptions that could challenge its potential. For instance, the assumption that ERA5 wind profiles are sufficiently accurate at each location. This is particularly challenging in nearshore areas, where ERA5's coarse spatial resolution often leads to coastal contamination of grid cells.
   While we acknowledge that other methodologies exist and may be promising, they fall outside the scope of this manuscript. Here, we have chosen to focus on the mean stability correction approach for several reasons:
   - The limited duration of the measurement campaign, together with the coarse temporal resolution of ASCAT (approximately two observations per day), results in insufficient data availability for applying data-driven approaches such as triple collocation or machine learning algorithms. SAR measurements were contemplated as an alternative to ASCAT due to their higher resolution and potential better performance in near-shore areas. However, given that SAR's lower temporal resolution (one overpass every couple of days) and the relatively short period of the campaign, we opted for ASCAT in order to maximize the amount of collocated data and ensure the consistency of the statistical metrics evaluated.

- ASCAT provides wind field measurements, while ERA5 are modelled. Therefore, ASCAT winds at 10 m can often be more accurate than ERA5, especially in areas where measurements are unavailable for the assimilation and validation of the model.
- We do use ERA5 stability data due to, again, lack of measurement data available. It is not possible to estimate site-specific stability conditions from on-site measurements across the entire are included in this study and along the ship track, simply because they are scarce.
- The motivation to use the long-term correction approach comes from its potential better performance compared to the instantaneous correction approach. Studies show that while numerical models can capture average meteorological conditions well (Peña and Hahmann 2012), the accuracy of instantaneous stability information from these datasets is less reliable, adding uncertainty to the extrapolated profiles (Badger et al. 2012). Another advantage of the long-term stability correction is that we avoid calculating wind speed for conditions and heights outside the valid range of the MOST model. MOST is tailored to characterize turbulent fluxes within the surface boundary layer (Lange et al. 2004; Högström et al. 2006), but struggles with instantaneous data analysis, especially in stable conditions. The long-term adaptation of MOST, however, remains effective up to turbine operating heights, as it falls within the range where MOST is applicable.

In summary, the approach we used was selected after carefully considering the available data and the limitations of alternative methods. While different techniques may offer specific advantages, their suitability depends on data availability and the specific objectives of the study. We believe our methodology offers a balanced and practical solution for the assessment of ASCAT-based wind profile extrapolation, and their validation against ship-based lidar measurements.

3. *Additionally, the abstract currently concludes by highlighting the issues with the extrapolation method for ASCAT surface wind speeds. However, it would be more effective to end on a positive description, emphasizing the advantages of ASCAT-based offshore wind resource assessment.*
This last sentence has been removed from the manuscript so the abstract focuses in introducing the study and presenting the main results.

4. *Page 25, Line 528–533: It is stated that there is a negative impact up to 40 km from the coast, but this seems rather extensive given the ASCAT data resolution of 12.5 km. This could lead to the impression that this dataset is unusable in near-shore areas. Considering the potential application of ASCAT data, it might be beneficial to examine this issue more carefully.*
This assertion referred to the comparison between ASCAT and ERA5 has been clarified now. We believe is not only due to ASCAT overestimation in nearshore areas, but also due to ERA5 coarse grid (about 17x30 km in the Baltic Sea region) and coastal contamination of sea grid boxes. This has been clarified in the manuscript: …stabilizing at approximately 0.2 m/s and 0.4 m/s at 10 m and 100 m, respectively, in grid cells beyond 40 km from the coast. These larger nearshore discrepancies can be attributed

to the inherent limitations of both datasets. For ERA5, its coarse spatial resolution leads to land contamination in grill cells near the coast, overestimating the surface roughness, and consequently underestimating wind speeds. Furthermore, ERA5's resolution limits its ability to simulate coastal atmospheric dynamics and small-scale wind flow variations, particularly in areas with abundant small islands and rocky islets, which are especially common in the coastal regions analysed in this study (Dörenkämper et al. 2015; Gualtieri 2021). In contrast, ASCAT tends to overestimate wind speeds in some coastal areas, potentially due to effects such as wave breaking and surface slicks (Johannessen 2005; Kudryavtsev 2005) caused by the large number of small islets that result in excessively high wind field retrievals at 10 m.

5. *Page 25, Line 524–540: It would be beneficial to describe the relationship between the validation results of ASCAT and ERA5 in this study and previous research. Additionally, instead of listing numerical values in the main text, presenting them in tables would make the information clearer and easier to understand.*
A couple of references for comparison have been added to the manuscript (This larger overestimation of ASCAT in the coastal areas of the Baltic Sea has also been reported by (Hasager et al. 2020) as well as the better agreement in far from some regions between ERA5 and ASCAT, but an increased bias of around 0.6 m/s in the North Sea's coastal regions, as reported by (Duncan et al. 2019).)
We have not added a table, but we hope the revised version of this section is now clearer and more accessible to the reader.

**Minor comments:**

6. *Page 5, Line 103–104: In Section 2.1, it would be beneficial to include information on the accuracy of the ship-based LiDAR, particularly regarding whether there is any difference in accuracy compared to fixed LiDAR systems.*
We have included a short discussion and clarification on ship-based lidar uncertainty in the Concluding discussion section of the paper.

7. *Page 5, Figure 2: It would be more informative if Figure 2 were improved by using a geobubble chart or other visualization methods to plot the data on the map.*
We opted to include a 2D histogram plot (Fig. 1b) because it allows to display both the temporal and spatial distribution of the lidar measurements. While a geobubble plot might more clearly show the locations with greater or fewer data points, it would not provide information about the time distribution of them, which is an essential characteristic of our ship-based lidar campaign conducted onboard a regularly scheduled ferry boat.
The 2D histogram might be a less straightforward alternative, but it effectively illustrates both when and where measurements were taken, providing a clearer insight into the dataset's structure. We believe this visualization better supports the interpretation of our results. In addition, Fig. 6 already shows the number of lidar recordings along the ship track.

8. *Page 6, Line 138–149: It would be helpful to include a comparison table of ASCAT and ERA5 specifications in the Data and Methods section (Section 2).*
We have included a table indicating the main specifications of these two datasets.

9. **Page 15, Figures 7 and 9: In Figure 7, points A–F are indicated, but longitude is used in the figure's labeling. It would be better to ensure consistency in notation.**
Figure 9 legend has been modified to ensure consistency.

**Publication bibliography**

Badger, Merete; Peña, Alfedo; Bredesen, Rolv Erlend; Berge, Erik; Hahmann, Andrea N; Badger, Jake et al. (2012): Bringing satellite winds to hub-height. In *Proceedings of EWEA 2012 - European Wind Energy Conference & Exhibition European Wind Energy Association (EWEA), Copenhagen, Denmark, 16-19 April 2012*.

Dhirendra, Dhruv; Crockford, Anthony; Holtslag, Erik (2016): Uncertainty Assessment Fugro OCEANOR SEAWATCH Wind LiDAR Buoy at RWE Meteomast Ijmuiden. Edited by Ecofis. Available online at https://offshorewind.rvo.nl/file/download/45051422.

Dörenkämper, Martin; Optis, Michael; Monahan, Adam; Steinfeld, Gerald (2015): On the Offshore Advection of Boundary-Layer Structures and the Influence on Offshore Wind Conditions. In *Boundary-Layer Meteorol* 155 (3), pp. 459–482. DOI: 10.1007/s10546-015-0008-x.

Duncan, J B; Marseille, G J; Wijnant, I L (2019): DOWA validation against ASCAT satellite winds.

G. Wolken-Möhlmann; J. Gottschall; B. Lange (2014): First Verification Test and Wake Measurement Results Using a SHIP-LIDAR System. In *Energy Procedia* 53. DOI: 10.1016/j.egypro.2014.07.223.

Gualtieri, Giovanni (2021): Reliability of ERA5 Reanalysis Data for Wind Resource Assessment: A Comparison against Tall Towers. In *Energies* 14 (14), p. 4169. DOI: 10.3390/EN14144169.

Hasager, Charlotte B.; Hahmann, Andrea N.; Ahsbahs, Tobias; Karagali, Ioanna; Sile, Tija; Badger, Merete; Mann, Jakob (2020): Europe's offshore winds assessed with synthetic aperture radar, ASCAT and WRF. In *Wind Energ. Sci.* 5 (1), pp. 375–390. DOI: 10.5194/wes-5-375-2020.

Högström, Ulf; Smedman, Ann-Sofi; Bergström, Hans (2006): Calculation of Wind Speed Variation with Height over the Sea. In *Wind Engineering* 30 (4), pp. 269–286. DOI: 10.1260/030952406779295480.

Johannessen, J. A. (2005): On radar imaging of current features: 2. Mesoscale eddy and current front detection. In *J. Geophys. Res.* 110 (C7), p. 245. DOI: 10.1029/2004JC002802.

Kudryavtsev, V. (2005): On radar imaging of current features: 1. Model and comparison with observations. In *J. Geophys. Res.* 110 (C7), p. 10529. DOI: 10.1029/2004JC002505.

Lange, Bernhard; Larsen, Søren; Højstrup, Jørgen; Barthelmie, Rebecca (2004): The Influence of Thermal Effects on the Wind Speed Profile of the Coastal Marine Boundary Layer. In *Boundary-Layer Meteorol* 112 (3), pp. 587–617. DOI: 10.1023/B:BOUN.0000030652.20894.83.

Peña, Alfredo; Hahmann, Andrea N. (2012): Atmospheric stability and turbulence fluxes at Horns Rev-an intercomparison of sonic, bulk and WRF model data. In *Wind Energ.* 15 (5), pp. 717–731. DOI: 10.1002/we.500.

Rubio, Hugo; Kühn, Martin; Gottschall, Julia (2022): Evaluation of low-level jets in the southern Baltic Sea: a comparison between ship-based lidar observational data and numerical models. In *Wind Energ. Sci.* 7 (6), pp. 2433–2455. DOI: 10.5194/wes-7-2433-2022.

Zentek, Rolf; Kohnemann, Svenja H. E.; Heinemann, Günther (2018): Analysis of the performance of a ship-borne scanning wind lidar in the Arctic and Antarctic. In *Atmos. Meas. Tech.* 11 (10), pp. 5781–5795. DOI: 10.5194/amt-11-5781-2018.

Zhai, Xiaochun; Wu, Songhua; Liu, Bingyi; Song, Xiaoquan; Yin, Jiaping (2018): Shipborne Wind Measurement and Motion-induced Error Correction of a Coherent Doppler Lidar over the Yellow Sea in 2014. In *Atmos. Meas. Tech.* 11 (3), pp. 1313–1331. DOI: 10.5194/amt-11-1313-2018.